# OPUS: Towards Efficient and Principled Data Selection in Large Language Model Pre-Training in *Every* Iteration

**Shaobo Wang** [* 1 2] **Xuan Ouyang** [* 1 3] **Tianyi Xu** [* 1 3] **Yuzheng Hu** [4] **Jialin Liu** [1] **Guo Chen** [1] **Tianyu Zhang** [5]
**Junhao Zheng** [2] **Kexin Yang** [2] **Xingzhang Ren** [2] **Dayiheng Liu** [2] **Linfeng Zhang** [1]

## Abstract

As high-quality public text approaches exhaustion, a phenomenon known as the *Data Wall* (Villalobos et al., 2024), pre-training is shifting from more tokens to better tokens. However, existing methods either rely on heuristic static filters that ignore training dynamics, or use dynamic yet optimizer-agnostic criteria based on raw gradients. We propose **OPUS** (**O**ptimizer-induced **P**rojected **U**tility **S**election), a dynamic framework that defines utility in the optimizer-induced update space. OPUS scores candidates by projecting their effective updates, shaped by modern optimizers, onto a target direction derived from a stable, in-distribution proxy. To ensure scalability, we employ Ghost technique with CountSketch for computational efficiency, and Boltzmann sampling for data diversity, incurring only 4.7% additional compute overhead. OPUS achieves remarkable results across diverse corpora, quality tiers, optimizers, and model scales. It also outperforms previous data selection methods across different stages of training, including from-scratch pre-training and also mid-training. Beyond online selection, the OPUS utility score also demonstrates potential as a static filter for flagging and removing toxic documents from contaminated training corpora prior to training. Code is available at https://github.com/gszfwsb/OPUS.

---

[*]Equal contribution [1]Shanghai Jiao Tong University [2]Qwen Team, Alibaba Group [3]University of Wisconsin–Madison [4]University of Illinois Urbana–Champaign [5]Mila–Quebec AI Institute. Correspondence to: Xingzhang Ren <xingzhang.rxz@alibaba-inc.com>, Dayiheng Liu <liudayiheng.ldyh@alibaba-inc.com>, Linfeng Zhang <zhanglinfeng@sjtu.edu.cn>.

*Proceedings of the 43rd International Conference on Machine Learning*, Seoul, South Korea. PMLR 306, 2026. Copyright 2026 by the author(s).

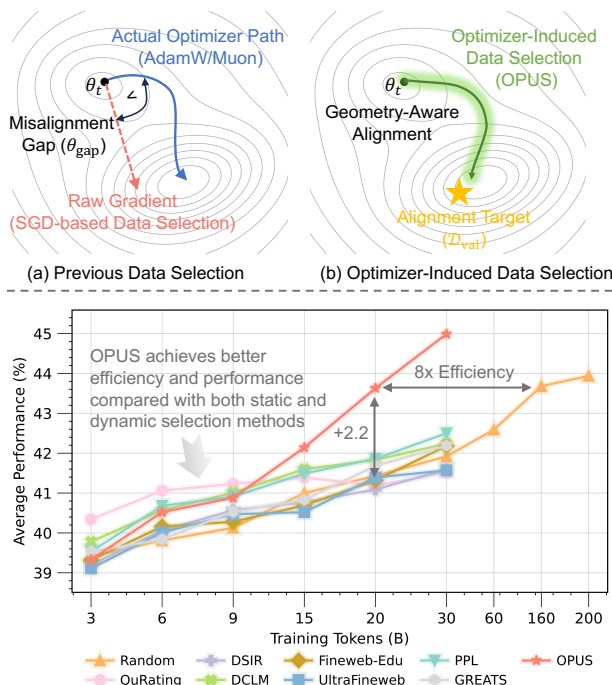

*Figure 1.* **Top:** Comparison of different data selection methods. **Bottom:** OPUS outperforms random selection by an average of 2.2% accuracy across 10 downstream benchmarks and achieves 8× reduction in computation on GPT-XL using FineWeb dataset.

## 1. Introduction

Large language model (LLM) pre-training has entered a critical phase, transitioning from an era of unconstrained data scaling to a regime where the efficiency and quality of every training token are paramount. For the past decade, progress in language modeling has been driven by scaling two primary factors: model size and data volume (Radford et al., 2019; Brown et al., 2020; Achiam et al., 2023; Yang et al., 2024a;b; 2025; Guo et al., 2025; Liu et al., 2024a; Anthropic, 2024). Scaling laws emphasize that performance is tightly coupled with the efficiency of converting compute into effective training signals (Hoffmann et al., 2022). Yet the data factor is now saturating: projections suggest that readily available high-quality public text may be exhausted by 2026–2028 (Villalobos et al., 2024). In this data-wall regime, pre-training must shift from a problem of ingestion

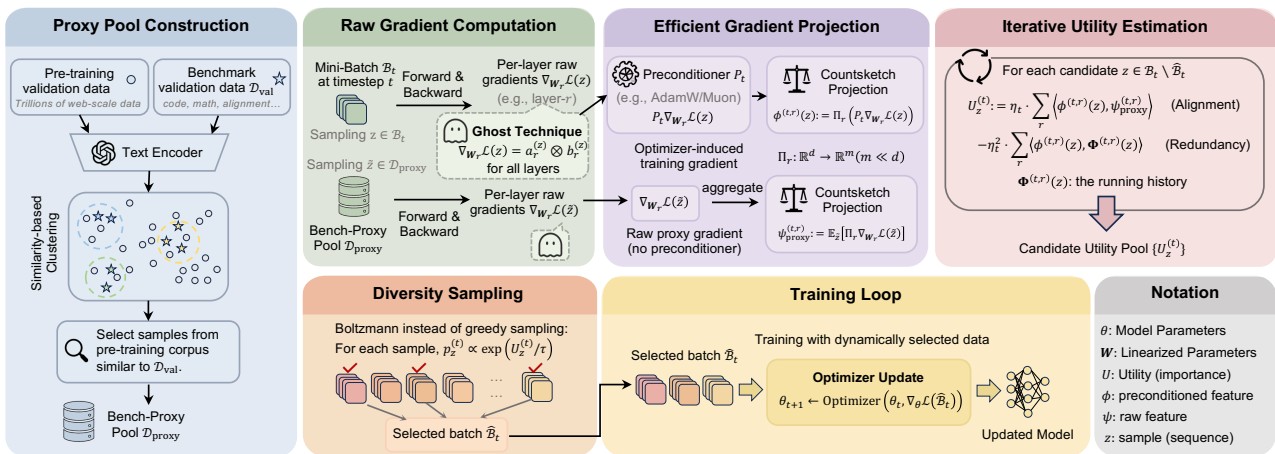

*Figure 2.* Overview of OPUS pipeline.

capacity to one of control: *which tokens should shape the model at this specific optimizer step?* When every update consumes scarce tokens, selection is no longer a preprocessing choice but an integral component of the optimization.

Existing approaches to this problem present distinct limitations. Static curation methods, such as FineWeb-Edu classifiers (Penedo et al., 2024) and the DCLM quality classifier (Li et al., 2024), rely on fixed, training-agnostic heuristics that assume a sample's utility remains constant as the model evolves. In contrast, prior dynamic selection methods (Wang et al., 2024; 2025a;b) score candidates in raw gradient space, implicitly assuming SGD dynamics. This induces a fundamental misalignment with modern LLM training, which relies on adaptive optimizers such as AdamW (Loshchilov & Hutter, 2019) and Muon (Jordan et al., 2024) that precondition and reshape the effective update direction. As shown in Figure 1, existing approaches depart from the optimizer's actual update geometry, causing unsatisfied optimization trajectory.

To bridge this gap, we introduce **OPUS** (**O**ptimizer-induced **P**rojected **U**tility **S**election), a framework designed to make data selection in pre-training both principled and scalable. OPUS achieves a principled objective by adapting during training to the model's evolving needs, unlike static filters, and by defining utility in the optimizer-induced update space. The core insight is that a batch is valuable only insofar as it moves parameters in a direction that improves the model's performance on a high-quality target distribution, referred to as the proxy, under the optimizer's specific geometry. OPUS scores each candidate by projecting its optimizer-induced effective update onto the descent direction of this proxy set, eliminating the discrepancy between scoring and training that arises when Adam or Muon training is treated as if it were SGD. To ensure scalability, OPUS estimates these utilities via lightweight projections, avoiding the prohibitive cost of materializing full gradients.

OPUS operationalizes this principle through an objective, an estimator, and a selection rule. First, we formalize utility as the expected one-step improvement on a held-out proxy distribution, measured in the optimizer-induced update geometry, so that scoring aligns with the trajectory induced by AdamW or Muon. Second, we make this objective practical at LLM scale by (i) constructing a stable, in-distribution target direction for the proxy signal and (ii) estimating the required inner products efficiently without materializing per-sample gradients. Third, we use Boltzmann sampling to preserve data diversity. Figure 2 summarizes the end-to-end workflow. Our contributions are as follows:

- **A principled, optimizer-aware utility for dynamic selection:** We introduce optimizer-induced utility as a theoretically grounded objective for dynamic data selection. By deriving closed-form approximations for the effective update directions of AdamW (Loshchilov & Hutter, 2019) and Muon (Jordan et al., 2024), OPUS scores data in the geometry that the optimizer actually follows, yielding a model- and optimizer-aware alternative to heuristic filters.
- **Stable in-distribution proxy construction:** We propose BENCH-PROXY, a procedure for constructing a proxy pool by retrieving benchmark-aligned samples directly from the pre-training corpus. This yields a reliable, in-distribution proxy direction that stabilizes utility estimation compared to using raw benchmark validation data.
- **Scalable utility estimation via ghost and CountSketch:** To make scoring efficient at LLM scale, we avoid per-sample gradient materialization by combining the ghost technique (Wang et al., 2024) with CountSketch projections (Cormode & Muthukrishnan, 2005), reducing inner products to computations in a low-dimensional space.
- **Boltzmann sampling to prevent diversity collapse:** To avoid biased or redundant selection induced by greedy top-$k$ under non-stationary streams, OPUS uses Boltzmann soft sampling with an in-step redundancy penalty.
- **OPUS achieves strong empirical gains over industrial baselines:** Across from-scratch pre-training of

GPT-2 Large/XL on FineWeb and FineWeb-Edu (Penedo et al., 2024) and continued pre-training of Qwen3-8B-Base (Yang et al., 2025) on SciencePedia (SciencePedia Team, 2025), OPUS outperforms prior industrial static filters and dynamic selectors with better efficiency.

**Conflict of Interest Disclosure.** Several authors (J.Z., K.Y., X.R., D.L.) are employed by the Qwen Team, Alibaba Group, which develops the Qwen3-8B-Base model evaluated in this paper. S.W. conducted part of this work during an internship at the Qwen Team, Alibaba Group. The remaining authors declare no competing financial interests.

## 2. Background

### 2.1. LLM Pre-Training

We consider an autoregressive language model $f_\theta$ parameterized by $\theta \in \mathbb{R}^d$. A training sample is a token sequence $z = (x_1, \ldots, x_L)$ with $x_i \in \mathcal{V}$, where $\mathcal{V}$ is the vocabulary and $L$ is the sequence length. The model defines the next-token distribution $p_\theta(x_i \mid x_{<i})$, and the per-sequence loss is the negative log-likelihood: $\mathcal{L}(z; \theta) = -\frac{1}{L} \sum_{i=1}^{L} \log p_\theta(x_i \mid x_{<i})$. For any distribution (or finite set) $\mathcal{Q}$ over sequences, we define the expected loss $\mathcal{L}(\mathcal{Q}; \theta) := \mathbb{E}_{z \sim \mathcal{Q}}[\mathcal{L}(z; \theta)]$ (or its empirical average for a finite $\mathcal{Q}$). Let $\mathcal{D}$ denote the full pre-training corpus. We partition it into (i) a training set $\mathcal{D}_{\text{tr}}$ used for parameter updates and (ii) a held-out validation set $\mathcal{D}_{\text{val}}$ used only to guide selection. Importantly, $\mathcal{D}_{\text{val}} \cap \mathcal{D}_{\text{tr}} = \emptyset$, so validation samples never appear in training updates.

### 2.2. Data Selection in Pre-Training

Data selection in pre-training aims to choose samples that compress knowledge both efficiently and effectively, which can be categorized into two domains.

**Static Data Selection.** Static methods operate offline, filtering the entire candidate pool $\mathcal{D}_{\text{tr}}$ before training begins. A scoring function $S(z)$ assigns a quality score to each sample $z \in \mathcal{D}_{\text{tr}}$. A subset $\mathcal{D}_{\text{selected}} \subset \mathcal{D}_{\text{tr}}$ is retained by thresholding or top-$k$ selection: $\mathcal{D}_{\text{selected}} = \{z \in \mathcal{D}_{\text{tr}} \mid S(z) \geq \text{threshold}\}$. The model is then trained on $\mathcal{D}_{\text{selected}}$ using a standard optimizer. While scalable, static selection ignores the model's evolving state $\theta_t$ during training.

**Dynamic Data Selection.** Dynamic methods select data during training at each step $t$, adapting to the current model parameter $\theta_t$ and optimizer state. At step $t$, the algorithm receives a candidate buffer $\mathcal{B}_t = \{z_1, \ldots, z_N\}$ of $N$ sequences from the update stream $\mathcal{D}_{\text{tr}}$. It selects a subset $\widehat{\mathcal{B}}_t \subset \mathcal{B}_t$ of size $K = \lfloor \rho N \rfloor$ (selection ratio $\rho \in (0, 1]$) to update the model: $\widehat{\mathcal{B}}_t = \text{SELECT}(\mathcal{B}_t; s_t(\cdot), K)$, where $s_t(z)$ is a step-dependent score (or sampling distribution) computed from the current model and proxy signal.

### 2.3. Modern Optimizers in Large-Scale Pre-Training

Many dynamic selection methods score candidates using the raw gradient $\nabla \mathcal{L}(z; \theta_t)$, implicitly assuming SGD-like geometry. Modern LLM training instead uses optimizers that transform gradients using state, such as momentum and adaptive preconditioning, changing the effective update direction. We write the optimizer-induced effective update at step $t$ using an optimizer-induced preconditioner (operator) $\mathbf{P}_t$ applied to per-sample gradients:

$$\Delta \theta_t(\widehat{\mathcal{B}}_t) = -\eta_t \sum_{z \in \widehat{\mathcal{B}}_t} \mathbf{P}_t \nabla \mathcal{L}(z; \theta_t). \tag{1}$$

Here, $\mathbf{P}_t$ encapsulates the optimizer state at step $t$ and induces the geometry that the training trajectory actually follows. When the optimizer's transformation is not strictly linear, $\mathbf{P}_t$ should be read as a state-dependent operator acting on the gradient. This motivates defining selection scores in the optimizer-induced geometry rather than raw-gradient space. The details of common optimizers (SGD, AdamW, and Muon) are attached in Appendix C.

## 3. Methodology: OPUS

We now describe OPUS and organize the section around the requirements that dynamic selection must satisfy in large-scale pre-training. Ideally, dynamic selection in large-scale pre-training should satisfy three desiderata:

- *Principled:* scores are derived from an explicit objective that measures improvement on a held-out proxy distribution under the optimizer-induced update geometry.
- *Efficient:* scoring avoids materializing per-sample gradients in high-dimensional space.
- *Scalable:* overhead remains modest as model dimension $m$ grows, enabling selection at every step.

Guided by these desiderata, we introduce **OPUS**, a dynamic data selection framework for LLM pre-training. At each step $t$, OPUS receives a candidate buffer $\mathcal{B}_t = \{z_1, \ldots, z_N\} \subset \mathcal{D}_{\text{tr}}$ and selects $K = \lfloor \rho N \rfloor$ sequences to form the update batch. OPUS also draws a proxy mini-batch of size $K_{\text{proxy}}$ from a proxy pool $\mathcal{D}_{\text{proxy}}$, a finite surrogate for the held-out proxy set $\mathcal{D}_{\text{val}}$. Let $\mathbf{P}_t$ denote the optimizer-induced preconditioner at step $t$. We use sketch dimension $m$ for scoring in a projected space and temperature $\tau > 0$ for stochastic sampling. **For details, please refer Algorithm 1 for the iterative OPUS algorithm.**

### 3.1. Optimizer-Induced Utility Objective

To obtain a principled scoring signal for selection, we define the utility of a candidate batch $\mathcal{S}$ as the reduction in loss on validation set $\mathcal{D}_{\text{val}}$ after one optimization step. Following (Wang et al., 2024), we define utility at step $t$ as:

$$U^{(t)}(\mathcal{S}) := \mathcal{L}(\mathcal{D}_{\text{val}}; \theta_t) - \mathcal{L}(\mathcal{D}_{\text{val}}; \theta_{t+1}(\mathcal{S})). \tag{2}$$

**Marginal gain.** At each training step $t$, we are given a candidate buffer $\mathcal{B}_t$ and aim to construct an update subset $\widehat{\mathcal{B}}_t \subseteq \mathcal{B}_t$. Let $z \in \mathcal{B}_t \setminus \widehat{\mathcal{B}}_t$ be a remaining candidate. We define the marginal utility of adding $z$ as

$$U_z^{(t)} := U^{(t)}(\widehat{\mathcal{B}}_t \cup \{z\}) - U^{(t)}(\widehat{\mathcal{B}}_t). \quad (3)$$

Let $\tilde{\theta}_t(\widehat{\mathcal{B}}_t)$ denote the *virtual parameters* obtained by applying one descent step on the selected subset $\widehat{\mathcal{B}}_t$: $\tilde{\theta}_t(\widehat{\mathcal{B}}_t) = \theta_t + \Delta\theta_t(\widehat{\mathcal{B}}_t)$. Adding $z$ induces an additional update $\Delta\theta_t(\{z\})$, so the marginal gain can be written as:

$$U_z^{(t)} = \mathcal{L}(\mathcal{D}_{\text{val}}; \tilde{\theta}_t(\widehat{\mathcal{B}}_t)) - \mathcal{L}(\mathcal{D}_{\text{val}}; \tilde{\theta}_t(\widehat{\mathcal{B}}_t) + \Delta\theta_t(\{z\})). \quad (4)$$

Using a first-order Taylor approximation of the validation loss at $\tilde{\theta}_t(\widehat{\mathcal{B}}_t)$, we have

$$\mathcal{L}\left(\mathcal{D}_{\text{val}}; \tilde{\theta}_t(\widehat{\mathcal{B}}_t) + \Delta\theta_t(\{z\})\right) \approx \mathcal{L}\left(\mathcal{D}_{\text{val}}; \tilde{\theta}_t(\widehat{\mathcal{B}}_t)\right) \\ + \nabla_\theta\mathcal{L}\left(\mathcal{D}_{\text{val}}; \tilde{\theta}_t(\widehat{\mathcal{B}}_t)\right)^\mathsf{T} \Delta\theta_t(\{z\}). \quad (5)$$

Substituting Eq. (5) into Eq. (4) yields

$$U_z^{(t)} \approx -\nabla_\theta\mathcal{L}\left(\mathcal{D}_{\text{val}}; \tilde{\theta}_t(\widehat{\mathcal{B}}_t)\right)^\mathsf{T} \Delta\theta_t(\{z\}). \quad (6)$$

**Optimizer-induced geometry.** Unlike vanilla SGD, modern LLM training relies on adaptive optimizers that reshape gradients through a state-dependent preconditioner. We denote the optimizer state operator at step $t$ as $\mathbf{P}_t$ and define the *optimizer-induced effective update direction* as

$$\mathbf{u}_z^{(t)} := \mathbf{P}_t \nabla_\theta\mathcal{L}(z; \theta_t). \quad (7)$$

Accordingly, the optimizer update induced by a subset $\mathcal{S}$ can be written as $\Delta\theta_t(\mathcal{S}) = -\eta_t \sum_{z \in \mathcal{S}} \mathbf{u}_z^{(t)}$. In particular, adding a single candidate $z$ contributes an additional update $\Delta\theta_t(\{z\}) = -\eta_t \mathbf{u}_z^{(t)}$. Substituting $\Delta\theta_t(\{z\})$ into the marginal approximation in Eq. (6) gives

$$U_z^{(t)} \approx \eta_t \left\langle \mathbf{u}_z^{(t)}, \nabla_\theta\mathcal{L}\left(\mathcal{D}_{\text{val}}; \tilde{\theta}_t(\widehat{\mathcal{B}}_t)\right) \right\rangle. \quad (8)$$

**Approximating the virtual validation gradient.** The marginal gain of adding a candidate $z$ to the current subset $\widehat{\mathcal{B}}_t$, denoted as $U_z^{(t)}$, depends on the validation gradient evaluated at the *virtual parameters* $\tilde{\theta}_t(\widehat{\mathcal{B}}_t)$. Specifically, the first-order approximation of the utility is given by the inner product between the optimizer-induced update and the gradient at the virtual point:

$$U_z^{(t)} \approx \eta_t \left\langle \mathbf{u}_z^{(t)}, \nabla_\theta\mathcal{L}(\mathcal{D}_{\text{val}}; \tilde{\theta}_t(\widehat{\mathcal{B}}_t)) \right\rangle. \quad (9)$$

Computing this virtual gradient exactly would require an additional backward pass on $\mathcal{D}_{\text{val}}$ after every selection step, which is prohibitively expensive. To avoid this cost, we linearize the gradient function $\mathbf{g}_{\text{val}}(\theta) := \nabla_\theta\mathcal{L}(\mathcal{D}_{\text{val}}; \theta)$ around

the current parameters $\theta_t$. Let $\Delta\theta_t(\widehat{\mathcal{B}}_t) := \tilde{\theta}_t(\widehat{\mathcal{B}}_t) - \theta_t$ be the accumulated update from the currently selected subset. A first-order Taylor expansion gives:

$$\nabla_\theta\mathcal{L}\left(\mathcal{D}_{\text{val}}; \tilde{\theta}_t(\widehat{\mathcal{B}}_t)\right) \approx \mathbf{g}_{\text{val}}(\theta_t) + \nabla_\theta\mathbf{g}_{\text{val}}(\theta_t)\Delta\theta_t(\widehat{\mathcal{B}}_t) \\ = \mathbf{g}_{\text{val}}^{(t)} + \mathbf{H}_{\text{val}}^{(t)}\Delta\theta_t(\widehat{\mathcal{B}}_t), \quad (10)$$

where $\mathbf{g}_{\text{val}}^{(t)}$ is the validation gradient at $\theta_t$ and $\mathbf{H}_{\text{val}}^{(t)}$ is the Hessian. Using the update rule, the accumulated update is $\Delta\theta_t(\widehat{\mathcal{B}}_t) = -\eta_t \sum_{z_j \in \widehat{\mathcal{B}}_t} \mathbf{u}_{z_j}^{(t)}$. Substituting the gradient approximation (Eq. (10)) and the explicit update form into Eq. (9), we obtain the final tractable scoring function:

$$U_z^{(t)} \approx \eta_t \left\langle \mathbf{u}_z^{(t)}, \mathbf{g}_{\text{val}}^{(t)} - \eta_t\mathbf{H}_{\text{val}}^{(t)} \sum_{z_j \in \widehat{\mathcal{B}}_t} \mathbf{u}_{z_j}^{(t)} \right\rangle \\ = \underbrace{\eta_t \left\langle \mathbf{u}_z^{(t)}, \mathbf{g}_{\text{val}}^{(t)} \right\rangle}_{\text{Alignment}} - \underbrace{\eta_t^2 \left\langle \mathbf{u}_z^{(t)}, \mathbf{H}_{\text{val}}^{(t)} \sum_{z_j \in \widehat{\mathcal{B}}_t} \mathbf{u}_{z_j}^{(t)} \right\rangle}_{\text{Redundancy Penalty}}. \quad (11)$$

**Handling the Hessian complexity.** Materializing $\mathbf{H}_{\text{val}}$ is intractable at LLM scale. Following (Wang et al., 2024), we adopt an isotropic approximation for this interaction term, $\mathbf{H}_{\text{val}} \approx \mathbf{I}$. Defining the accumulated effective direction $\mathbf{G}^{(t)} := \sum_{z_j \in \widehat{\mathcal{B}}_t} \mathbf{u}_{z_j}^{(t)}$, we obtain the practical redundancy-adjusted score

$$U_z^{(t)} \approx \eta_t \left\langle \mathbf{u}_z^{(t)}, \mathbf{g}_{\text{val}}^{(t)} \right\rangle - \eta_t^2 \left\langle \mathbf{u}_z^{(t)}, \mathbf{G}^{(t)} \right\rangle. \quad (12)$$

**Stable proxy construction via BENCH-PROXY.** The quality of the proxy direction $\mathbf{g}_{\text{val}}^{(t)}$ is critical for principled selection. While a random hold-out set provides a low-variance signal, it often fails to capture the specific distribution of downstream tasks. Conversely, using raw benchmark samples directly as the proxy introduces severe distribution shift and gradient noise, destabilizing the ranking. To bridge this gap, we introduce **BENCH-PROXY**, a retrieval-based construction shown in Fig. 2(a). We embed both (i) the target benchmark validation set and (ii) candidate documents from the pre-training corpus using a frozen text encoder, and retrieve the top-$M$ most similar pre-training documents to form an *in-distribution* proxy pool $\mathcal{D}_{\text{proxy}}$. This approach yields a proxy that is aligned with the target tasks yet remains within the pre-training manifold, ensuring valid gradient estimation. Concretely, at step $t$ we draw a proxy mini-batch $\{\tilde{z}_k\}_{k=1}^{K_{\text{proxy}}} \subset \mathcal{D}_{\text{proxy}}$ and estimate the direction via $\mathbf{g}_{\text{proxy}}^{(t)} = \frac{1}{K} \sum_{k=1}^{K_{\text{proxy}}} \nabla_\theta\mathcal{L}(\tilde{z}_k; \theta_t)$. Substituting this proxy estimate into Eq. (12), we obtain the final scoring rule:

$$U_z^{(t)} \leftarrow \eta_t \left\langle \mathbf{u}_z^{(t)}, \mathbf{g}_{\text{proxy}}^{(t)} \right\rangle - \eta_t^2 \left\langle \mathbf{u}_z^{(t)}, \mathbf{G}^{(t)} \right\rangle. \quad (13)$$

This formulation ensures that selected updates not only reduce loss but specifically align with the benchmark-relevant

subspace of the optimization landscape. Further details of BENCH-PROXY construction are provided in Appendix D.

## 3.2. Scalable Utility Estimation

To score candidates at scale, we leverage the ghost technique (Wang et al., 2024; 2025a; Hu et al., 2025) to avoid per-sample forward/backward passes and the materialization of full gradients. We further apply a low-dimensional sketch to efficiently compute the inner products required for the utility score in Eq. (13).

**Ghost technique.** Following GREATS (Wang et al., 2024), we exploit the *rank-1 outer product structure* of backpropagated gradients in linear layers. Consider a linear layer $r$ with weights $\mathbf{W}_r$. For a sample $z$, let $\mathbf{a}_r^{(z)}$ denote the input activation vector and $\mathbf{b}_r^{(z)}$ the output gradient vector (error signal). The per-sample gradient with respect to the weights factorizes as the outer product $\nabla_{\mathbf{W}_r} \mathcal{L}(z; \theta_t) = \mathbf{a}_r^{(z)} \otimes \mathbf{b}_r^{(z)}$, where $\otimes$ denotes the outer product. Since $\mathbf{a}_r^{(z)}$ and $\mathbf{b}_r^{(z)}$ are available during the standard forward/backward passes, *we can compute gradient statistics without ever materializing the high-dimensional matrix* $\nabla_{\mathbf{W}_r} \mathcal{L}$. In OPUS, we apply it over a set of layers $\mathcal{R}$ (e.g., linear and embedding matrices). We concatenate the proxy batch and candidate batch within a single forward/backward pass to collect $\{\mathbf{a}_r^{(z)}, \mathbf{b}_r^{(z)}\}$ for all samples. These quantities contain all information required to compute the projected scores, and are discarded layer-by-layer to maintain low memory overhead.

**CountSketch projection.** Computing the utility $U_z^{(t)}$ in Eq. (13) requires applying the optimizer preconditioner $\mathbf{P}_t$. We project the resulting effective updates into a low-dimensional sketch space using a sparse CountSketch map $\Pi : \mathbb{R}^d \to \mathbb{R}^m$ ($m \ll d$). For a linear layer $r$ with dimensions $d_{\text{in}} \times d_{\text{out}}$, the per-sample preconditioned sketch feature $\phi^{(t,r)}(z) \in \mathbb{R}^m$ is computed implicitly as:

$$\phi^{(t,r)}(z) = \Pi_r \Big( \mathbf{P}_{t,r} \big( \mathbf{a}_r^{(z)} \otimes \mathbf{b}_r^{(z)} \big) \Big). \tag{14}$$

We instantiate $\Pi_r$ using CountSketch (Cormode & Muthukrishnan, 2005), which enables computing the projection by streaming over the coordinates of the outer-product gradient without explicitly materializing it. This choice yields concrete computational benefits depending on the structure of $\mathbf{P}_{t,r}$. For AdamW, $\mathbf{P}_{t,r}$ is diagonal (Appendix C), preserving the coordinate-wise separable structure of the outer-product gradient. This allows the CountSketch projection to be interleaved with preconditioning by applying the diagonal weights on the fly, yielding a projection cost of $\mathcal{O}(d_{\text{in}} + d_{\text{out}})$ rather than the $\mathcal{O}(d_{\text{in}} d_{\text{out}})$ cost required for a dense projection. In contrast, for optimizers with dense preconditioners such as Muon, coordinate mixing destroys this separability, resulting in a projection cost of $\mathcal{O}(d_{\text{in}} d_{\text{out}})$. We approximate the alignment and redundancy terms by

summing dot products in the sketch space across layers:

$$U_z^{(t)} \approx \eta_t \sum_{r \in \mathcal{R}} \langle \phi^{(t,r)}(z), \psi_{\text{proxy}}^{(t,r)} \rangle - \eta_t^2 \sum_{r \in \mathcal{R}} \langle \phi^{(t,r)}(z), \mathbf{\Phi}^{(t,r)} \rangle, \tag{15}$$

where $\mathbf{\Phi}^{(t,r)} = \sum_{z_j \in \widehat{\mathcal{B}}_t} \phi^{(t,r)}(z_j)$ is the running history of selected sketches. Note that $\psi_{\text{proxy}}^{(t,r)} := \Pi_r \Big( \frac{1}{K_{\text{proxy}}} \sum_{k=1}^{K_{\text{proxy}}} \mathbf{a}_r^{(\tilde{z}_k)} \otimes \mathbf{b}_r^{(\tilde{z}_k)} \Big)$ represents the sketched *unpreconditioned* proxy gradient direction.

## 3.3. Boltzmann Sampling

To preserve diversity under dynamic selection, we replace deterministic greedy top-$k$ with stochastic sampling. While our utility formulation in Eq. (15) explicitly penalizes *geometric* redundancy (vector alignment), greedy selection remains brittle to *estimation noise*: it assumes the proxy direction $\psi_{\text{proxy}}^{(t,r)}$ is perfect. In practice, the proxy is a stochastic estimate from a small batch, and the data stream is non-stationary. Always picking the current top-$k$ can lock the model into transient, noisy features of the proxy batch. We therefore adopt Boltzmann sampling to improve robustness:

$$p_z^{(t)} \propto \exp \big( U_z^{(t)} / \tau \big). \tag{16}$$

This ensures that high-utility candidates are favored, while complementary candidates maintain non-zero probability, preventing overfitting to local proxy noise.

## 4. Experiments

### 4.1. Experimental Setup

**Models and training settings.** We pre-train GPT-2 Large and GPT-2 XL (Radford et al., 2019) from scratch under a fixed optimization budget of 30B update tokens. Unless stated otherwise, all methods are compute-matched by performing parameter updates on exactly 30B update tokens. We also evaluate OPUS in a continued pre-training setting. Specifically, we continue from a Qwen3-8B-Base checkpoint (Yang et al., 2025) on a science-domain stream, keep the training recipe fixed, and vary only the selection policy.

**Pre-training corpus.** For from-scratch pre-training, all methods draw candidates from the same 3T-token pool constructed from FineWeb (Penedo et al., 2024). To test robustness on a higher-quality corpus, we also run the same recipe on FineWeb-Edu (Penedo et al., 2024). FineWeb-Edu provides a document-level quality classifier that assigns each document a discrete score in {3,4,5}. We partition the FineWeb-Edu pool into two buckets: a 120B-token mid-quality bucket consisting of all score-3 documents, and a 80B-token high-quality bucket formed by merging score-4 and score-5 documents. For static filtering baselines, we score the full pool once and materialize a fixed 30B-token subset for training. For dynamic methods, candidates are streamed from the pool and selected during training. For

*Table 1.* **Evaluation results after training on FineWeb dataset with 30B tokens**. Blocks correspond to model size and optimizer. Bold marks the best compute-matched method per benchmark within each block; a longer-training random-sampling reference at 60B update tokens is included for context. Abbreviations: W.G. = Winogrande; C.QA = CommonsenseQA; WSC = Winograd Schema Challenge. See Table 9 for results on additional settings.

| Method | MMLU | ANLI | HellaSwag | PIQA | SIQA | W.G. | ARC-E | ARC-C | C.QA | WSC | Avg. |
|---|---|---|---|---|---|---|---|---|---|---|---|
| *GPT-2 XL with Muon on 30B update tokens of FineWeb* | | | | | | | | | | | |
| Random | 28.73 | 33.98 | 48.01 | 70.46 | 39.61 | 47.91 | 38.98 | 25.42 | **33.25** | 36.54 | 40.29 |
| PPL | 29.35 | 33.42 | 47.87 | 71.55 | 40.69 | 45.86 | 38.45 | 24.07 | 30.38 | 36.54 | 39.82 |
| GREATS (Wang et al., 2024) | 29.95 | 33.58 | 42.26 | 70.18 | 39.61 | 47.67 | 36.33 | 23.73 | 30.55 | 38.46 | 39.23 |
| QuRating (Wettig et al., 2024) | **33.28** | 33.19 | **48.62** | 70.95 | 41.20 | **48.70** | 37.04 | 26.78 | 30.88 | 36.54 | 40.72 |
| DSIR (Xie et al., 2023) | 29.58 | 33.98 | 48.49 | **71.93** | 39.51 | 47.59 | 38.10 | 26.44 | 32.68 | 38.46 | 40.68 |
| DCLM-FastText (Li et al., 2024) | 30.40 | **34.08** | 44.07 | 71.38 | **41.97** | 48.38 | 38.80 | **29.49** | 30.88 | 36.54 | 40.60 |
| FineWeb-Edu (Penedo et al., 2024) | 29.66 | 33.12 | 48.45 | 71.71 | 41.25 | 46.17 | 39.19 | 28.14 | 31.29 | 38.46 | 40.74 |
| UltraFineweb (Wang et al., 2025c) | 29.95 | 33.31 | 43.11 | 70.57 | 40.79 | 47.51 | 36.51 | 26.44 | 31.70 | 36.54 | 39.64 |
| OPUS (Ours) | 29.89 | 33.29 | 48.39 | 71.27 | 41.10 | 47.99 | **39.68** | 26.44 | 31.37 | **48.08** | **41.75** |
| Random (60B) | 30.24 | 33.84 | 51.10 | 72.25 | 40.89 | 48.78 | 41.98 | 23.05 | 32.35 | 38.46 | 41.29 |
| *GPT-2 Large with AdamW on 30B update tokens of FineWeb* | | | | | | | | | | | |
| Random | 28.19 | 32.91 | 42.65 | 69.37 | **40.79** | 50.12 | 37.21 | 25.08 | 30.06 | 36.54 | 39.29 |
| PPL | 28.69 | 33.44 | 42.23 | 68.77 | 40.43 | 47.36 | 36.68 | 22.37 | 32.84 | 36.54 | 38.94 |
| GREATS (Wang et al., 2024) | 28.77 | 33.46 | 43.00 | 70.46 | 40.63 | 49.96 | 38.45 | 23.39 | 32.02 | 36.54 | 39.67 |
| QuRating (Wettig et al., 2024) | **31.87** | 33.08 | 43.22 | 70.24 | 40.74 | 49.88 | 37.21 | 24.75 | 33.58 | 36.54 | 40.11 |
| DSIR (Xie et al., 2023) | 28.22 | 33.18 | 43.42 | 69.53 | 40.02 | 48.93 | 37.92 | 25.08 | 31.20 | **38.46** | 39.60 |
| DCLM-FastText (Li et al., 2024) | 29.11 | 33.05 | 43.60 | **70.67** | 39.41 | 47.51 | 39.33 | 25.08 | 33.42 | 36.54 | 39.77 |
| FineWeb-Edu (Penedo et al., 2024) | 29.03 | **35.41** | 42.82 | 70.29 | 40.38 | 47.51 | 39.51 | **27.12** | 31.86 | **38.46** | 40.24 |
| UltraFineweb (Wang et al., 2025c) | 29.05 | 33.51 | 43.51 | **70.67** | 40.38 | 48.62 | 41.62 | 25.76 | **34.15** | 36.54 | 40.38 |
| OPUS (Ours) | 31.09 | 34.04 | **45.52** | 69.97 | 40.69 | **51.62** | **42.50** | 26.44 | 33.99 | **38.46** | **41.43** |
| Random (60B) | 29.08 | 33.08 | 44.40 | 70.89 | 41.15 | 48.70 | 37.74 | 22.03 | 32.43 | 36.54 | 39.60 |

*Table 2.* **Evaluation results after training on FineWeb-Edu dataset with 30B tokens**. OPUS is evaluated under a strict constraint: selecting from the mid-quality subset against baselines trained on high-quality data. Additional results are in Table 10.

| Method | MMLU | ANLI | HellaSwag | PIQA | SIQA | W.G. | ARC-E | ARC-C | C.QA | WSC | Avg. |
|---|---|---|---|---|---|---|---|---|---|---|---|
| *GPT-2 XL with Muon on 30B update tokens of FineWeb-Edu* | | | | | | | | | | | |
| Random (Score 3) | 31.92 | 33.56 | 48.39 | 70.13 | 41.10 | 48.86 | 44.86 | 28.47 | 34.23 | 36.54 | 41.81 |
| Random (Score 4+5) | **34.32** | **33.78** | 46.39 | 68.72 | 39.36 | 47.59 | 50.44 | 32.54 | 29.48 | 36.54 | 41.92 |
| PPL (Score 4+5) | 32.60 | 33.58 | 46.14 | 69.10 | 40.33 | 51.70 | 50.79 | 30.17 | 31.78 | 36.54 | 42.27 |
| GREATS (Score 4+5) | 33.58 | 33.02 | 46.32 | 68.93 | 39.61 | **52.57** | 49.21 | 33.90 | 28.01 | 36.54 | 42.17 |
| QuRating (Score 4+5) | 33.10 | 33.58 | 44.22 | 66.70 | 39.97 | 49.64 | 50.09 | 32.54 | 28.99 | 36.54 | 41.54 |
| DSIR (Score 4+5) | 34.13 | 33.63 | 45.10 | 67.79 | 39.82 | 48.15 | 49.03 | 32.88 | 28.83 | 36.54 | 41.59 |
| DCLM-FastText (Score 4+5) | 33.19 | 33.02 | 44.36 | 68.23 | **41.15** | 48.86 | 51.32 | **35.59** | 30.14 | 36.54 | 42.24 |
| FineWeb-Edu (Score 4+5) | 32.94 | 33.64 | 43.14 | 68.28 | 39.61 | 51.30 | **52.73** | 32.20 | 31.37 | 36.54 | 42.18 |
| UltraFineweb (Score 4+5) | 33.41 | 33.48 | 44.34 | 68.93 | 38.64 | 48.30 | 49.38 | 33.56 | 29.07 | 36.54 | 41.57 |
| OPUS (Score 4+5) | 33.83 | 33.64 | 46.30 | 70.67 | 38.95 | 51.14 | 50.62 | 29.15 | 30.47 | 39.42 | 42.42 |
| OPUS (Score 3) | 32.62 | 33.11 | **50.54** | **72.20** | 41.04 | 51.46 | 47.62 | 30.85 | **35.63** | **54.81** | **44.99** |
| Random (60B) (Score 4+5) | 33.77 | 33.54 | 46.94 | 69.64 | 39.82 | 49.80 | 50.44 | 32.54 | 30.96 | 38.46 | 42.59 |

CPT, we construct a 3B-token pool from SciencePedia (SciencePedia Team, 2025) for continued pre-training.

**Optimizers and hyperparameters.** We evaluate two optimizer settings under the same learning-rate schedule and training recipe. In Muon setting, we apply Muon (Jordan et al., 2024)[1] updates to matrix-shaped parameters and use AdamW (Loshchilov & Hutter, 2019) for parameter types where Muon-style matrix preconditioning is not directly applicable, such as biases and normalization parameters. In AdamW setting, we use AdamW (Loshchilov & Hutter, 2019) for all parameters as a unified baseline. Full experimental details are reported in Appendix E.

**Evaluation.** We evaluate pre-trained models on a suite of standard benchmarks spanning knowledge, reasoning, and commonsense: MMLU (Hendrycks et al., 2021), ANLI (Nie et al., 2020), HellaSwag (Zellers et al., 2019), PIQA (Bisk et al., 2020), SIQA (Sap et al., 2019), Winogrande (Sakaguchi et al., 2021), ARC-E/ARC-C (Clark et al., 2018), CommonsenseQA (Talmor et al., 2019), and WSC (Levesque et al., 2012). In the continued pre-training setting, we evaluate checkpoints on the science-focused benchmarks SciAssess (Cai et al., 2025) and OlympicArena (Huang et al., 2024) and report the official accuracy metric. All benchmarks are evaluated at zero-shot settings. We additionally report generalization on the following held-out benchmarks: BBH (Suzgun et al., 2023), Race (Lai et al., 2017), AX-B/AX-G (Wang et al., 2019), and StoryCloze (Mostafazadeh et al., 2016), that is not used in the

---

[1]The optimizer employed in our implementation combines Muon and AdamW. To simplify notation, we use "Muon" as shorthand for this hybrid optimizer in the remainder of this paper.

*Table 3.* **Domain-specific perplexity analysis.** Perplexity (PPL) on ten distinct domains for GPT-2 XL after 30B update tokens. Following (Wettig et al., 2025), we evaluate on 1K held-out samples per domain. *Lower PPL indicates better compression performance.*

| Method | Health | Business | Politics | Education | History | Lifestyle | Science | Arts | Entertainment | Computing | Avg. |
|---|---|---|---|---|---|---|---|---|---|---|---|
| | *GPT-2 XL with Muon optimizer on 30B update tokens of FineWeb-Edu Subset (score $\geq$ 3)* | | | | | | | | | | |
| Random (30B) | 3.25 | 3.51 | 3.55 | 3.48 | 3.45 | 3.79 | 3.42 | 3.73 | 4.00 | 3.83 | 3.60 |
| DSIR | 3.24 | 3.50 | 3.54 | 3.47 | 3.44 | 3.78 | 3.41 | 3.72 | 4.00 | 3.81 | 3.59 |
| DCLM-FastText | 3.36 | 3.64 | 3.70 | 3.62 | 3.61 | 3.94 | 3.52 | 3.86 | 4.13 | 3.94 | 3.73 |
| FineWeb-Edu | 3.29 | 3.55 | 3.58 | 3.50 | 3.49 | 3.82 | 3.45 | 3.75 | 4.02 | 3.83 | 3.63 |
| QuRating | 3.50 | 3.79 | 3.93 | 3.70 | 3.83 | 4.18 | 3.73 | 4.04 | 4.39 | 4.24 | 3.93 |
| UltraFineweb | 3.43 | 3.74 | 3.90 | 3.68 | 3.80 | 4.07 | 3.59 | 3.99 | 4.28 | 4.02 | 3.85 |
| PPL | 3.22 | 3.47 | 3.50 | 3.44 | 3.40 | 3.74 | 3.39 | 3.69 | 3.96 | 3.77 | 3.56 |
| GREATS | 3.29 | 3.55 | 3.60 | 3.52 | 3.49 | 3.84 | 3.45 | 3.77 | 4.05 | 3.88 | 3.64 |
| **OPUS (Ours)** | **3.11** | **3.31** | **3.37** | **3.34** | **3.31** | **3.59** | **3.33** | **3.58** | **3.83** | **3.69** | **3.45** |

construction of bench proxy to check that our proxy-guided selection does not overfit to a certain benchmarks.

**Baselines.** We compare OPUS against representative data selection methods. (1) Static baselines. We evaluate five representative static filtering methods: QuRating (Wettig et al., 2024), DSIR (Xie et al., 2023), DCLM-FastText (Li et al., 2024), FineWeb-Edu Classifier (Penedo et al., 2024), and UltraFineweb Classifier (Wang et al., 2025c). (2) Dynamic selection. We include HIGH-PPL (PPL), which selects the highest-loss sequences under the current model following (Ankner et al., 2025), and GREATS (Wang et al., 2024), which selects samples whose per-sample gradients best align with a SGD-based proxy direction in post-training. We also report results of random selection at 30B and 60B update tokens for baseline comparison.

### 4.2. Pre-Training from Scratch

**Performance on web-scale corpora: FineWeb.** We first evaluate OPUS on FineWeb, a standard large-scale web corpus. Table 1 compares OPUS against prior static and dynamic baselines under a fixed budget of 30B update tokens. Across model scales and optimizer settings, OPUS achieves the best compute-matched average and consistently improves over strong baselines. We also include a longer-training random-sampling reference at 60B update tokens to contextualize the magnitude of these efficiency gains; notably, OPUS often matches or exceeds the performance of baselines trained for twice as long.

**Robustness on curated corpora: FineWeb-Edu.** We next evaluate performance on FineWeb-Edu. To test the limits of our method, we subject OPUS to a strict evaluation regime: it selects dynamically from the lower-quality subset (FineWeb-Edu score 3), whereas baselines are trained on the superior high-quality partition (scores 4 and 5). As shown in Table 2, despite this disadvantage in raw data quality, OPUS matches or exceeds prior methods trained on the superior data. For GPT-2 XL with Muon, OPUS achieves the best compute-matched average of 44.99, outperforming all baselines trained on the higher-quality data partitions. We also report our validation-loss trajectories in Figure 6.

Building on this, we further provide a data-filter perspective. Under heavy data contamination, OPUS not only maintains training stability but its utility score can also be applied as a static filter to flag and remove toxic documents from the training corpus prior to training. See Figure 5 and the accompanying analysis in Appendix H.

**Optimizer-induced selection matters: strong gains under AdamW and Muon.** Under AdamW, which utilizes diagonal preconditioning, OPUS achieves the best compute-matched performance for both GPT-2 Large and GPT-2 XL (Table 1). Crucially, this advantage extends to Muon, which employs non-linear matrix preconditioning via Newton-Schulz orthogonalization. For instance, on GPT-2 XL with Muon on FineWeb, OPUS outperforms Random selection by a significant margin ($40.29 \rightarrow 41.75$). This empirically validates our central hypothesis: aligning data selection with the preconditioned update trajectory yields a more effective training signal than raw gradient-based selection. The faithfulness of this preconditioned scoring is further supported by our approximation-fidelity study, which shows that the errors introduced by the curvature simplification, the ghost factorization, the first-order Taylor surrogate, and the optimizer-induced preconditioner all remain small enough to preserve the candidate ranking that drives selection. See Table 11 in Appendix H.

**Generalization beyond proxy-aligned benchmarks.** Since OPUS uses a benchmark-matched proxy direction to guide training-time selection, it is important to verify that gains are not merely driven by overfitting to the specific evaluation suite used to construct the proxy. We therefore evaluate on a set of *out-of-distribution* benchmarks covering challenging reasoning and general language comprehension for generalization evaluation. As shown in Table 4, OPUS achieves the best performance, suggesting that it reflects more general training signal quality, rather than narrow specialization to the proxy-aligned benchmark.

**OPUS enhances compression intelligence measured by domain perplexity across domains.** To ensure that our selection strategy does not overfit to specific patterns at the expense of broad coverage, we evaluate domain-wise perplexity (PPL). Following the evaluation protocol of WE-

*Table 4.* **Evaluation on out-of-distribution benchmarks.** We evaluate the same GPT2-XL checkpoints from Table 1 on out-of-distribution benchmarks that are not included in BENCH-PROXY.

| Method | BBH | RACE-M | RACE-H | AX-b | AX-g | StoryCloze | Avg. |
|---|---|---|---|---|---|---|---|
| Random | 9.87 | 24.58 | 25.19 | 52.54 | 50.00 | 66.38 | 38.09 |
| PPL | 9.88 | 24.37 | 25.73 | 54.98 | 51.12 | 67.34 | 38.90 |
| GREATS | 10.44 | 26.04 | 26.04 | 57.34 | 50.84 | 65.79 | 39.42 |
| QuRating | 10.65 | 24.79 | 23.33 | 54.35 | **51.97** | 66.70 | 38.63 |
| DSIR | 9.92 | 25.07 | 26.21 | 53.53 | 49.44 | **67.72** | 38.65 |
| DCLM-FastText | 10.65 | 26.53 | 25.59 | 52.08 | **51.97** | 66.86 | 38.95 |
| FineWeb-Edu | 9.73 | **26.81** | 25.90 | 55.25 | 50.00 | 66.76 | 39.08 |
| UltraFineweb | 9.69 | 23.26 | 22.58 | 48.73 | 48.31 | 67.13 | 36.62 |
| OPUS (Ours) | **11.02** | 25.77 | **27.50** | **58.42** | 50.56 | 67.13 | **40.07** |

BORGANIZER (Wettig et al., 2025), we first label documents using the WebOrganizer topic classifier to classify documents into 24 topics and merge these semantically similar topics into ten domains. We then construct a held-out test set by randomly sampling 1,000 documents from each of ten distinct domains (e.g., Health, Law, Science) to ensure a balanced evaluation. Table 3 indicates that OPUS achieves the lowest average perplexity on FineWeb-Edu dataset.

## 4.3. Continued Pre-Training

We extend our evaluation to continued pre-training (CPT), a critical setting for adapting general-purpose LLMs to specialized verticals. We continue training Qwen3-8B-Base on SciencePedia. Figure 3 reports the average downstream performance on the specialized SciAssess benchmark and the reasoning-heavy OlympicArena versus CPT tokens. Notably, OPUS reaches the best performance using only 0.5B tokens and already outperforms random CPT trained for 3B tokens, implying a 6× gain in data efficiency. Furthermore, the improvement on both benchmarks suggests that OPUS enhances domain-specific knowledge without catastrophic forgetting of general reasoning capabilities (see Figure 7).

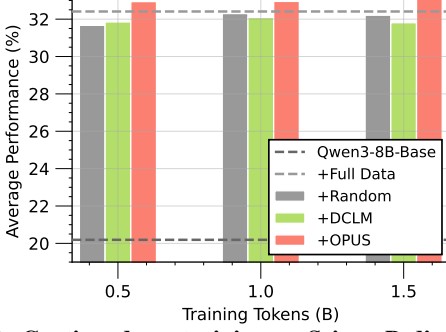

*Figure 3.* **Continued pre-training on SciencePedia.** Average performance versus CPT tokens for random CPT, DCLM, and OPUS. Detailed domain-specific results are in Figure 7.

## 4.4. Efficiency Analysis

A key advantage of OPUS is its minimal computational overhead. Static filtering methods incur a substantial one-time cost to score the entire corpus, while dynamic selection adds per-iteration scoring during training. As shown in Figure 4,

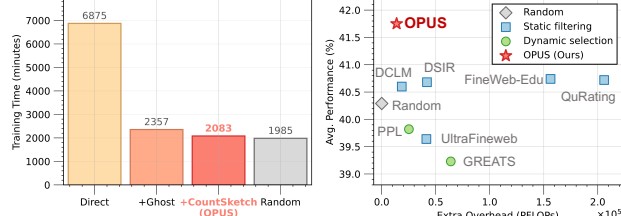

*Figure 4.* **Efficiency and computational cost analysis.** Time (minutes) and total compute (PFLOPs) are evaluated on GPT-2 XL after pre-training on FineWeb (30B tokens) with Muon.

a naïve direct implementation of online selection would incur *over 3.5× slowdown* compared to random sampling. By incorporating ghost gradients and CountSketch projections, OPUS reduces this overhead to *only 4.7%* while achieving the best benchmark performance. In contrast, static methods like QuRating require an order of magnitude more compute for selection yet fail to outperform OPUS.

## 4.5. Ablation Study

**Soft sampling vs. greedy top-$k$.** We replace Boltzmann soft sampling with a deterministic greedy variant that always selects the top-$K$ candidates by utility. Table 5 shows that greedy selection improves over Random, but remains notably behind full OPUS: the greedy variant reaches an Avg. of 40.49, whereas OPUS achieves 41.75. This supports our motivation that purely greedy top-$k$ selection can over-concentrate on a narrow set of high-score but overlapping candidates, while stochastic sampling better preserves update diversity and stabilizes training.

*Table 5.* Ablation study on sampling and validation strategy.

| Benchmark | Random | OPUS Variants | | |
| | | Greedy | Std. proxy | OPUS |
|---|---|---|---|---|
| MMLU | 28.73 | 29.63 | 29.50 | **29.89** |
| ANLI | 33.98 | 33.52 | 33.70 | **33.29** |
| HellaSwag | 48.01 | 48.17 | 48.18 | **48.39** |
| PIQA | 70.46 | **72.25** | 71.60 | 71.27 |
| SIQA | 39.61 | **41.61** | 40.28 | 41.10 |
| Winogrande | 47.91 | 49.88 | **51.85** | 47.99 |
| ARC-E | 38.98 | 37.39 | 38.80 | **39.68** |
| ARC-C | 25.42 | 24.75 | 26.10 | **26.44** |
| C.QA | 33.25 | 31.12 | **32.76** | 31.37 |
| WSC | 36.54 | 36.54 | 37.50 | **48.08** |
| Average | 40.29 | 40.49 | 41.03 | **41.75** |

**Benchmark-matched proxy vs. standard proxy.** OPUS estimates the target update direction using a small proxy pool. We compare the default proxy construction with a benchmark-matched proxy that is retrieved to better reflect the downstream evaluation distribution (Sec. 3). As shown in Table 5, the benchmark-matched proxy yields a measurable improvement over the default setting, increasing the average from 41.03 to 41.75. This indicates that sharpening the proxy direction can further increase the effectiveness of utility-based selection. Table 5 also shows that the standard proxy already provides strong gains over RANDOM, improving the average from 40.29 to 41.03.

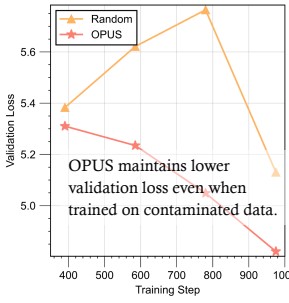

*Figure 5.* **Validation loss under heavy data contamination.** We conducted continued pre-traning on Qwen3-0.6B-Base on a FineWeb-Edu pool in which 80% of documents are corrupted. OPUS maintains substantially lower validation loss than baseline even when trained on contaminated data.

### 4.6. OPUS as an Online and Post-hoc Data Filter

**OPUS's benefits persist when the candidate pool is dominated by harmful content.** A practical use case of optimizer-aware utility is identifying corrupted samples that would otherwise harm pre-training. We design a small-scale exploration on the Qwen3-0.6B-Base checkpoint, where we run continued pre-training on a heavily contaminated FineWeb-Edu pool. In this pool, 80% *of the documents have their tokens randomly permuted within each document*, which preserves the marginal token distribution but destroys coherence. The remaining 20% are kept intact. We train for 1,000 update steps with disabled gradient clipping so that any selection of toxic samples is immediately reflected in the validation loss. OPUS itself is run with its standard hyperparameter configuration except that we tighten the selection ratio to $\rho=0.25$. The OPUS proxy is a small held-out shard of clean FineWeb-Edu, and the validation set is drawn from the same clean distribution. As shown in Figure 5, the contamination drives the Random baseline's validation loss sharply upward as training proceeds, before the model partially recovers. OPUS exhibits a clearly different trajectory. Its validation loss peaks both earlier and lower, and from that point decreases steadily.

**OPUS scores can flag real toxic data.** A natural follow-up question is *whether the same utility score can also identify real-world toxic content*, without any training. We move from synthetic shuffles to two openly available corpora of genuinely low-quality online text. The first is the labeled toxic split of the Jigsaw Toxic Comment Classification corpus (cjadams et al., 2017) and the second is the spam split of the Enron email corpus (Metsis et al., 2006). To match the scale at which OPUS operates during training, we run a single scoring pass on an OPUS buffer of size $N=32$ with the selection ratio set to $\rho=0.25$. The buffer is balanced between clean FineWeb-Edu documents and toxic documents drawn from the two corpora above. The proxy is a small held-out shard of clean FineWeb-Edu, and the scoring model is the unmodified Qwen3-0.6B-Base checkpoint. OPUS then assigns a utility score to each of the 32 candidates and splits them into a *selected* top quartile of 8 samples and a *not-selected* group of 24 samples.

We present the 8 highest-scoring candidates within the selected group and the 8 lowest-scoring candidates within the rejected group. The high-scoring selected samples are all well-formed educational prose covering history, science, linguistics, and ecology, while the low-scoring not-selected samples are unambiguous spam patterns such as pharmacy ads, Nigerian-prince scams, knock-off product offers, and auto-generated mail-server bounces. This shows that the optimizer-aware utility carries a real direction signal that aligns with human-perceived data quality for evaluation.

**OPUS-Selected: clean candidates**      *mean score = +5.293*

| S1 Selected +6.274 | S2 Selected +5.153 |
|---|---|
| JOHN BAKER UNRAVELS HISTORY Genealogy Expert John F. Baker Jr., has written the most accessible and exciting work of African American... | Ayurveda traces its origins to the Rig Veda, the world's oldest surviving book in an Indo-European language. The Rig Veda, 3000 B.C., is a c... |
| **S3** Selected +5.153 | **S4** Selected +5.153 |
| The R/V David Folger's route home takes it north along the Hudson River and then through the Champlain Canal to Lake Champlain. On July 30, ... | Within a library collection, materials are typically organized by subject. Librarians assign a call number based on a work's subject and sou... |
| **S5** Selected +5.153 | **S6** Selected +5.153 |
| NY1's "Making Census Of It" series continues this week with a spotlight on the Bronx, the borough that experienced the largest growth in the... | Developed in the 1920s by the legendary physical trainer, Joseph H. Pilates, "The Pilates Method" is an exercise system focused on improving... |
| **S7** Selected +5.153 | **S8** Selected +5.153 |
| The State of Food Insecurity in the World 2011 highlights the differential impacts that the world food crisis of 2006-08 had on different co... | In diachronic comparison of languages, say PIE to Latin to Romance, it is a classic recognition that the later languages strictly lose some ... |

**OPUS-Rejected: toxic candidates**      *mean score = +0.548*

| S1 Rejected −0.025 | S2 Rejected +0.037 |
|---|---|
| greece based professionals and organizations greece based professionals and organizations you areinvited to register on training consortiumj... | confidence confidence attn : manager hello dear , i am well confidence of your capability to assist me in a transaction for mutual benefit f... |
| **S3** Rejected +0.074 | **S4** Rejected +0.310 |
| fantastic investors portfoiio fantastic investors portfoiio shirley , investor alert - lrcj - brand new stock for your attention lauraan cor... | failure notice failure notice hi . this is the qmail - send program at harrisburg . villagetech . com . i ' m afraid i wasn ' t able to deli... |
| **S5** Rejected +0.438 | **S6** Rejected +0.571 |
| preferred non - smoker rates for smokers preferred non - smoker rates for smokers case study # 1 male - 63 $ 5 , 000 , 000 face good health ... | reminder reminder psychologists bottled overflowing deeply bloomfield singers hepburn odder hopkinsian reassembles stupidity coolies picasso... |
| **S7** Rejected +0.601 | **S8** Rejected +2.375 |
| impress your girl with a huge c*****t ! impress your girl with a huge c*****t ! heya ! has your c*m ever dribbled and you wish it had shot o... | business assistance business assistance dear sir . first , i must solicit your confidence in this transaction , this is by virtue of its na... |

## 5. Conclusion and Future Direction

We introduced OPUS, a dynamic data selection framework for LLM pre-training that aligns training-time selection with the optimizer's effective update geometry. Across model scales, optimizers, and corpus quality settings, OPUS consistently improves compute-matched pre-training, suggesting that selection can be substantially strengthened by accounting for how the optimizer actually moves parameters. A natural next step is to extend this optimizer-aligned idea to richer training regimes, such as data mixtures.

## Acknowledgements

We thank Jiachen T. Wang at Princeton University and Meng Ding at University at Buffalo for helpful feedback and discussions.

## Impact Statement

This work aims to make training-time data selection for LLM pre-training more principled and scalable. OPUS defines utility in the optimizer-induced update space and uses lightweight projection with stochastic sampling to keep overhead low while avoiding overly narrow or redundant selections. By improving compute efficiency in both from-scratch pre-training and continued pre-training, the approach can reduce training cost and energy usage and make careful data selection easier to reproduce in practice.

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

---

**Algorithm 1** OPUS: Optimizer-induced Projected Utility Selection

---

1: **Input:** Model $f_\theta$; Data stream $\mathcal{D}_{\text{tr}}$; Proxy pool $\mathcal{D}_{\text{proxy}}$; Optimizer $\mathcal{O}$ (AdamW or Muon); Selection ratio $\rho$; Projection dim $m$.

2: **Initialize:** Implicit sketch operator $\Pi$ using CountSketch with hash $h : [d] \to [m]$ and sign $s : [d] \to \{-1, +1\}$.

3: **for** $t = 0, 1, \dots$ **do**

4:     **1. Batch Sampling:** Read candidate buffer $\mathcal{B}_t = \{z_1, \dots, z_N\}$ from $\mathcal{D}_{\text{tr}}$.

5:     **2. Preconditioner Computation:** Construct optimizer-induced preconditioner $\mathbf{P}_t = \mathbf{P}(\mathcal{O}_t)$ from $\mathcal{O}$'s state at step $t$.

6:     **3. Proxy Feature Generation:** Sample $K_{\text{proxy}}$ samples $\{\tilde{z}_k\}$ from $\mathcal{D}_{\text{proxy}}$, obtain ghost factors $\{\mathbf{a}_r^{(\tilde{z}_k)}, \mathbf{b}_r^{(\tilde{z}_k)}\}$, and compute per-layer proxy sketches $\psi_{\text{proxy}}^{(t,r)} \leftarrow \Pi_r \left( \frac{1}{K_{\text{proxy}}} \sum_{k=1}^{K_{\text{proxy}}} \mathbf{a}_r^{(\tilde{z}_k)} \otimes \mathbf{b}_r^{(\tilde{z}_k)} \right)$ for all $r \in \mathcal{R}$.

7:     **4. Candidate Feature Generation:** Compute per-layer sketches $\phi^{(t,r)}(z) \in \mathbb{R}^m$ implicitly from ghost factors $\{\mathbf{a}_r^{(z)}, \mathbf{b}_r^{(z)}\}_{r \in \mathcal{R}}$:
$$\phi^{(t,r)}(z) \leftarrow \Pi_r \left( \mathbf{P}_{t,r} \left( \mathbf{a}_r^{(z)} \otimes \mathbf{b}_r^{(z)} \right) \right), \quad \forall r \in \mathcal{R}.$$

8:     **5. Soft Sampling Loop:**

9:     Let target batch size $K = \lfloor \rho N \rfloor$, Selected set $\widehat{\mathcal{B}}_t \leftarrow \emptyset$, and per-layer history $\Phi^{(t,r)} \leftarrow \mathbf{0}$ for all $r \in \mathcal{R}$.

10:     **for** $j = 1$ to $K$ **do**

11:         For each $z \in \mathcal{B}_t \setminus \widehat{\mathcal{B}}_t$, compute $U_z^{(t)}$:
$$U_z^{(t)} \leftarrow \eta_t \sum_{r \in \mathcal{R}} \langle \phi^{(t,r)}(z), \psi_{\text{proxy}}^{(t,r)} \rangle - \eta_t^2 \sum_{r \in \mathcal{R}} \langle \phi^{(t,r)}(z), \Phi^{(t,r)} \rangle$$

12:         Sample index $z^*$ via Softmax: $p_t(z^*) \propto \exp(U_z^{(t)}/\tau)$.

13:         Add to batch: $\widehat{\mathcal{B}}_t \leftarrow \widehat{\mathcal{B}}_t \cup \{z^*\}$.

14:         Update history (redundancy): $\Phi^{(t,r)} \leftarrow \Phi^{(t,r)} + \phi^{(t,r)}(z^*)$ for all $r \in \mathcal{R}$.

15:     **end for**

16:     **6. Update:** Train $\theta_{t+1}$ using batch $\widehat{\mathcal{B}}_t$ with optimizer $\mathcal{O}$.

17: **end for**

---

## A. Pseudo Code for OPUS Algorithm

Algorithm 1 summarizes OPUS, a step-wise dynamic selection method that scores candidates in the *optimizer-induced update space*. At each step $t$, OPUS samples a candidate buffer $\mathcal{B}_t$, constructs the preconditioner $\mathbf{P}_t$ from the optimizer state, and builds a proxy *target direction* from an in-distribution pool $\mathcal{D}_{\text{proxy}}$ via ghost factors, yielding per-layer proxy sketches $\psi_{\text{proxy}}^{(t,r)}$ for $r \in \mathcal{R}$. For each candidate $z \in \mathcal{B}_t$, it forms a sketch feature $\phi^{(t,r)}(z)$ by applying $\mathbf{P}_{t,r}$ to the ghost outer-product gradient and projecting with CountSketch $\Pi_r$ into $\mathbb{R}^m$ for efficiency. OPUS then selects $K = \lfloor \rho N \rfloor$ samples using Boltzmann sampling with a marginal-gain objective that balances proxy alignment and redundancy control, and updates the model on the selected subset $\widehat{\mathcal{B}}_t$.

## B. Related Work

**Static pre-training data selection.** Most large-scale LLM pre-training pipelines rely on static corpus filtering, where documents are filtered or reweighted once before training. Representative approaches include classifier- or rule-based filtering over web corpora, exemplified by FineWeb and its educational subset FineWeb-Edu (Penedo et al., 2024), which document large-scale deduplication and quality filtering choices for Common Crawl derived data. Recent work has also studied more targeted quality signals: QuRating (Wettig et al., 2024) learns scalar quality ratings from pairwise preferences and shows that balancing quality and diversity improves downstream performance, while DSIR (Xie et al., 2023) formalizes dataset matching via importance resampling in a reduced feature space, enabling scalable selection without human curation. Complementary benchmark and pipeline efforts such as DataComp-LM (DCLM) (Li et al., 2024) provide standardized corpora and evaluation suites to compare filtering strategies, and UltraFineweb (Wang et al., 2025c) proposes efficient filtering and verification mechanisms (including lightweight classifier-based pipelines) to further improve web-scale data quality. While effective at removing low-quality noise, these static approaches are inherently training-agnostic: they assume sample utility is time-invariant and do not adapt to the model's evolving needs across optimization.

**Dynamic data selection during pre-training.** To move beyond fixed corpora, dynamic selection chooses samples on-the-fly based on an estimated training utility. Early and widely-used heuristics prioritize samples with large loss or high perplexity, and several works formalize this intuition via online batch selection and importance sampling (Loshchilov & Hutter, 2016; Katharopoulos & Fleuret, 2018). A more rigorous approach uses influence functions (IF) to estimate the impact of training points on validation loss (Koh & Liang, 2017). While classic IF methods are computationally intensive and require Hessian inversion, recent approximations have made them more feasible for deep learning. In LLM pre-training, GREATS proposes a principled objective by approximating per-sample validation loss reduction via a Taylor expansion, and then selects a subset each step, typically greedily. It can incur substantial scoring overhead due to per-sample gradient and influence approximations (Wang et al., 2024). More recently, MATES (Yu et al., 2024) learns a lightweight influence model to track evolving data preferences during pre-training, and Group-MATES (Yu et al., 2026) emphasizes that utility is not additive and that group-level interactions matter, mitigating redundancy induced by greedy top-$k$ selection. In parallel, perplexity-based pruning remains a competitive, simple signal for data selection and pruning, including settings where a small reference model computes PPL to prune large-scale corpora (Ankner et al., 2025). OPUS fits this dynamic-selection family, but differs by aligning utility with the optimizer-induced update and by using efficient projected scoring with soft sampling.

**Influence-function scores and data-attribution.** A large line of work studies training-data influence and attribution (Hammoudeh & Lowd, 2024; Deng et al., 2025)—estimating how individual samples affect model behavior or validation loss. Classical influence functions approximate the effect of upweighting a training point via Hessian-based sensitivity analysis, enabling fine-grained data attribution without retraining (Koh & Liang, 2017). To make influence estimation practical in deep, non-convex settings, some works replace exact second-order IF computation with scalable surrogates (Pruthi et al., 2020; Guo et al., 2021; Yeh et al., 2018). Related directions also develop first-order or early-training proxies for data importance, such as selecting informative subsets early in training (Paul et al., 2021), leveraging forgetting events to identify noisy or hard-to-learn samples (Toneva et al., 2019), and optimizing subset selection via gradient-matching (Killamsetty et al., 2021) or influence functions (Hu et al., 2024). Another line of research explores Shapley value, a concept from cooperative game theory, to quantify the value of data (Ghorbani & Zou, 2019; Jia et al., 2021; Wang et al., 2025a). Recently, influence and data-attribution signals have been adapted from classical IF literature to practical data selection for large language models, including LoRA-aware influence approximations and gradient-datastore based retrieval (Xia et al., 2024), as well as more structured selection pipelines that optimize selection objectives for instruction tuning (Du et al., 2023; Liu et al., 2024b). Moreover, many approaches implicitly operate in raw-gradient geometry and/or employ deterministic top-$k$ retrieval, which can become brittle under rapidly changing training dynamics and optimizer-induced transformations. These limitations motivate online selection objectives that remain faithful to the effective optimizer update while preserving scalability and diversity.

# C. Optimizer-Induced Preconditioners

## C.1. Stochastic Gradient Descent

We include SGD as a minimal reference point, since many prior dynamic selection methods implicitly assume an SGD-like update geometry and score candidates directly using raw gradients. In SGD, the optimizer applies a uniform scalar learning rate (and optional weight decay) without stateful preconditioning, so the effective update direction is aligned with the mini-batch gradient. Consequently, at a fixed step $t$, SGD induces an (approximately) identity update geometry, $\mathbf{P}_t \approx \mathbf{I}$, making raw-gradient similarity a natural scoring signal.

---

**SGD**

Stochastic gradient descent updates parameters by moving along the negative mini-batch gradient:

$$\mathbf{g}_t = \nabla_\theta \mathcal{L}(\mathcal{B}_t; \theta_t), \qquad \Delta\theta_t = -\eta_t \mathbf{g}_t.$$

With optional weight decay, the one-step update becomes

$$\Delta\theta_t = -\eta_t(\mathbf{g}_t + \lambda\theta_t).$$

For online scoring at a fixed step $t$, SGD induces an identity update geometry $\mathbf{P}_t \approx \mathbf{I}$, so utility is naturally measured in raw-gradient space.

---

## C.2. Muon Preconditioner

We derive the Muon-instantiated preconditioner by linearizing Muon's one-step *lookahead* update at a fixed training step $t$ (the regime used for online selection). Consider a linear weight matrix $W_{\mathcal{L}} \in \mathbb{R}^{o \times i}$ updated by Muon. Ignoring bias-corrections for exposition, Muon maintains an EMA momentum on the (mini-batch) gradient $\mathbf{g}_{t,\mathcal{L}}(S) := \frac{1}{|S|} \sum_{z \in S} \nabla_{W_{\mathcal{L}}} \mathcal{L}(z; \theta_t)$:

$$\mathbf{m}_{t+1,\mathcal{L}}(S) = \mu \mathbf{m}_{t,\mathcal{L}} + (1 - \mu) \mathbf{g}_{t,\mathcal{L}}(S). \tag{17}$$

In practice, Muon forms a "double-smoothed" direction fed to the orthogonalizer,

$$\mathbf{q}_{t+1,\mathcal{L}}(S) := (1 - \mu) \mathbf{g}_{t,\mathcal{L}}(S) + \mu \mathbf{m}_{t+1,\mathcal{L}}(S) = \mu^2 \mathbf{m}_{t,\mathcal{L}}(S) + (1 - \mu^2) \mathbf{g}_{t,\mathcal{L}}(S). \tag{18}$$

and takes the parameter step

$$\Delta W_{t,\mathcal{L}}(S) := W_{t+1,\mathcal{L}}(S) - W_{t,\mathcal{L}} = -\eta_t \mathcal{O}_{t,\mathcal{L}}\big(\mathbf{q}_{t+1,\mathcal{L}}(S)\big). \tag{19}$$

**Online-selection view.** For scoring at fixed step $t$, we hold Muon's state fixed (learning rate $\eta_t$, momentum coefficient $\mu$, and the history buffer $\mathbf{m}_{t,\mathcal{L}}$). Moreover, we *freeze* the Newton–Schulz (NS) operator during selection by constructing it from a reference direction $\bar{\mathbf{q}}_{t,\mathcal{L}}$ available at the start of step $t$ (e.g., from the current optimizer buffer / proxy batch), and reuse it for all candidates. Under this approximation, NS induces an approximately linear left-multiplication map

$$\mathcal{O}_{t,\mathcal{L}}(Z) \approx \mathbf{S}_{t,\mathcal{L}} Z, \ \mathbf{S}_{t,\mathcal{L}} = a\mathbf{I} + b\mathbf{A}_{t,\mathcal{L}} + c\mathbf{A}_{t,\mathcal{L}}^2, \ \mathbf{A}_{t,\mathcal{L}} := \tilde{\bar{\mathbf{q}}}_{t,\mathcal{L}} \tilde{\bar{\mathbf{q}}}_{t,\mathcal{L}}^\top. \tag{20}$$

where $\tilde{\bar{\mathbf{q}}}_{t,\mathcal{L}} := \bar{\mathbf{q}}_{t,\mathcal{L}} / \|\bar{\mathbf{q}}_{t,\mathcal{L}}\|_F$ (and $a, b, c$ are fixed NS polynomial coefficients). Substituting (18) into (19) and using (20) yields the linearized lookahead update

$$\Delta W_{t,\mathcal{L}}(S) \approx \mathbf{b}_{t,\mathcal{L}} - \kappa_t \mathbf{S}_{t,\mathcal{L}} \mathbf{g}_{t,\mathcal{L}}(S), \ \mathbf{b}_{t,\mathcal{L}} := -\eta_t \mu^2 \mathbf{S}_{t,\mathcal{L}} \mathbf{m}_{t,\mathcal{L}}, \ \kappa_t := \eta_t (1 - \mu^2). \tag{21}$$

Since OPUS ranks candidates/subsets by *relative* utility at fixed $t$, the $S$-independent shift can be dropped for scoring purposes, and the effective data-dependent update is captured by a layerwise preconditioner

$$\Delta W_{t,\mathcal{L}}(S) \approx -\mathbf{P}_{t,\mathcal{L}}^{\text{Muon}} \mathbf{g}_{t,\mathcal{L}}(S) + \text{const}, \qquad \mathbf{P}_{t,\mathcal{L}}^{\text{Muon}} := \kappa_t \mathbf{S}_{t,\mathcal{L}}. \tag{22}$$

Thus, Muon induces a *dense, sample-independent* (at fixed $t$ under frozen $\mathbf{S}_{t,\mathcal{L}}$) left-preconditioner that reshapes gradient directions before scoring; OPUS remains optimizer-agnostic by plugging $\mathbf{P}_{t,\mathcal{L}}^{\text{Muon}}$ into the same utility machinery used for AdamW.

---

**MUON**

Muon targets matrix-shaped parameters $W \in \mathbb{R}^{o \times i}$ by maintaining an accumulated matrix direction and applying a Newton–Schulz orthogonalization (matrix-sign style) transform:

$$\mathbf{M}_t = \mu \mathbf{M}_{t-1} + (1 - \mu) \mathbf{g}_t,$$
$$\mathbf{Q}_t := \text{NewtonSchulz}(\mathbf{M}_t),$$
$$\Delta W_t \propto -\mathbf{Q}_t.$$

For online selection at fixed step $t$, we hold the optimizer state and freeze the Newton–Schulz operator across candidates, yielding an approximately linear map $\text{NewtonSchulz}(Z) \approx \mathbf{S}_t Z$. This induces a dense, layerwise preconditioner $\mathbf{P}_t$ that reshapes update geometry beyond raw-gradient space.

## C.3. AdamW Preconditioner

We derive the AdamW-instantiated preconditioner by linearizing the one-step *lookahead* update that OPUS uses to score candidate subsets. Consider the (decoupled) AdamW update applied to a subset $S$ at iteration $t$:

$$\mathbf{m}_t(S) = \beta_1 \mathbf{m}_{t-1} + (1 - \beta_1)\mathbf{g}_t(S), \qquad \mathbf{v}_t(S) = \beta_2 \mathbf{v}_{t-1} + (1 - \beta_2)\mathbf{g}_t(S)^{\odot 2}, \tag{23}$$

$$\widehat{\mathbf{m}}_t(S) = \frac{\mathbf{m}_t(S)}{1 - \beta_1^t}, \qquad \widehat{\mathbf{v}}_t(S) = \frac{\mathbf{v}_t(S)}{1 - \beta_2^t}, \qquad \theta_{t+1}(S) = \theta_t - \alpha_t \frac{\widehat{\mathbf{m}}_t(S)}{\sqrt{\widehat{\mathbf{v}}_t(S)} + \epsilon} - \alpha_t \lambda \theta_t. \tag{24}$$

where $\mathbf{g}_t(S) := \frac{1}{|S|} \sum_{z \in S} \nabla_\theta \mathcal{L}(z; \theta_t)$ and $\odot$ denotes elementwise operations.

**Online-selection view.** At a fixed training step $t$, OPUS compares subsets $S$ via their *relative* utility under a one-step lookahead while *holding the optimizer state fixed at the start of step $t$*. Concretely, we treat $\alpha_t, \beta_1, \beta_2, \epsilon, \lambda$ and the history buffers $(\mathbf{m}_{t-1}, \mathbf{v}_{t-1})$ as constants with respect to $S$.

**Affine dependence on the batch gradient.** Under this view, the bias-corrected first moment is affine in $\mathbf{g}_t(S)$:

$$\widehat{\mathbf{m}}_t(S) = \frac{\beta_1}{1 - \beta_1^t}\mathbf{m}_{t-1} + \frac{1 - \beta_1}{1 - \beta_1^t}\mathbf{g}_t(S). \tag{25}$$

**Frozen preconditioner approximation.** To keep scoring tractable, we freeze the RMS geometry during selection by dropping the $S$-dependence in the second moment update. Using $\widehat{\mathbf{v}}_t(S) = \mathbf{v}_t(S)/(1 - \beta_2^t)$ with $\mathbf{v}_t(S) = \beta_2 \mathbf{v}_{t-1} + (1 - \beta_2)\mathbf{g}_t(S)^{\odot 2}$, we approximate

$$\sqrt{\widehat{\mathbf{v}}_t(S)} + \epsilon = \sqrt{\frac{\beta_2 \mathbf{v}_{t-1} + (1 - \beta_2)\mathbf{g}_t(S)^{\odot 2}}{1 - \beta_2^t}} + \epsilon \approx \sqrt{\overline{\mathbf{v}}_t} + \epsilon, \qquad \overline{\mathbf{v}}_t := \frac{\beta_2 \mathbf{v}_{t-1}}{1 - \beta_2^t}. \tag{26}$$

Substituting (25) and (26) into (24) yields the linearized form. Let $\mathbf{D}_t := \text{Diag}\left(\frac{1}{\sqrt{\overline{\mathbf{v}}_{t-1}} + \epsilon}\right)$, $A_t := \alpha_t \frac{\beta_1}{1 - \beta_1^t}$, and $C_t := \alpha_t \frac{1 - \beta_1}{1 - \beta_1^t}$. Then

$$\Delta\theta_t(S) := \theta_{t+1}(S) - \theta_t \approx \underbrace{-A_t \mathbf{D}_t \mathbf{m}_{t-1} - \alpha_t \lambda \theta_t}_{\text{independent of } S} - C_t \mathbf{D}_t \mathbf{g}_t(S). \tag{27}$$

Since OPUS ranks subsets by *relative* utility at fixed step $t$, the $S$-independent shift contributes an additive constant to the (first-order) utility term and does not affect ranking. Therefore, the effective *data-dependent* update can be written as

$$\Delta\theta_t(S) \approx -\mathbf{P}_t^{\text{AdamW}}\mathbf{g}_t(S) + \text{const}, \quad \mathbf{P}_t^{\text{AdamW}} := C_t \text{Diag}\left(\frac{1}{\sqrt{\overline{\mathbf{v}}_{t-1}} + \epsilon}\right), \quad C_t := \alpha_t \frac{1 - \beta_1}{1 - \beta_1^t}. \tag{28}$$

---

**ADAMW**

AdamW maintains exponential moving averages of the gradient and its elementwise square:

$$\mathbf{m}_t = \beta_1 \mathbf{m}_{t-1} + (1 - \beta_1)\mathbf{g}_t, \qquad \widehat{\mathbf{m}}_t = \mathbf{m}_t/(1 - \beta_1^t),$$
$$\mathbf{v}_t = \beta_2 \mathbf{v}_{t-1} + (1 - \beta_2)\mathbf{g}_t^{\odot 2}, \qquad \widehat{\mathbf{v}}_t = \mathbf{v}_t/(1 - \beta_2^t).$$

With decoupled weight decay, the one-step update is

$$\Delta\theta_t = -\alpha_t \frac{\widehat{\mathbf{m}}_t}{\sqrt{\widehat{\mathbf{v}}_t} + \epsilon} - \alpha_t \lambda \theta_t.$$

For online scoring at a fixed step $t$, we freeze the RMS geometry and obtain an approximate diagonal preconditioner $\mathbf{P}_t \approx \alpha_t \text{Diag}\left((\sqrt{\widehat{\mathbf{v}}_{t-1}} + \epsilon)^{-1}\right)$ that rescales coordinates before measuring utility.

---

# D. Bench-Proxy Construction

We describe how to construct BENCH-PROXY, which estimates the validation direction in Eq. (12) via the retrieval pipeline in Fig. 2(a). The goal is to build a small proxy set $\mathcal{D}_{\text{proxy}}$ that matches the target benchmark's distribution, while being sampled from the pre-training corpus so gradients can be computed efficiently and consistently during pre-training.

**Similarity scoring.** We first assign each pre-training document a benchmark relevance score based on its semantic similarity to the benchmark validation set $\mathcal{D}_{\text{val}}$. Concretely, we use a frozen sentence embedding model Arctic-Embed-L v2 (Yu et al., 2025) to encode (i) each benchmark sample and (ii) each pre-training document into a shared embedding space, and compute cosine similarities between document embeddings and benchmark embeddings. To obtain a single scalar score per document, we reduce the similarity vector by taking the maximum similarity over all benchmark samples, which captures whether a document is strongly aligned with *any* benchmark instance. This produces a scored version of the pre-training corpus, where each document is annotated with a benchmark alignment score.

**Proxy construction.** We then construct the proxy pool $\mathcal{D}_{\text{proxy}}$ by selecting the highest-scoring documents from the scored corpus. In practice, we sort documents by their benchmark relevance scores in descending order and greedily accumulate them until reaching a fixed token budget (**30M tokens** in our experiments), which yields a compact but benchmark-aligned proxy shard. During training, we repeatedly sample mini-batches from $\mathcal{D}_{\text{proxy}}$ to estimate the proxy gradient direction used for within-step ranking. This design keeps scoring stable and low-variance, while steering selection toward data that matches the target benchmark distribution.

# E. Experimental Details

**Models.** For from-scratch pre-training, we use GPT-2 Large, which has 36 layers, hidden size 1280, $\sim$774M parameters, and GPT-2 XL, which has 48 layers, hidden size 1600, $\sim$1.5B parameters. For continued pre-training, we start from Qwen3-8B-Base that has 36 layers, hidden size 4096, $\sim$8B parameters.

**Precision and compilation.** We train with mixed precision in `bfloat16`. For GPT-2 models, we keep most modules in FP32 but cast the token embedding layers to BF16 for efficiency. For Llama/Qwen-compatible models, we cast the *entire* model to BF16 to maintain dtype consistency.

**Distributed training.** All experiments run with synchronous data-parallel training using NCCL. Let $W$ be the number of GPUs (world size) and $G$ be the gradient accumulation steps; then the global batch size per optimizer update is $B = W \cdot G$ sequences of length $L$, i.e., $W \cdot G \cdot L$ update tokens per step. We apply global gradient-norm clipping with threshold 1.0.

**Sequence lengths, batch sizes for OPUS.** We use model-specific training sequence lengths due to memory constraints. For GPT-2 we set $L_{\text{train}}$=24,576 (GPT-2 Large) and $L_{\text{train}}$=6,144 (GPT-2 XL), with $L_{\text{val}}$=32,768 (Large) and $L_{\text{val}}$=8,192 (XL).[2] For OPUS, at each optimization step we score candidates using only $L_{\text{score}}$=512 tokens of each sequence. We form a candidate buffer of $N$=32 sequences for GPT-2 runs. For Qwen3-8B, we use $M$=16 as a buffer-size multiplier; selection is performed *globally* by gathering scores across all GPUs and selecting the top $K=\lfloor \rho N \rfloor$ sequences with $\rho$=0.5. We use the validation split as the proxy set for scoring (proxy batch size 8) and refresh it every step. After selection, the model performs a full forward/backward update on the selected sequences of length $L_{\text{train}}$, and the token budget is counted using $L_{\text{train}}$. The additional forward computation used for scoring is treated as overhead (Sec. 4.4). Random projection is disabled in these runs unless stated otherwise.

**Learning rate and optimization hyperparameters.** For GPT-2 XL, we use $\text{lr}_{\text{adam}}$=2×$10^{-3}$ and $\text{lr}_{\text{muon}}$=1×$10^{-2}$. AdamW uses $\beta_1$=0.8, $\beta_2$=0.95, $\epsilon$=$10^{-8}$, and no weight decay ($\lambda$=0). Muon uses momentum $\mu$=0.95 with a short warmup from $0.85 \rightarrow 0.95$ over the first 300 steps, and no weight decay. For Qwen3-8B CPT (SciPedia), we use $\text{lr}_{\text{adam}}$=$10^{-6}$ and $\text{lr}_{\text{muon}}$=$10^{-5}$ with AdamW hyperparameters $\beta_1$=0.9, $\beta_2$=0.95, and weight decay $\lambda$=0.01. We apply global gradient-norm clipping with threshold 1.0 in all experiments. The global batch per optimization step is $B = W \cdot G$ sequences of length $L$, where $W$ is the number of GPUs and $G$ is the number of gradient-accumulation steps (Qwen3-8B uses $W$=8 and $G$=1). We train Qwen3-8B for a token budget of 1.5B tokens and evaluate every 0.5B tokens. The learning-rate schedule is implemented as a piecewise multiplier over the base LR with a warmup fraction of 0.01.

**Muon optimizer configuration.** We use a hybrid optimizer in which Muon updates the matrix parameters inside Transformer

---

[2]For the Qwen3-8B CPT runs, we use $L_{\text{train}}$=4,096 and $L_{\text{val}}$=4,096 with FlexAttention.

*Table 6.* **Optimizer assignment by parameter.** In our Muon+AdamW setting, Muon is applied to matrix-shaped parameters inside Transformer blocks (`model.blocks`, ndim ≥ 2), while AdamW is applied to embeddings, LM head, and all 0/1D parameters. In the AdamW setting, AdamW is applied to all parameters. Patterns with i=0..L−1 repeat per Transformer layer.

| Model | Parameter pattern | Repeats | ndim | Optimizer | Notes |
|---|---|---|---|---|---|
| GPT2-Large | `embed.weight` | – | 2D | AdamW | Token embedding table |
| GPT2-Large | `lm_head.weight` | – | 2D | AdamW | Tied to `embed.weight` |
| GPT2-Large | `blocks.{i}.attn.qkv_proj.weight` | i=0..35 | 2D | Muon | Attention QKV projection |
| GPT2-Large | `blocks.{i}.attn.c_proj.weight` | i=0..35 | 2D | Muon | Attention output projection |
| GPT2-Large | `blocks.{i}.mlp.c_fc.weight` | i=0..35 | 2D | Muon | MLP expansion projection |
| GPT2-Large | `blocks.{i}.mlp.c_proj.weight` | i=0..35 | 2D | Muon | MLP contraction projection |
| GPT2-XL | `embed.weight` | – | 2D | AdamW | Token embedding table |
| GPT2-XL | `lm_head.weight` | – | 2D | AdamW | Tied to `embed.weight` |
| GPT2-XL | `blocks.{i}.attn.qkv_proj.weight` | i=0..47 | 2D | Muon | Attention QKV projection |
| GPT2-XL | `blocks.{i}.attn.c_proj.weight` | i=0..47 | 2D | Muon | Attention output projection |
| GPT2-XL | `blocks.{i}.mlp.c_fc.weight` | i=0..47 | 2D | Muon | MLP expansion projection |
| GPT2-XL | `blocks.{i}.mlp.c_proj.weight` | i=0..47 | 2D | Muon | MLP contraction projection |
| Qwen3-8B-Base | `embed.weight` | – | 2D | AdamW | Token embedding table |
| Qwen3-8B-Base | `lm_head.weight` | – | 2D | AdamW | Tied |
| Qwen3-8B-Base | `ln_f.weight` | – | 1D | AdamW | Final RMSNorm weight |
| Qwen3-8B-Base | `blocks.{i}.input_layernorm.weight` | i=0..35 | 1D | AdamW | RMSNorm weight |
| Qwen3-8B-Base | `blocks.{i}.post_attention_layernorm.weight` | i=0..35 | 1D | AdamW | RMSNorm weight |
| Qwen3-8B-Base | `blocks.{i}.self_attn.q_norm.weight` | i=0..35 | 1D | AdamW | QK-norm weight |
| Qwen3-8B-Base | `blocks.{i}.self_attn.k_norm.weight` | i=0..35 | 1D | AdamW | QK-norm weight |
| Qwen3-8B-Base | `blocks.{i}.self_attn.q_proj.weight` | i=0..35 | 2D | Muon | Attention Q projection |
| Qwen3-8B-Base | `blocks.{i}.self_attn.k_proj.weight` | i=0..35 | 2D | Muon | Attention K projection |
| Qwen3-8B-Base | `blocks.{i}.self_attn.v_proj.weight` | i=0..35 | 2D | Muon | Attention V projection |
| Qwen3-8B-Base | `blocks.{i}.self_attn.o_proj.weight` | i=0..35 | 2D | Muon | Attention output projection |
| Qwen3-8B-Base | `blocks.{i}.mlp.gate_proj.weight` | i=0..35 | 2D | Muon | SwiGLU gate projection |
| Qwen3-8B-Base | `blocks.{i}.mlp.up_proj.weight` | i=0..35 | 2D | Muon | SwiGLU up projection |
| Qwen3-8B-Base | `blocks.{i}.mlp.down_proj.weight` | i=0..35 | 2D | Muon | SwiGLU down projection |
| All | `(any remaining parameters)` | – | any | AdamW | |

blocks (parameters with ndim ≥ 2), excluding the token embedding table and the final LM head. All remaining parameters are updated with AdamW. Muon applies SGD with momentum ($\mu = 0.95$) with no weight decay, followed by an orthogonalization post-processing step on each 2D update. Specifically, we run a Newton–Schulz quintic iteration for 5 steps in BF16 to produce an approximate zeroth-power transform, serving as an efficient surrogate to the $UV^{\top}$ factor in SVD-based orthogonalization. To stabilize updates across differently-shaped matrices, we rescale the effective learning rate for each matrix parameter $W \in \mathbb{R}^{m \times n}$ as

$$\eta_{\text{eff}} = \eta \cdot \sqrt{\max\left(1, \frac{m}{n}\right)}.$$

For the AdamW-updated parameter groups in this hybrid setup, we use $\beta_1 = 0.8$, $\beta_2 = 0.95$, $\epsilon = 10^{-8}$, and weight decay $\lambda = 0$, synchronizing gradients via memory-efficient reduce-scatter when dimensions are divisible by the world size and otherwise falling back to all-reduce for correctness.

**AdamW optimizer configuration.** For settings that use AdamW, we update all model parameters—including token embeddings, all Transformer block parameters, and the final LM head—with a distributed AdamW optimizer using $\beta_1 = 0.8$, $\beta_2 = 0.95$, $\epsilon = 10^{-8}$, and weight decay $\lambda = 0$. Gradients are synchronized using reduce-scatter when tensor dimensions are divisible by the world size, and otherwise using all-reduce to ensure numerically correct distributed updates.

**Optimizer assignment.** For clarity and reproducibility, we explicitly specify how parameters are assigned to optimizers in our experimental settings (Table 6). In the **Muon+AdamW** setting, we apply Muon updates only to *matrix-shaped* parameters inside Transformer blocks, i.e., parameters under `model.blocks` with ndim ≥ 2 (e.g., attention and MLP projection matrices). All remaining parameters—including token embeddings, the LM head, and all 0/1D parameters such as RMSNorm weights and biases—are optimized with a distributed AdamW optimizer. This hybrid design follows the recommended usage of Muon, which is intended for 2D matrices and is not directly applicable to 0/1D parameter types. In the **AdamW** setting, we instead optimize all parameters with AdamW optimizer.

**Random projection configuration.** To accelerate OPUS scoring, we apply a CountSketch-based random projection to

*Table 7.* **Benchmark evaluation configuration.** For most benchmarks we use multiple-choice perplexity: score each candidate option by negative log-likelihood and choose the best-scoring option; we report accuracy. MMLU is evaluated separately using zero-shot and log-likelihood on the entire answer following FineWeb-Edu.

| Benchmark | Domain | #Choices | Eval mode | Metric |
|---|---|---|---|---|
| *Core Benchmarks (in-domain)* | | | | |
| MMLU | Knowledge | 4 | LL | Accuracy |
| ANLI | Understanding | 3 | PPL | Accuracy |
| HellaSwag | Commonsense and Reasoning | 4 | PPL | Accuracy |
| PIQA | Commonsense and Reasoning | 2 | PPL | Accuracy |
| SIQA | Commonsense and Reasoning | 3 | PPL | Accuracy |
| WinoGrande | Language | 2 | LL | Accuracy |
| ARC-Easy | Science and Reasoning | 4 | PPL | Accuracy |
| ARC-Challenge | Science and Reasoning | 4 | PPL | Accuracy |
| CommonsenseQA | Commonsense and Reasoning | 5 | PPL | Accuracy |
| WSC | Language | 2 | PPL | Accuracy |
| *Other Benchmarks (out-of-domain)* | | | | |
| BBH | Reasoning (hard) | – | Generation | Exact Match |
| RACE-Middle | Understanding | 4 | PPL | Accuracy |
| RACE-High | Understanding | 4 | PPL | Accuracy |
| AX-b | Language | 2 | PPL | Accuracy |
| AX-g | Language | 2 | PPL | Accuracy |
| StoryCloze | Understanding | 2 | PPL | Accuracy |

per-sample gradients, implementing the sketching operator. Concretely, for each trainable linear weight we form the per-sample gradient in outer-product form (aggregated over time when applicable) and then sketch the flattened gradient into an $m$-dimensional vector using CountSketch with a deterministic hash/sign pair; this yields an unbiased estimator of inner products, $\mathbb{E}\langle\Pi(g_1),\Pi(g_2)\rangle = \langle g_1, g_2\rangle$, enabling us to compute gradient dot-products (and similarity matrices) in the projected space. We set the sketch dimension to $m = 8192$ with seed 42, which provides substantial compression for GPT-2 XL where the largest matrix-gradient has dimension on the order of 10.24M, corresponding to an effective compression of roughly $1250\times$ while preserving the ranking signal used by OPUS. The projection is enabled during scoring and uses cached hash/sign tensors per parameter shape for efficiency; when disabled, we fall back to exact full-dimensional dot-products.

## F. Evaluation Details

**GPT-2 pretraining evaluation.** We evaluate all GPT-2 pretraining checkpoints on a variety of benchmarks target diverse capabilities. See Table 7 for the summary of the configurations.

We evaluate on the following benchmarks to test the general capabilities of our pretrained models:

- **MMLU** (Hendrycks et al., 2021): broad factual and academic knowledge across many subjects.

- **ANLI** (Nie et al., 2020): adversarial natural language inference, testing robust entailment and contradiction reasoning.

- **HellaSwag** (Zellers et al., 2019): commonsense reasoning for plausible continuations.

- **PIQA** (Bisk et al., 2020): physical commonsense reasoning about everyday actions.

- **SIQA** (Sap et al., 2019): social commonsense and intent reasoning.

- **WinoGrande** (Sakaguchi et al., 2021): pronoun/coreference resolution with adversarial bias reduction.

- **ARC-E / ARC-C** (Clark et al., 2018): grade-school science questions; Easy and Challenge splits measure increasing reasoning difficulty.

- **CommonsenseQA** (Talmor et al., 2019): commonsense knowledge and reasoning over concepts.

- **WSC** (Levesque et al., 2012): hard coreference requiring commonsense.

*Table 8.* FineWeb results after 30B update tokens for GPT-2 Large pre-trained on FineWeb with the Muon optimizer under varying buffer size $b_t$, temperature $\tau$ and CountSketch projection dimension $m$. See sampling and validation strategy ablations at Table 5

| Method | MMLU | ANLI | HellaSwag | PIQA | SIQA | W.G. | ARC-E | ARC-C | C.QA | WSC | Avg. |
|---|---|---|---|---|---|---|---|---|---|---|---|
| *GPT-2 Large with Muon optimizer ($\tau = 0.9\ m = 8192$)* | | | | | | | | | | | |
| Random | 28.46 | 32.93 | 42.71 | 69.70 | 40.07 | 49.17 | 37.57 | 28.14 | 31.94 | 36.54 | **39.72** |
| *GPT-2 Large with Muon optimizer on different buffer size $b_t$ ($\tau = 0.9\ d = 8192$)* | | | | | | | | | | | |
| OPUS (Buffer size 16) | 28.37 | 33.30 | 42.60 | 69.53 | 40.02 | 48.78 | 38.45 | 27.46 | 32.51 | 36.54 | **39.76** |
| OPUS (Buffer size 32) | 29.23 | 33.36 | 42.76 | 70.4 | 39.30 | 49.72 | 37.39 | 25.42 | 33.42 | 36.54 | **39.75** |
| OPUS (Buffer size 64) | 28.76 | 33.12 | 42.92 | 69.97 | 39.56 | 50.43 | 38.98 | 29.15 | 33.09 | 36.54 | **40.25** |
| *GPT-2 Large with Muon optimizer on different temperature $\tau$ ($b_t = 64\ m = 8192$)* | | | | | | | | | | | |
| OPUS (temperature 0.8) | 28.54 | 34.19 | 42.92 | 69.59 | 40.23 | 49.33 | 37.92 | 26.78 | 32.76 | 36.54 | **39.88** |
| OPUS (temperature 1.0) | 28.62 | 33.64 | 43.63 | 70.46 | 39.97 | 50.12 | 37.21 | 24.41 | 32.19 | 38.46 | **39.87** |
| OPUS (temperature 0.9) | 28.76 | 33.12 | 42.92 | 69.97 | 39.56 | 50.43 | 38.98 | 29.15 | 33.09 | 36.54 | **40.25** |
| *GPT-2 Large with Muon optimizer on different CountSketch projection dimension $m$ ($b_t = 64\ \tau = 0.9$)* | | | | | | | | | | | |
| OPUS (projection dimension 4096) | 28.57 | 33.46 | 42.75 | 68.39 | 40.79 | 48.46 | 38.27 | 26.10 | 33.01 | 36.54 | **39.63** |
| OPUS (projection dimension 16384) | 28.31 | 33.47 | 42.64 | 70.02 | 40.33 | 49.57 | 36.68 | 22.71 | 32.19 | 37.50 | **39.34** |
| OPUS (projection dimension 8192) | 28.76 | 33.12 | 42.92 | 69.97 | 39.56 | 50.43 | 38.98 | 29.15 | 33.09 | 36.54 | **40.25** |

For all above benchmarks except for MMLU, we use OpenCompass (Contributors, 2023) with a multiple-choice perplexity scoring rule: for each candidate answer option, we compute its average negative log-likelihood conditioned on the prompt, and predict the option with the lowest perplexity; we then report accuracy. For WinoGrande, we follow the OpenCompass log-likelihood variant that compares the likelihood of the two candidates. All these benchmarks are evaluated zero-shot. MMLU is evaluated separately with Lighteval (Habib et al., 2023) following the implementation in FineWeb-Edu (Penedo et al., 2024) evaluation protocol. Since the typical MMLU implementation (which uses "A", "B", etc as answer targets) gives generally random results on non instruction tuned models, instead, we use the full MMLU answer as the target. We also use zero-shot prompting and then select the answer by comparing the log-likelihood of the entire option string.

In addition, we use the following benchmarks that are not in our bench-proxy set for the generalization evaluation:

- **BBH** (Suzgun et al., 2023): a challenging subset of BIG-Bench tasks emphasizing multi-step reasoning. We select a set of BBH tasks where base models produce non-degenerate outputs: Tracking Shuffled Objects, Reasoning about Colored Objects, Logical Deduction, Disambiguation QA, Penguins in a Table, and Sports Understanding.

- **RACE-M / RACE-H** (Lai et al., 2017): exam-style reading comprehension with multiple choice questions; we use the Middle and High school subsets.

- **AX-B / AX-G** (Wang et al., 2019): diagnostic evaluation sets from SuperGLUE designed to stress-test linguistic phenomena and generalization.

- **StoryCloze** (Mostafazadeh et al., 2016): story ending prediction to test narrative coherence and commonsense continuation.

We evaluate these benchmarks using the OpenCompass framework. All these benchmarks are evaluated zero-shot except for BBH, which uses three-shot. For BBH, many subtasks are near-chance at our model scale, so an aggregate score over all subtasks becomes unstable and less informative. We therefore report results on the curated subset above, where the base model achieves non-trivial accuracy and methods exhibit meaningful separation.

**CPT evaluation.** We evaluate continued pre-training checkpoints of Qwen3-8B-Base on two science focused benchmarks, OlympicArena (Huang et al., 2024) and SciAssess (Cai et al., 2025). For OlympicArena, we evaluate on the test split and use zero-shot prompting. For SciAssess, we evaluate four subdomains in biology, chemistry, material, medicine using a 3-shot prompting setting with chain-of-thought enabled where available. We use stochastic decoding with temperature 0.6, top-$p = 0.95$, and top-$k = 20$, and max sequence length of 1024. We report the official accuracy metric for both benchmarks.

## G. Additional Ablation Results

We report our additional ablation results about different hyperparameters in Table 8.

*Table 9.* Remaining settings for FineWeb results after 30B update tokens. Bold marks the best compute-matched method per benchmark within each block; Random (60B) is shown as a non compute-matched reference. Main results in Table 1.

| Method | MMLU | ANLI | HellaSwag | PIQA | SIQA | W.G. | ARC-E | ARC-C | C.QA | WSC | Avg. |
|---|---|---|---|---|---|---|---|---|---|---|---|
| *GPT-2 Large with Muon optimizer on 30B update tokens of FineWeb* | | | | | | | | | | | |
| Random | 28.46 | 32.93 | 42.71 | 69.70 | 40.07 | 49.17 | 37.57 | 28.14 | 31.94 | 36.54 | 39.72 |
| PPL | 28.40 | 33.24 | 42.69 | 70.13 | 40.17 | 48.38 | 36.16 | 23.05 | 31.86 | 36.54 | 39.06 |
| GREATS (Wang et al., 2024) | 28.49 | 33.31 | 42.22 | 70.18 | 39.46 | 49.41 | 36.86 | 24.41 | 33.25 | 36.54 | 39.41 |
| QuRating (Wettig et al., 2024) | **31.53** | **34.12** | 39.47 | 66.38 | 39.82 | **50.59** | **40.92** | **30.51** | 30.22 | **38.46** | 40.20 |
| DSIR (Xie et al., 2023) | 28.50 | 33.39 | 43.04 | 69.70 | **40.53** | 49.64 | 37.39 | 24.41 | 32.27 | 36.54 | 39.54 |
| DCLM-FastText (Li et al., 2024) | 29.36 | 33.17 | 44.26 | **71.16** | 39.82 | 49.96 | 37.92 | 24.75 | 32.02 | 36.54 | 39.90 |
| FineWeb-Edu (Penedo et al., 2024) | 28.83 | 32.67 | 43.09 | 70.02 | 40.28 | 47.75 | 39.15 | 24.75 | 33.66 | **38.46** | 39.87 |
| UltraFineweb (Wang et al., 2025c) | 29.00 | 32.99 | **44.38** | 71.11 | 40.17 | 48.78 | 37.57 | 25.08 | **33.91** | **38.46** | 40.15 |
| OPUS (Ours) | 28.76 | 33.12 | 42.92 | 69.97 | 39.56 | 50.43 | 38.98 | 29.15 | 33.09 | 36.54 | **40.25** |
| Random (60B) | 28.70 | 33.23 | 45.20 | 71.16 | 40.79 | 49.41 | 39.68 | 25.42 | 31.12 | 36.54 | 40.13 |
| *GPT-2 XL with AdamW optimizer on 30B update tokens of FineWeb* | | | | | | | | | | | |
| Random | 28.76 | 33.56 | **46.63** | 70.35 | 42.37 | 49.19 | 39.15 | 24.41 | 32.68 | 36.54 | 40.36 |
| PPL | 29.32 | 33.67 | 45.31 | 70.08 | 41.71 | 49.72 | 39.68 | 24.75 | 31.29 | **38.46** | 40.02 |
| GREATS (Wang et al., 2024) | 28.81 | 33.49 | 40.73 | 69.53 | **42.48** | 49.01 | 34.22 | 24.75 | 31.04 | **38.46** | 39.25 |
| QuRating (Wettig et al., 2024) | **32.24** | 32.61 | 34.66 | 66.65 | 38.54 | **50.43** | 36.86 | 24.75 | 28.42 | 36.54 | 38.71 |
| DSIR (Xie et al., 2023) | 29.37 | 33.09 | 45.88 | 70.67 | 39.97 | 47.51 | 38.80 | 24.41 | 33.42 | 36.54 | 39.97 |
| DCLM-FastText (Li et al., 2024) | 29.43 | **34.47** | 42.45 | 69.91 | 41.86 | 47.59 | 36.33 | 24.41 | 31.53 | 36.54 | 39.45 |
| FineWeb-Edu (Penedo et al., 2024) | 29.71 | 33.51 | 46.62 | **71.93** | 41.91 | 46.88 | **40.04** | 25.08 | 32.10 | 36.54 | 40.43 |
| UltraFineweb (Wang et al., 2025c) | 29.25 | 33.51 | 41.76 | 69.21 | 41.40 | 49.57 | 37.92 | 24.07 | 32.76 | 36.54 | 39.60 |
| OPUS (Ours) | 29.43 | 33.51 | 46.12 | 70.35 | 41.35 | 50.36 | 39.33 | **29.15** | **33.99** | 36.54 | **41.01** |
| Random (60B) | 29.55 | 33.57 | 48.75 | 72.09 | 41.10 | 48.78 | 40.92 | 27.12 | 34.48 | 36.54 | 41.29 |

*Table 10.* **Additional evaluation on FineWeb-Edu dataset**. We evaluate OPUS under a constrained setting: it dynamically selects from the mid-quality subset (score 3), whereas baselines are trained on the superior high-quality partitions (scores ≥ 4). Main results in Table 2

| Method | MMLU | ANLI | HellaSwag | PIQA | SIQA | W.G. | ARC-E | ARC-C | C.QA | WSC | Avg. |
|---|---|---|---|---|---|---|---|---|---|---|---|
| *GPT-2 Large with Muon optimizer on 30B update tokens of FineWeb-Edu* | | | | | | | | | | | |
| Random (Score 3) | 30.52 | 33.16 | 43.95 | 68.87 | 40.58 | 49.02 | 48.39 | 25.08 | 35.54 | 36.54 | 41.17 |
| Random (Score 4+5) | 32.92 | 33.38 | 41.95 | 67.46 | 38.84 | 47.75 | 53.97 | 29.15 | 30.79 | 36.54 | 41.28 |
| PPL (Score 4+5) | **33.17** | 33.87 | 42.25 | 67.63 | **40.33** | 48.22 | 50.79 | 28.47 | 29.48 | **38.46** | 41.27 |
| GREATS (Score 4+5) | 32.73 | **34.38** | 45.86 | **70.95** | 39.30 | 50.36 | 44.62 | 24.75 | 32.92 | **38.46** | 41.43 |
| QuRating (Score 4+5) | 31.32 | 34.07 | 41.70 | 66.92 | 39.71 | 47.83 | 50.79 | 32.88 | 31.94 | 36.54 | 41.37 |
| DSIR (Score 4+5) | 32.54 | 33.54 | 41.07 | 67.95 | 39.36 | 47.28 | 48.68 | **33.90** | 29.57 | **38.46** | 41.24 |
| DCLM-FastText (Score 4+5) | 32.64 | 33.67 | 41.66 | 66.38 | 38.74 | **51.30** | 49.38 | 30.85 | 31.04 | 36.54 | 41.22 |
| FineWeb-Edu (Score 4+5) | 32.00 | 33.46 | 39.95 | 64.74 | 39.87 | 50.51 | 52.20 | 29.15 | 30.30 | 36.54 | 40.87 |
| UltraFineweb (Score 4+5) | 32.60 | 33.02 | 40.70 | 66.05 | 38.23 | 49.72 | 48.32 | 30.17 | 29.24 | 36.54 | 40.46 |
| OPUS (Score 3) | 30.39 | 34.31 | **46.36** | 70.51 | 39.41 | 50.20 | 45.33 | 28.47 | **33.74** | **38.46** | 41.72 |
| OPUS (Score 4+5) | 32.17 | 33.38 | 42.52 | 67.30 | 39.51 | 51.07 | **54.14** | 30.85 | 31.04 | **38.46** | **42.04** |
| Random (60B) (Score 4+5) | 33.21 | 34.03 | 43.66 | 67.95 | 40.07 | 50.04 | 52.56 | 31.86 | 31.61 | 36.54 | 42.15 |

# H. Additional Results

**Additional pretraining results.** For complete per-task results, we refer to the main FineWeb results in Tables 1 and 2. For the remaining FineWeb settings not shown in the main text, we report those in Table 9 and Table 10. Together, these provide the full set of scratch pre-training results across all evaluated model/optimizer settings.

**Approximation fidelity study.** The practical OPUS score involves several approximations for tractability, including curvature simplification, the ghost inner product factorization, a first-order Taylor surrogate, and a frozen optimizer preconditioner. Our claim is not that this score is an exact estimator of the marginal utility, but that it is a tractable surrogate faithful enough to preserve candidate ranking. To verify this claim, we run an approximation-fidelity study on GPT-2 XL trained from scratch on FineWeb with Muon, at three checkpoints covering early, mid, and late training. For each component, we isolate its effect by comparing against the exact non-approximated counterpart. Val. Err. $\ell_2$ measures the $\ell_2$ distance between the exact and approximated values averaged over candidates, and Util. Err. measures the absolute difference

*Table 11.* **Approximation fidelity study on GPT-2 XL with Muon on FineWeb.** Checkpoints at step 61,035 (early, ∼3B tokens), step 301,575 (mid, ∼15B tokens), and step 610,350 (late, ∼30B tokens). Val. Err. $\ell_2$ is the $\ell_2$ distance between exact and approximated values averaged over candidates. Util. Err. is the absolute difference between exact and approximated utility scores averaged over candidates. Smaller is better.

| Approximation Component | Early (∼3B) | | Mid (∼15B) | | Late (∼30B) | |
|---|---|---|---|---|---|---|
| | Val. Err. $\ell_2$ ↓ | Util. Err. ↓ | Val. Err. $\ell_2$ ↓ | Util. Err. ↓ | Val. Err. $\ell_2$ ↓ | Util. Err. ↓ |
| Curvature simplification, $\mathbf{H}_{\text{val}} \approx \mathbf{I}$ | $7.50 \times 10^{-9}$ | $3.67 \times 10^{-6}$ | $1.03 \times 10^{-10}$ | $6.51 \times 10^{-8}$ | $1.09 \times 10^{-9}$ | $6.07 \times 10^{-7}$ |
| Ghost inner product factorization | $2.04 \times 10^{-6}$ | $1.32 \times 10^{-5}$ | $5.71 \times 10^{-8}$ | $3.89 \times 10^{-8}$ | $1.33 \times 10^{-7}$ | $6.97 \times 10^{-9}$ |
| First-order Taylor surrogate | $1.35 \times 10^{-5}$ | $2.27 \times 10^{-1}$ | $6.39 \times 10^{-8}$ | $7.21 \times 10^{-3}$ | $2.83 \times 10^{-7}$ | $1.62 \times 10^{-3}$ |
| AdamW preconditioner, $\mathbf{P}_t\nabla \approx \text{diag}(v_t)^{-1/2}\nabla$ | $2.52 \times 10^{-10}$ | $2.41 \times 10^{-2}$ | $4.70 \times 10^{-11}$ | $3.02 \times 10^{-4}$ | $3.25 \times 10^{-10}$ | $5.34 \times 10^{-1}$ |
| Muon preconditioner, $\mathbf{P}_t\nabla \approx \text{NS}(\mathbf{G})\nabla$ | $1.35 \times 10^{-5}$ | $2.30 \times 10^{-1}$ | $6.00 \times 10^{-8}$ | $5.69 \times 10^{-3}$ | $1.48 \times 10^{-7}$ | $2.48 \times 10^{-3}$ |

between exact and approximated utility scores averaged over candidates.

As shown in Table 11, all components yield errors small enough to preserve faithful candidate ranking across training. The curvature simplification and the ghost factorization are essentially exact at every checkpoint. The Taylor surrogate and the optimizer preconditioner introduce somewhat larger Util. Err. early in training, but the errors decrease as training progresses and remain in a range that preserves ranking. What matters for selection is preserving the ranking of candidates rather than exact point-wise reconstruction of the utility, and this is consistent with the strong downstream gains observed under both AdamW and Muon in our main results.

**Validation loss curves on FineWeb-Edu dataset.** We report validation-loss trajectories in Figure 6 for GPT-2 XL and GPT-2 Large trained from scratch on FineWeb-Edu under the same training recipe and a fixed budget of 30B update tokens. To make the comparison conservative for OPUS, OPUS selects dynamically from the mid-quality pool with score 3, whereas the baselines are trained on the high-quality pool with scores 4+5. All curves are evaluated on the same held-out FineWeb-Edu validation split. We also include a longer-training Random reference at 60B update tokens (not compute-matched) to contextualize convergence speed.

As shown in Fig. 6, OPUS consistently improves optimization dynamics for both model scales: across training, it attains lower validation loss than representative baselines despite selecting from the lower-quality candidate pool. For GPT-2 XL, OPUS reaches the validation loss achieved by Random trained for 60B update tokens using only 17B update tokens, demonstrating substantially faster convergence. For GPT-2 Large, OPUS exhibits the same trend and maintains a clear gap over baselines throughout training.

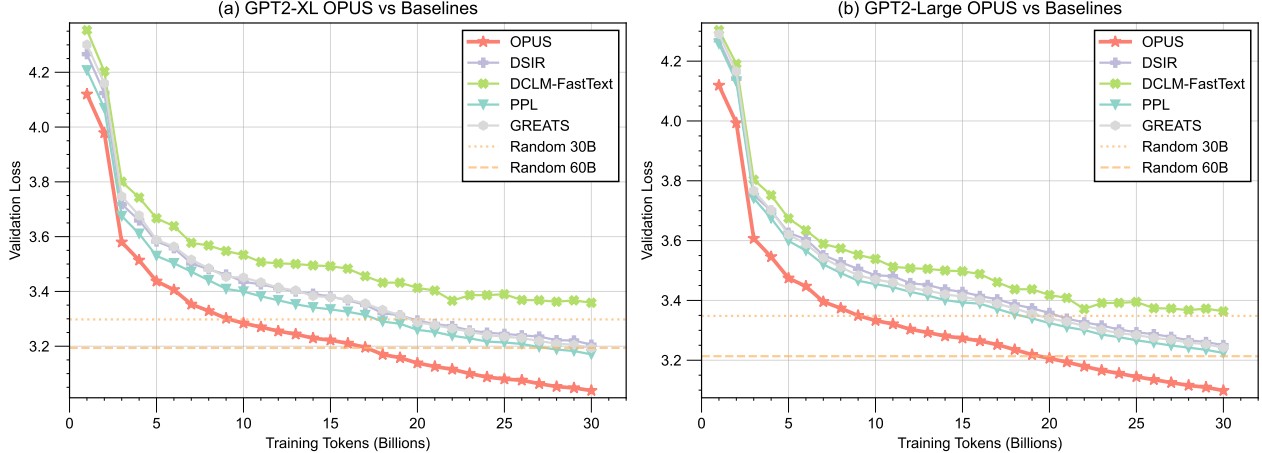

*Figure 6.* Validation-loss curves on GPT-2 XL and GPT-2 Large pre-trained from scratch on FineWeb-Edu dataset. Left: Results on GPT-2 XL. OPUS compared with representative baselines trained on the high-quality pool, with Random 60B shown as a non compute-matched reference. Curves are shown up to 30B update tokens for compute-matched comparison. Right: Results on GPT2-Large.

**Detailed domain-wise CPT Results.** Figure 7 reports domain breakdowns for continued pre-training on SciencePedia across three token budgets 0.5B, 1B, and 1.5B. Across OlympicArena (Fig. 7a) OPUS consistently improves over the base Qwen3-8B-Base and the compute-matched Random baseline in most scientific domains like physics, chemistry, biology, and

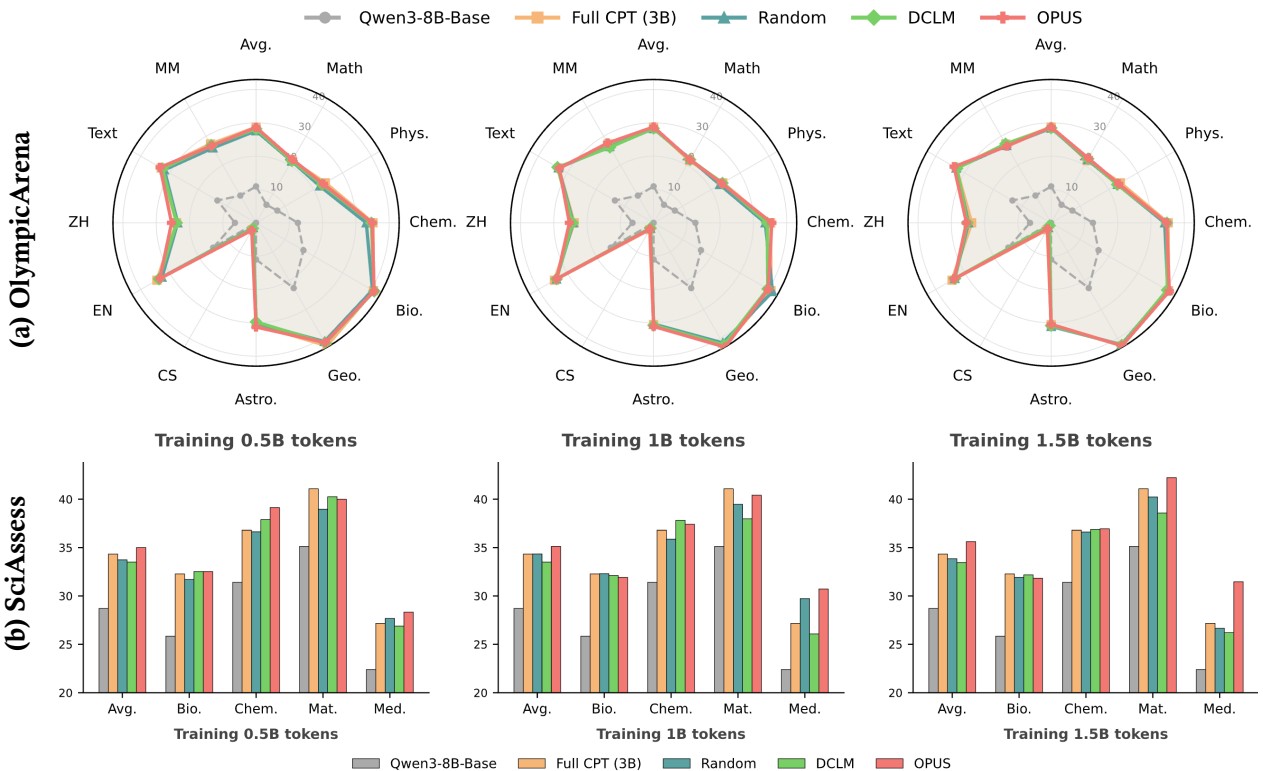

*Figure 7.* **CPT domain breakdown on SciencePedia.** Domain-level accuracy of Qwen3-8B-Base and CPT baselines across three token budgets 0.5B, 1B, and 1.5B. Rows correspond to the CPT token budget. Columns show (a) OlympicArena domains with an appended Avg. and (b) SciAssess domains. For each panel, we compare Qwen3-8B-Base, Full CPT (3B), Random, DCLM, and OPUS. All results use the official benchmark metrics.

geography, as well as the text-only and multimodal subsets., with gains that are broadly distributed rather than concentrated in a single category. Importantly, OPUS is competitive with, and sometimes surpasses, DCLM and even the Full CPT reference despite using at most 1.5B update tokens, indicating strong data efficiency. On SciAssess (Fig. 7b), OPUS yields substantial gains on the material and medicine subsets and ties the best baseline on chemistry, leading to the highest average overall, again with at most 1.5B update tokens.

**Hyperparameter sensitivity analysis.** We conduct further ablation studies in Appendix G on key hyperparameters of OPUS, including (i) the candidate buffer size $b_t$, (ii) the Boltzmann sampling temperature $\tau$, and (iii) the CountSketch projection dimension $m$ (Table 8). Overall, OPUS is reasonably stable across the tested settings and improves over random selection in most configurations. Increasing the buffer size tends to help, with $b_t=64$ yielding the best average performance among the evaluated choices. For stochastic selection, a moderate temperature offers a better exploration–exploitation trade-off: $\tau=0.9$ performs best compared to both a lower temperature (more greedy) and a higher temperature (closer to uniform sampling). For random projection, we observe sensitivity to the sketch dimension: $m=8192$ provides the strongest results among the tested dimensions. Based on these results, we adopt $b_t=64$, $\tau=0.9$, and $m=8192$ as our default configuration.

**Qualitative comparison of selected samples** We show the selection from a single candidate buffer of size $N=32$ and selected $K=16$ samples. For each method, we display selected candidates and not selected samples, candidate index, and the method's raw score. Overall, OPUS tends to select a more diverse mixture of documents, covering both instructional content and broader web text, rather than concentrating on a narrow "educational-only" slice. In contrast, several static filtering method exhibit more extreme preferences—either strongly favoring highly low-diversity patterns or focusing on a limited subset of high-loss samples. These examples support our empirical findings: OPUS's optimizer-aware utility and stochastic sampling encourage selections that remain broadly suitable for general-purpose pre-training, while still being guided towards high quality samples that align with the proxy direction.

## Random

**Sample 1**
**Candidate #0**
Selected
score=–

As it turns out, the exercises synonymous with strong, attractive abs may not be the best way to train your core—and may be doing damage to your back. Read more If you are worried about the excess holiday pounds many of us are still carrying around. There are a few easy, natural things you can do to shed them, and none of them requires an. . .

**Sample 2**
**Candidate #1**
Selected
score=–

Wedding & Party Venues - Sort By: Edgartown : (508) 627-9510 A 19th century gothic revival home transformed into the island's premier eco-boutique hotel. Guests either stay in the 17-room Hob Knob hotel or in the privacy of their own Hob Knob House. Guests can expect individualized Hob Knob hospitality and modern luxury amenities in a rel. . .

**Sample 3**
**Candidate #2**
Selected
score=–

With the advent of new technologies for sneakers such as Vac Tech, Hyperfuse and Flyknit, the mid 90s and early 2000s methods of production and designing are becoming obsolete in this sneaker world. Nike Running is the future for Nike, generating billions of dollars per year, and we see Nike also not afraid to experiment with technology s. . .

**Sample 4**
**Candidate #3**
Selected
score=–

starring John Travolta and Sam Jackson The first thing to understand about Basic –the basic thing, let's say– is that although the commercials make it look like a war movie, it is not, for which we can all be grateful. No, Basic is a plot-twisty whodunnit. If The Usual Suspects died, and its body turned to cheese, and then that cheese-b. . .

**Sample 5**
**Candidate #4**
Selected
score=–

5 Types of Women's Underwear That Men Love Underwear can say a lot about a woman. It's something that men are obsessed with, to the point that, a mere glimpse of a thong waistband causes us to go into shock. On the surface we find them sexy, revealing. We're able to see who a woman actually is—or maybe some guys are just plain horny. Howe. . .

**Sample 6**
**Candidate #7**
Selected
score=–

Elizabeth Hurley played as Dalila Release: Dec 8, 1996 Mara and her husband Manoa are both upstanding and religious Israelites living under the harsh and unjust rule of the Philistines. Much to their regret, they have not been able to have children. One day, a mysterious stranger appears to Mara and promises her that she will bear a son w. . .

**Sample 7**
**Candidate #8**
Selected
score=–

The Unsung Heroes of Your HVAC System: Understanding the Importance of Filters When it comes to your HVAC (Heating, Ventilation, and Air Conditioning) system, you might be quick to think about the thermostat, air ducts, or even the unit itself. However, there's an unsung hero in your HVAC system that plays a pivotal role in maintaining in. . .

**Sample 8**
**Candidate #13**
Selected
score=–

In Heart of Darkness it is the white invaders for instance, who are, almost without exception, embodiments of blindness, selfishness, and cruelty; and even in the cognitive domain, where such positive phrases as "to enlighten," for instance, are conventionally opposed to negative ones such as "to be in the dark," the traditional expectati. . .

**Sample 9**
**Candidate #17**
Selected
score=–

Political Parties and Elections Political parties are an established part of modern mass democracy, and the conduct of elections in India is largely dependent on the behaviour of political parties. Although many candidates for Indian elections are independent, the winning candidates for Lok Sabha and Vidhan Sabha elections usually stand a. . .

**Sample 10**
**Candidate #18**
Selected
score=–

This article originally appeared in the December 2015 issue of Resource Recycling. Subscribe today for access to all print content. Since the 1990s, curbside and drop-off recycling has grown substantially – nearly 90 percent of households now have access, according to recent surveys from Moore Recycling Associates, the American Forest and. . .

**Sample 11**
**Candidate #21**
Selected
score=–

Nestled in the leafy suburbs of western Berlin, the Wannsee Conference House stands as a poignant reminder of a dark chapter in human history. The Wannsee Conference: A Pivotal Moment The Wannsee Conference, held on January 20, 1942, marked a pivotal moment in the implementation of Nazi Germany's genocidal plans. Organized by SS-Obergrupp. . .

**Sample 12**
**Candidate #23**
Selected
score=–

The St. James kindergarteners have been working up to Project Week over the past month. We started slowly by taking walks in our neighborhood while Ms. Meghan and I noted what caught the children's interest. It became apparent that the class was very interested in the L trains that they saw on our walks. It started with a simple question,. . .

**Sample 13**
**Candidate #27**
Selected
score=–

24/7 writing help on your phone Save to my list Remove from my list In the tumultuous 19th century, both Italy and Germany found themselves fragmented into numerous separate ruling states. The impetus for change came in the form of rising nationalism and liberalism, paving the way for the unification of these disparate entities. However,. . .

**Sample 14**
**Candidate #29**
Selected
score=–

Earthquakes are the result of sudden movement along faults within the Earth. The movement releases stored-up 'elastic strain' energy in the form of seismic waves, which propagate through the Earth and cause the ground surface to shake. Such movement on the faults is generally a response to long-term deformation and the buildup of stress.. . .

**Sample 15**
**Candidate #30**
Selected
score=–

Over 1.8 million professionals use CFI to learn accounting, financial analysis, modeling and more. Start with a free account to explore 20+ always-free courses and hundreds of finance templates and cheat sheets. What is the Central Limit Theorem (CLT)? The Central Limit Theorem (CLT) is a statistical concept that states that the sample me. . .

**Sample 16**
**Candidate #31**
Selected
score=–

One of the challenges of working with ancient DNA samples is that damage accumulates over time, breaking the double helix structure into ever-smaller fragments. In the samples we worked with, these fragments were scattered and mixed with contaminants, making genome reconstruction a major technical challenge. But a shocking paper published. . .

**Sample 17**
**Candidate #5**
Not selected
score=–

Well this is the big one. So big apparently, that I had to take it there and raise the number from 10 to 15. There's just that many fails in the world of female rap. Some slight missteps, some EPIC. Nevertheless, they are all worth mentioning. You can probably think of a bunch more, but this is what I have gathered picking up from my prev. . .

**Sample 18**
**Candidate #6**
Not selected
score=–

Skaters need to check their skate helmets every so often and ask yourself, "Is it time to replace this helmet?" Well, that depends. Did you crash in it? For starters, most people are aware that you must replace a helmet after any crash where your head hit. The foam part of a helmet is made for one-time use, and after crushing once it is n. . .

**Sample 19**
**Candidate #9**
Not selected
score=–

"Last night three cargoes of Bohea Tea were emptied into the sea. This is the most magnificent movement of all. There is a dignity, a majesty, a sublimity, in this last effort of the Patriots that I greatly admire." - John Adams, diary entry, December 17, 1773 - John Adams, diary entry, December 17, 1773 A Novel Idea Is something so new a. . .

**Sample 20**
**Candidate #10**
Not selected
score=–

Deforestation isn't just happening in well-known global hotspots like Indonesia and Brazil's rainforest. A new analysis says forests are also shrinking on state and private land in Oregon, where an estimated 522,000 acres of forest cover have disappeared since 2000. That's an area six times larger than the city of Portland, equal to more. . .

**Sample 21**
**Candidate #11**
Not selected
score=–

In decades past, classroom design was often an afterthought and followed a standardised layout. Plain boxed shaped classrooms, with identical chairs and tables throughout were commonplace in many schools. Read the latest issue of School News HERE Recently, though, there has been a shift away from this one-size-fits all approach to classro. . .

**Sample 22**
**Candidate #12**
Not selected
score=–

Can you please give us a little short bio? (education, professional experiences, select publications, academic specialty, awards won) Public school teacher for 5 years BA art (UC Irvine) PhD. (UCLA) educational psychology Professor of Child Development, (25 years) CSUS Senior Research Scientist (Oregon Research Institute with Institute of. . .

**Sample 23**
**Candidate #14**
Not selected
score=–

Is your major sustainable enough? Whether you're pursuing a sustainability degree and want to further your knowledge, or are interested in supplementing your major in another area with sustainability education, plenty of independent learning resources are available. A wide range of credit and noncredit courses—including university- and or. . .

**Sample 24**
**Candidate #15**
Not selected
score=–

Origami is an art form that combines precision, creativity, and patience. While basic origami is obtainable to every one, mastering complex origami designs can be quite a rewarding and impressive achievement. In this article, we'll show you with the procedure for creating intricate origami while highlighting essential techniques for achie. . .

**Sample 25**
**Candidate #16**
Not selected
score=–

What is rotavirus and why does my baby need to be immunised? Rotavirus is a very infectious virus that causes the majority of serious cases of gastroenteritis in babies. It causes diarrhoea, vomiting and abdominal pain, usually lasting around a week. Most children will be infected by rotavirus once by the age of five. Gastroenteritis (cau. . .

**Sample 26**
**Candidate #19**
Not selected
score=–

Dividing Fractions Using Models Worksheet. This worksheet has six division with fractions issues to be solved — three must be solved with fashions and three with algorithms — options are on the second page. Answer key divide the unit fractions by whole numbers using th e fashions given. Use these resources to help reinforce the following. . .

**Sample 27**
**Candidate #20**
Not selected
score=–

Conduct Disorder (CD) is a complex and serious behavioural and emotional disorder that can occur in children and adolescents. It's characterised by a repetitive and persistent pattern of behaviour where the basic rights of others or major age-appropriate societal norms or rules are violated. Here's an outline of Conduct Disorder in line w. . .

**Sample 28**
**Candidate #22**
Not selected
score=–

How To Choose Decodable Readers for First Grade To decode or not to decode: really, there is no question. To help rising first graders become successful and enthusiastic readers this summer, decodable readers are essential reading resources. Although "decodable text" might sound like yet another form of educational lingo, parents and educ. . .

**Sample 29**
**Candidate #24**
Not selected
score=–

Next we will talk about solar radiation, that is, the forms of solar radiation that we receive on earth. Solar radiation is generated by a series of nuclear fusion reactions that occur in the Sun and, as a consequence, emit electromagnetic radiation that reaches the earth. This radiation received by the earth's surface is measured in W /. . .

**Sample 30**
**Candidate #25**
Not selected
score=–

KS2 Maths is an important core subject in the National Curriculum and this area of the website covers all the major aspects of the curriculum including numbers, calculations, problems and measures. Each subject area is designed to help children develop their knowledge, whether they are learning in a classroom or home schooling environment. . .

**Sample 31**
**Candidate #26**
Not selected
score=–

Unveiling the Power: Key Provisions of the Civil Rights Act of 1864 What were the Civil Rights Act of 1864's key provisions? The Civil Rights Act of 1864 was a pivotal moment in American history, establishing crucial legal protections for African Americans in the face of rampant discrimination. Editor Note: The Civil Rights Act of 1864 la. . .

**Sample 32**
**Candidate #28**
Not selected
score=–

You really have to be alert when studying science. Galaxies were created after matter. The stars in those galaxies were supposed to move slowly because there was more mass in the center of the galaxy. However, after dark matter was added, the stars appeared to move faster; however, this is not the case in our galaxy, suggesting that there. . .

## OPUS

**Sample 1**  
**Candidate #8**  
**Selected** score=0.00589

The Unsung Heroes of Your HVAC System: Understanding the Importance of Filters When it comes to your HVAC (Heating, Ventilation, and Air Conditioning) system, you might be quick to think about the thermostat, air ducts, or even the unit itself. However, there's an unsung hero in your HVAC system that plays a pivotal role in maintaining in. . .

**Sample 2**  
**Candidate #22**  
**Selected** score=0.00471

How To Choose Decodable Readers for First Grade To decode or not to decode: really, there is no question. To help rising first graders become successful and enthusiastic readers this summer, decodable readers are essential reading resources. Although "decodable text" might sound like yet another form of educational lingo, parents and educ. . .

**Sample 3**  
**Candidate #27**  
**Selected** score=0.00466

24/7 writing help on your phone Save to my list Remove from my list In the tumultuous 19th century, both Italy and Germany found themselves fragmented into numerous separate ruling states. The impetus for change came in the form of rising nationalism and liberalism, paving the way for the unification of these disparate entities. However,. . .

**Sample 4**  
**Candidate #4**  
**Selected** score=0.0046

5 Types of Women's Underwear That Men Love Underwear can say a lot about a woman. It's something that men are obsessed with, to the point that, a mere glimpse of a thong waistband causes us to go into shock. On the surface we find them sexy, revealing. We're able to see who a woman actually is—or maybe some guys are just plain horny. Howe. . .

**Sample 5**  
**Candidate #18**  
**Selected** score=0.0044

This article originally appeared in the December 2015 issue of Resource Recycling. Subscribe today for access to all print content. Since the 1990s, curbside and drop-off recycling has grown substantially – nearly 90 percent of households now have access, according to recent surveys from Moore Recycling Associates, the American Forest and. . .

**Sample 6**  
**Candidate #30**  
**Selected** score=0.0042

Over 1.8 million professionals use CFI to learn accounting, financial analysis, modeling and more. Start with a free account to explore 20+ always-free courses and hundreds of finance templates and cheat sheets. What is the Central Limit Theorem (CLT)? The Central Limit Theorem (CLT) is a statistical concept that states that the sample me. . .

**Sample 7**  
**Candidate #0**  
**Selected** score=0.0042

As it turns out, the exercises synonymous with strong, attractive abs may not be the best way to train your core—and may be doing damage to your back. Read more If you are worried about the excess holiday pounds many of us are still carrying around. There are a few easy, natural things you can do to shed them, and none of them requires an. . .

**Sample 8**  
**Candidate #23**  
**Selected** score=0.00418

The St. James kindergarteners have been working up to Project Week over the past month. We started slowly by taking walks in our neighborhood while Ms. Meghan and I noted what caught the children's interest. It became apparent that the class was very interested in the L trains that they saw on our walks. It started with a simple question,. . .

**Sample 9**  
**Candidate #31**  
**Selected** score=0.00411

One of the challenges of working with ancient DNA samples is that damage accumulates over time, breaking the double helix structure into ever-smaller fragments. In the samples we worked with, these fragments were scattered and mixed with contaminants, making genome reconstruction a major technical challenge. But a shocking paper published. . .

**Sample 10**  
**Candidate #11**  
**Selected** score=0.00401

In decades past, classroom design was often an afterthought and followed a standardised layout. Plain boxed shaped classrooms, with identical chairs and tables throughout were commonplace in many schools. Read the latest issue of School News HERE Recently, though, there has been a shift away from this one-size-fits all approach to classro. . .

**Sample 11**  
**Candidate #25**  
**Selected** score=0.00396

KS2 Maths is an important core subject in the National Curriculum and this area of the website covers all the major aspects of the curriculum including numbers, calculations, problems and measures. Each subject area is designed to help children develop their knowledge, whether they are learning in a classroom or home schooling environment. . .

**Sample 12**  
**Candidate #7**  
**Selected** score=0.0039

Elizabeth Hurley played as Dalila Release: Dec 8, 1996 Mara and her husband Manoa are both upstanding and religious Israelites living under the harsh and unjust rule of the Philistines. Much to their regret, they have not been able to have children. One day, a mysterious stranger appears to Mara and promises her that she will bear a son w. . .

**Sample 13**  
**Candidate #5**  
**Selected** score=0.00389

Well this is the big one. So big apparently, that I had to take it there and raise the number from 10 to 15. There's just that many fails in the world of female rap. Some slight missteps, some EPIC. Nevertheless, they are all worth mentioning. You can probably think of a bunch more, but this is what I have gathered picking up from my prev. . .

**Sample 14**  
**Candidate #9**  
**Selected** score=0.00384

"Last night three cargoes of Bohea Tea were emptied into the sea. This is the most magnificent movement of all. There is a dignity, a majesty, a sublimity, in this last effort of the Patriots that I greatly admire." - John Adams, diary entry, December 17, 1773 - John Adams, diary entry, December 17, 1773 A Novel Idea Is something so new a. . .

**Sample 15**  
**Candidate #19**  
**Selected** score=0.00376

Dividing Fractions Using Models Worksheet. This worksheet has six division with fractions issues to be solved — three must be solved with fashions and three with algorithms — options are on the second page. Answer key divide the unit fractions by whole numbers using th e fashions given. Use these resources to help reinforce the following. . .

**Sample 16**  
**Candidate #1**  
**Selected** score=0.00348

Wedding & Party Venues - Sort By: Edgartown : (508) 627-9510 A 19th century gothic revival home transformed into the island's premier eco-boutique hotel. Guests either stay in the 17-room Hob Knob hotel or in the privacy of their own Hob Knob House. Guests can expect individualized Hob Knob hospitality and modern luxury amenities in a rel. . .

**Sample 17**
**Candidate #15**
Not selected
score=0.00524

Origami is an art form that combines precision, creativity, and patience. While basic origami is obtainable to every one, mastering complex origami designs can be quite a rewarding and impressive achievement. In this article, we'll show you with the procedure for creating intricate origami while highlighting essential techniques for achie...

**Sample 18**
**Candidate #20**
Not selected
score=0.00518

Conduct Disorder (CD) is a complex and serious behavioural and emotional disorder that can occur in children and adolescents. It's characterised by a repetitive and persistent pattern of behaviour where the basic rights of others or major age-appropriate societal norms or rules are violated. Here's an outline of Conduct Disorder in line w...

**Sample 19**
**Candidate #14**
Not selected
score=0.00472

Is your major sustainable enough? Whether you're pursuing a sustainability degree and want to further your knowledge, or are interested in supplementing your major in another area with sustainability education, plenty of independent learning resources are available. A wide range of credit and noncredit courses—including university- and or...

**Sample 20**
**Candidate #28**
Not selected
score=0.0046

You really have to be alert when studying science. Galaxies were created after matter. The stars in those galaxies were supposed to move slowly because there was more mass in the center of the galaxy. However, after dark matter was added, the stars appeared to move faster; however, this is not the case in our galaxy, suggesting that there...

**Sample 21**
**Candidate #21**
Not selected
score=0.00457

Nestled in the leafy suburbs of western Berlin, the Wannsee Conference House stands as a poignant reminder of a dark chapter in human history. The Wannsee Conference: A Pivotal Moment The Wannsee Conference, held on January 20, 1942, marked a pivotal moment in the implementation of Nazi Germany's genocidal plans. Organized by SS-Obergrupp...

**Sample 22**
**Candidate #16**
Not selected
score=0.00456

What is rotavirus and why does my baby need to be immunised? Rotavirus is a very infectious virus that causes the majority of serious cases of gastroenteritis in babies. It causes diarrhoea, vomiting and abdominal pain, usually lasting around a week. Most children will be infected by rotavirus once by the age of five. Gastroenteritis (cau...

**Sample 23**
**Candidate #29**
Not selected
score=0.00448

Earthquakes are the result of sudden movement along faults within the Earth. The movement releases stored-up 'elastic strain' energy in the form of seismic waves, which propagate through the Earth and cause the ground surface to shake. Such movement on the faults is generally a response to long-term deformation and the buildup of stress...

**Sample 24**
**Candidate #13**
Not selected
score=0.00445

In Heart of Darkness it is the white invaders for instance, who are, almost without exception, embodiments of blindness, selfishness, and cruelty; and even in the cognitive domain, where such positive phrases as "to enlighten," for instance, are conventionally opposed to negative ones such as "to be in the dark," the traditional expectati...

**Sample 25**
**Candidate #26**
Not selected
score=0.00443

Unveiling the Power: Key Provisions of the Civil Rights Act of 1864 What were the Civil Rights Act of 1864's key provisions? The Civil Rights Act of 1864 was a pivotal moment in American history, establishing crucial legal protections for African Americans in the face of rampant discrimination. Editor Note: The Civil Rights Act of 1864 la...

**Sample 26**
**Candidate #24**
Not selected
score=0.00439

Next we will talk about solar radiation, that is, the forms of solar radiation that we receive on earth. Solar radiation is generated by a series of nuclear fusion reactions that occur in the Sun and, as a consequence, emit electromagnetic radiation that reaches the earth. This radiation received by the earth's surface is measured in W /...

**Sample 27**
**Candidate #17**
Not selected
score=0.00427

Political Parties and Elections Political parties are an established part of modern mass democracy, and the conduct of elections in India is largely dependent on the behaviour of political parties. Although many candidates for Indian elections are independent, the winning candidates for Lok Sabha and Vidhan Sabha elections usually stand a...

**Sample 28**
**Candidate #10**
Not selected
score=0.00427

Deforestation isn't just happening in well-known global hotspots like Indonesia and Brazil's rainforest. A new analysis says forests are also shrinking on state and private land in Oregon, where an estimated 522,000 acres of forest cover have disappeared since 2000. That's an area six times larger than the city of Portland, equal to more...

**Sample 29**
**Candidate #6**
Not selected
score=0.00401

Skaters need to check their skate helmets every so often and ask yourself, "Is it time to replace this helmet?" Well, that depends. Did you crash in it? For starters, most people are aware that you must replace a helmet after any crash where your head hit. The foam part of a helmet is made for one-time use, and after crushing once it is n...

**Sample 30**
**Candidate #2**
Not selected
score=0.00384

With the advent of new technologies for sneakers such as Vac Tech, Hyperfuse and Flyknit, the mid 90s and early 2000s methods of production and designing are becoming obsolete in this sneaker world. Nike Running is the future for Nike, generating billions of dollars per year, and we see Nike also not afraid to experiment with technology s...

**Sample 31**
**Candidate #12**
Not selected
score=0.00369

Can you please give us a little short bio? (education, professional experiences, select publications, academic specialty, awards won) Public school teacher for 5 years BA art (UC Irvine) PhD. (UCLA) educational psychology Professor of Child Development, (25 years) CSUS Senior Research Scientist (Oregon Research Institute with Institute of...

**Sample 32**
**Candidate #3**
Not selected
score=0.00333

starring John Travolta and Sam Jackson The first thing to understand about Basic –the basic thing, let's say– is that although the commercials make it look like a war movie, it is not, for which we can all be grateful. No, Basic is a plot-twisty whodunnit. If The Usual Suspects died, and its body turned to cheese, and then that cheese-b...

## High-PPL

**Sample 1**
**Candidate #3**
*Selected*
score=4.57

starring John Travolta and Sam Jackson The first thing to understand about Basic –the basic thing, let's say– is that although the commercials make it look like a war movie, it is not, for which we can all be grateful. No, Basic is a plot-twisty whodunnit. If The Usual Suspects died, and its body turned to cheese, and then that cheese-b...

**Sample 2**
**Candidate #19**
*Selected*
score=4.26

Dividing Fractions Using Models Worksheet. This worksheet has six division with fractions issues to be solved — three must be solved with fashions and three with algorithms — options are on the second page. Answer key divide the unit fractions by whole numbers using th e fashions given. Use these resources to help reinforce the following...

**Sample 3**
**Candidate #12**
*Selected*
score=4.26

Can you please give us a little short bio? (education, professional experiences, select publications, academic specialty, awards won) Public school teacher for 5 years BA art (UC Irvine) PhD. (UCLA) educational psychology Professor of Child Development, (25 years) CSUS Senior Research Scientist (Oregon Research Institute with Institute of...

**Sample 4**
**Candidate #5**
*Selected*
score=4.21

Well this is the big one. So big apparently, that I had to take it there and raise the number from 10 to 15. There's just that many fails in the world of female rap. Some slight missteps, some EPIC. Nevertheless, they are all worth mentioning. You can probably think of a bunch more, but this is what I have gathered picking up from my prev...

**Sample 5**
**Candidate #2**
*Selected*
score=4.04

With the advent of new technologies for sneakers such as Vac Tech, Hyperfuse and Flyknit, the mid 90s and early 2000s methods of production and designing are becoming obsolete in this sneaker world. Nike Running is the future for Nike, generating billions of dollars per year, and we see Nike also not afraid to experiment with technology s...

**Sample 6**
**Candidate #1**
*Selected*
score=3.89

Wedding & Party Venues - Sort By: Edgartown : (508) 627-9510 A 19th century gothic revival home transformed into the island's premier eco-boutique hotel. Guests either stay in the 17-room Hob Knob hotel or in the privacy of their own Hob Knob House. Guests can expect individualized Hob Knob hospitality and modern luxury amenities in a rel...

**Sample 7**
**Candidate #7**
*Selected*
score=3.89

Elizabeth Hurley played as Dalila Release: Dec 8, 1996 Mara and her husband Manoa are both upstanding and religious Israelites living under the harsh and unjust rule of the Philistines. Much to their regret, they have not been able to have children. One day, a mysterious stranger appears to Mara and promises her that she will bear a son w...

**Sample 8**
**Candidate #0**
*Selected*
score=3.82

As it turns out, the exercises synonymous with strong, attractive abs may not be the best way to train your core—and may be doing damage to your back. Read more If you are worried about the excess holiday pounds many of us are still carrying around. There are a few easy, natural things you can do to shed them, and none of them requires an...

**Sample 9**
**Candidate #13**
*Selected*
score=3.79

In Heart of Darkness it is the white invaders for instance, who are, almost without exception, embodiments of blindness, selfishness, and cruelty; and even in the cognitive domain, where such positive phrases as "to enlighten," for instance, are conventionally opposed to negative ones such as "to be in the dark," the traditional expectati...

**Sample 10**
**Candidate #6**
*Selected*
score=3.76

Skaters need to check their skate helmets every so often and ask yourself, "Is it time to replace this helmet?" Well, that depends. Did you crash in it? For starters, most people are aware that you must replace a helmet after any crash where your head hit. The foam part of a helmet is made for one-time use, and after crushing once it is n...

**Sample 11**
**Candidate #4**
*Selected*
score=3.74

5 Types of Women's Underwear That Men Love Underwear can say a lot about a woman. It's something that men are obsessed with, to the point that, a mere glimpse of a thong waistband causes us to go into shock. On the surface we find them sexy, revealing. We're able to see who a woman actually is—or maybe some guys are just plain horny. Howe...

**Sample 12**
**Candidate #9**
*Selected*
score=3.64

"Last night three cargoes of Bohea Tea were emptied into the sea. This is the most magnificent movement of all. There is a dignity, a majesty, a sublimity, in this last effort of the Patriots that I greatly admire." - John Adams, diary entry, December 17, 1773 - John Adams, diary entry, December 17, 1773 A Novel Idea Is something so new a...

**Sample 13**
**Candidate #23**
*Selected*
score=3.43

The St. James kindergarteners have been working up to Project Week over the past month. We started slowly by taking walks in our neighborhood while Ms. Meghan and I noted what caught the children's interest. It became apparent that the class was very interested in the L trains that they saw on our walks. It started with a simple question,...

**Sample 14**
**Candidate #31**
*Selected*
score=3.43

One of the challenges of working with ancient DNA samples is that damage accumulates over time, breaking the double helix structure into ever-smaller fragments. In the samples we worked with, these fragments were scattered and mixed with contaminants, making genome reconstruction a major technical challenge. But a shocking paper published...

**Sample 15**
**Candidate #11**
*Selected*
score=3.40

In decades past, classroom design was often an afterthought and followed a standardised layout. Plain boxed shaped classrooms, with identical chairs and tables throughout were commonplace in many schools. Read the latest issue of School News HERE Recently, though, there has been a shift away from this one-size-fits all approach to classro...

**Sample 16**
**Candidate #10**
*Selected*
score=3.38

Deforestation isn't just happening in well-known global hotspots like Indonesia and Brazil's rainforest. A new analysis says forests are also shrinking on state and private land in Oregon, where an estimated 522,000 acres of forest cover have disappeared since 2000. That's an area six times larger than the city of Portland, equal to more...

**Sample 17**
**Candidate #28**

Not selected
score=3.22

You really have to be alert when studying science. Galaxies were created after matter. The stars in those galaxies were supposed to move slowly because there was more mass in the center of the galaxy. However, after dark matter was added, the stars appeared to move faster; however, this is not the case in our galaxy, suggesting that there...

**Sample 18**
**Candidate #17**

Not selected
score=3.19

Political Parties and Elections Political parties are an established part of modern mass democracy, and the conduct of elections in India is largely dependent on the behaviour of political parties. Although many candidates for Indian elections are independent, the winning candidates for Lok Sabha and Vidhan Sabha elections usually stand a...

**Sample 19**
**Candidate #25**

Not selected
score=3.05

KS2 Maths is an important core subject in the National Curriculum and this area of the website covers all the major aspects of the curriculum including numbers, calculations, problems and measures. Each subject area is designed to help children develop their knowledge, whether they are learning in a classroom or home schooling environment...

**Sample 20**
**Candidate #27**

Not selected
score=3.05

24/7 writing help on your phone Save to my list Remove from my list In the tumultuous 19th century, both Italy and Germany found themselves fragmented into numerous separate ruling states. The impetus for change came in the form of rising nationalism and liberalism, paving the way for the unification of these disparate entities. However,...

**Sample 21**
**Candidate #18**

Not selected
score=3.03

This article originally appeared in the December 2015 issue of Resource Recycling. Subscribe today for access to all print content. Since the 1990s, curbside and drop-off recycling has grown substantially – nearly 90 percent of households now have access, according to recent surveys from Moore Recycling Associates, the American Forest and...

**Sample 22**
**Candidate #15**

Not selected
score=2.95

Origami is an art form that combines precision, creativity, and patience. While basic origami is obtainable to every one, mastering complex origami designs can be quite a rewarding and impressive achievement. In this article, we'll show you with the procedure for creating intricate origami while highlighting essential techniques for achie...

**Sample 23**
**Candidate #29**

Not selected
score=2.91

Earthquakes are the result of sudden movement along faults within the Earth. The movement releases stored-up 'elastic strain' energy in the form of seismic waves, which propagate through the Earth and cause the ground surface to shake. Such movement on the faults is generally a response to long-term deformation and the buildup of stress....

**Sample 24**
**Candidate #22**

Not selected
score=2.90

How To Choose Decodable Readers for First Grade To decode or not to decode: really, there is no question. To help rising first graders become successful and enthusiastic readers this summer, decodable readers are essential reading resources. Although "decodable text" might sound like yet another form of educational lingo, parents and educ...

**Sample 25**
**Candidate #24**

Not selected
score=2.83

Next we will talk about solar radiation, that is, the forms of solar radiation that we receive on earth. Solar radiation is generated by a series of nuclear fusion reactions that occur in the Sun and, as a consequence, emit electromagnetic radiation that reaches the earth. This radiation received by the earth's surface is measured in W /...

**Sample 26**
**Candidate #14**

Not selected
score=2.73

Is your major sustainable enough? Whether you're pursuing a sustainability degree and want to further your knowledge, or are interested in supplementing your major in another area with sustainability education, plenty of independent learning resources are available. A wide range of credit and noncredit courses—including university- and or...

**Sample 27**
**Candidate #21**

Not selected
score=2.61

Nestled in the leafy suburbs of western Berlin, the Wannsee Conference House stands as a poignant reminder of a dark chapter in human history. The Wannsee Conference: A Pivotal Moment The Wannsee Conference, held on January 20, 1942, marked a pivotal moment in the implementation of Nazi Germany's genocidal plans. Organized by SS-Obergrupp...

**Sample 28**
**Candidate #20**

Not selected
score=2.60

Conduct Disorder (CD) is a complex and serious behavioural and emotional disorder that can occur in children and adolescents. It's characterised by a repetitive and persistent pattern of behaviour where the basic rights of others or major age-appropriate societal norms or rules are violated. Here's an outline of Conduct Disorder in line w...

**Sample 29**
**Candidate #16**

Not selected
score=2.42

What is rotavirus and why does my baby need to be immunised? Rotavirus is a very infectious virus that causes the majority of serious cases of gastroenteritis in babies. It causes diarrhoea, vomiting and abdominal pain, usually lasting around a week. Most children will be infected by rotavirus once by the age of five. Gastroenteritis (cau...

**Sample 30**
**Candidate #26**

Not selected
score=2.31

Unveiling the Power: Key Provisions of the Civil Rights Act of 1864 What were the Civil Rights Act of 1864's key provisions? The Civil Rights Act of 1864 was a pivotal moment in American history, establishing crucial legal protections for African Americans in the face of rampant discrimination. Editor Note: The Civil Rights Act of 1864 la...

**Sample 31**
**Candidate #30**

Not selected
score=2.28

Over 1.8 million professionals use CFI to learn accounting, financial analysis, modeling and more. Start with a free account to explore 20+ always-free courses and hundreds of finance templates and cheat sheets. What is the Central Limit Theorem (CLT)? The Central Limit Theorem (CLT) is a statistical concept that states that the sample me...

**Sample 32**
**Candidate #8**

Not selected
score=1.58

The Unsung Heroes of Your HVAC System: Understanding the Importance of Filters When it comes to your HVAC (Heating, Ventilation, and Air Conditioning) system, you might be quick to think about the thermostat, air ducts, or even the unit itself. However, there's an unsung hero in your HVAC system that plays a pivotal role in maintaining in...

## GREATS

---

**Sample 1**
**Candidate #8** **Selected**
`score=17.40`

The Unsung Heroes of Your HVAC System: Understanding the Importance of Filters When it comes to your HVAC (Heating, Ventilation, and Air Conditioning) system, you might be quick to think about the thermostat, air ducts, or even the unit itself. However, there's an unsung hero in your HVAC system that plays a pivotal role in maintaining in...

---

**Sample 2**
**Candidate #15** **Selected**
`score=15.47`

Origami is an art form that combines precision, creativity, and patience. While basic origami is obtainable to every one, mastering complex origami designs can be quite a rewarding and impressive achievement. In this article, we'll show you with the procedure for creating intricate origami while highlighting essential techniques for achie...

---

**Sample 3**
**Candidate #20** **Selected**
`score=15.18`

Conduct Disorder (CD) is a complex and serious behavioural and emotional disorder that can occur in children and adolescents. It's characterised by a repetitive and persistent pattern of behaviour where the basic rights of others or major age-appropriate societal norms or rules are violated. Here's an outline of Conduct Disorder in line w...

---

**Sample 4**
**Candidate #14** **Selected**
`score=13.88`

Is your major sustainable enough? Whether you're pursuing a sustainability degree and want to further your knowledge, or are interested in supplementing your major in another area with sustainability education, plenty of independent learning resources are available. A wide range of credit and noncredit courses—including university- and or...

---

**Sample 5**
**Candidate #22** **Selected**
`score=13.87`

How To Choose Decodable Readers for First Grade To decode or not to decode: really, there is no question. To help rising first graders become successful and enthusiastic readers this summer, decodable readers are essential reading resources. Although "decodable text" might sound like yet another form of educational lingo, parents and educ...

---

**Sample 6**
**Candidate #27** **Selected**
`score=13.71`

24/7 writing help on your phone Save to my list Remove from my list In the tumultuous 19th century, both Italy and Germany found themselves fragmented into numerous separate ruling states. The impetus for change came in the form of rising nationalism and liberalism, paving the way for the unification of these disparate entities. However,...

---

**Sample 7**
**Candidate #4** **Selected**
`score=13.64`

5 Types of Women's Underwear That Men Love Underwear can say a lot about a woman. It's something that men are obsessed with, to the point that, a mere glimpse of a thong waistband causes us to go into shock. On the surface we find them sexy, revealing. We're able to see who a woman actually is—or maybe some guys are just plain horny. Howe...

---

**Sample 8**
**Candidate #28** **Selected**
`score=13.60`

You really have to be alert when studying science. Galaxies were created after matter. The stars in those galaxies were supposed to move slowly because there was more mass in the center of the galaxy. However, after dark matter was added, the stars appeared to move faster; however, this is not the case in our galaxy, suggesting that there...

---

**Sample 9**
**Candidate #21** **Selected**
`score=13.45`

Nestled in the leafy suburbs of western Berlin, the Wannsee Conference House stands as a poignant reminder of a dark chapter in human history. The Wannsee Conference: A Pivotal Moment The Wannsee Conference, held on January 20, 1942, marked a pivotal moment in the implementation of Nazi Germany's genocidal plans. Organized by SS-Obergrupp...

---

**Sample 10**
**Candidate #16** **Selected**
`score=13.43`

What is rotavirus and why does my baby need to be immunised? Rotavirus is a very infectious virus that causes the majority of serious cases of gastroenteritis in babies. It causes diarrhoea, vomiting and abdominal pain, usually lasting around a week. Most children will be infected by rotavirus once by the age of five. Gastroenteritis (cau...

---

**Sample 11**
**Candidate #29** **Selected**
`score=13.17`

Earthquakes are the result of sudden movement along faults within the Earth. The movement releases stored-up 'elastic strain' energy in the form of seismic waves, which propagate through the Earth and cause the ground surface to shake. Such movement on the faults is generally a response to long-term deformation and the buildup of stress....

---

**Sample 12**
**Candidate #13** **Selected**
`score=13.14`

In Heart of Darkness it is the white invaders for instance, who are, almost without exception, embodiments of blindness, selfishness, and cruelty; and even in the cognitive domain, where such positive phrases as "to enlighten," for instance, are conventionally opposed to negative ones such as "to be in the dark," the traditional expectati...

---

**Sample 13**
**Candidate #26** **Selected**
`score=13.01`

Unveiling the Power: Key Provisions of the Civil Rights Act of 1864 What were the Civil Rights Act of 1864's key provisions? The Civil Rights Act of 1864 was a pivotal moment in American history, establishing crucial legal protections for African Americans in the face of rampant discrimination. Editor Note: The Civil Rights Act of 1864 la...

---

**Sample 14**
**Candidate #18** **Selected**
`score=12.93`

This article originally appeared in the December 2015 issue of Resource Recycling. Subscribe today for access to all print content. Since the 1990s, curbside and drop-off recycling has grown substantially – nearly 90 percent of households now have access, according to recent surveys from Moore Recycling Associates, the American Forest and...

---

**Sample 15**
**Candidate #24** **Selected**
`score=12.92`

Next we will talk about solar radiation, that is, the forms of solar radiation that we receive on earth. Solar radiation is generated by a series of nuclear fusion reactions that occur in the Sun and, as a consequence, emit electromagnetic radiation that reaches the earth. This radiation received by the earth's surface is measured in W/...

---

**Sample 16**
**Candidate #17** **Selected**
`score=12.64`

Political Parties and Elections Political parties are an established part of modern mass democracy, and the conduct of elections in India is largely dependent on the behaviour of political parties. Although many candidates for Indian elections are independent, the winning candidates for Lok Sabha and Vidhan Sabha elections usually stand a...

---

**Sample 17**
**Candidate #10**
Not selected
score=12.53

Deforestation isn't just happening in well-known global hotspots like Indonesia and Brazil's rainforest. A new analysis says forests are also shrinking on state and private land in Oregon, where an estimated 522,000 acres of forest cover have disappeared since 2000. That's an area six times larger than the city of Portland, equal to more...

**Sample 18**
**Candidate #0**
Not selected
score=12.38

As it turns out, the exercises synonymous with strong, attractive abs may not be the best way to train your core—and may be doing damage to your back. Read more If you are worried about the excess holiday pounds many of us are still carrying around. There are a few easy, natural things you can do to shed them, and none of them requires an...

**Sample 19**
**Candidate #23**
Not selected
score=12.36

The St. James kindergarteners have been working up to Project Week over the past month. We started slowly by taking walks in our neighborhood while Ms. Meghan and I noted what caught the children's interest. It became apparent that the class was very interested in the L trains that they saw on our walks. It started with a simple question,...

**Sample 20**
**Candidate #30**
Not selected
score=12.31

Over 1.8 million professionals use CFI to learn accounting, financial analysis, modeling and more. Start with a free account to explore 20+ always-free courses and hundreds of finance templates and cheat sheets. What is the Central Limit Theorem (CLT)? The Central Limit Theorem (CLT) is a statistical concept that states that the sample me...

**Sample 21**
**Candidate #31**
Not selected
score=12.04

One of the challenges of working with ancient DNA samples is that damage accumulates over time, breaking the double helix structure into ever-smaller fragments. In the samples we worked with, these fragments were scattered and mixed with contaminants, making genome reconstruction a major technical challenge. But a shocking paper published...

**Sample 22**
**Candidate #11**
Not selected
score=11.85

In decades past, classroom design was often an afterthought and followed a standardised layout. Plain boxed shaped classrooms, with identical chairs and tables throughout were commonplace in many schools. Read the latest issue of School News HERE Recently, though, there has been a shift away from this one-size-fits all approach to classro...

**Sample 23**
**Candidate #6**
Not selected
score=11.82

Skaters need to check their skate helmets every so often and ask yourself, "Is it time to replace this helmet?" Well, that depends. Did you crash in it? For starters, most people are aware that you must replace a helmet after any crash where your head hit. The foam part of a helmet is made for one-time use, and after crushing once it is n...

**Sample 24**
**Candidate #25**
Not selected
score=11.51

KS2 Maths is an important core subject in the National Curriculum and this area of the website covers all the major aspects of the curriculum including numbers, calculations, problems and measures. Each subject area is designed to help children develop their knowledge, whether they are learning in a classroom or home schooling environment...

**Sample 25**
**Candidate #7**
Not selected
score=11.49

Elizabeth Hurley played as Dalila Release: Dec 8, 1996 Mara and her husband Manoa are both upstanding and religious Israelites living under the harsh and unjust rule of the Philistines. Much to their regret, they have not been able to have children. One day, a mysterious stranger appears to Mara and promises her that she will bear a son w...

**Sample 26**
**Candidate #5**
Not selected
score=11.49

Well this is the big one. So big apparently, that I had to take it there and raise the number from 10 to 15. There's just that many fails in the world of female rap. Some slight missteps, some EPIC. Nevertheless, they are all worth mentioning. You can probably think of a bunch more, but this is what I have gathered picking up from my prev...

**Sample 27**
**Candidate #9**
Not selected
score=11.29

"Last night three cargoes of Bohea Tea were emptied into the sea. This is the most magnificent movement of all. There is a dignity, a majesty, a sublimity, in this last effort of the Patriots that I greatly admire." - John Adams, diary entry, December 17, 1773 - John Adams, diary entry, December 17, 1773 A Novel Idea Is something so new a...

**Sample 28**
**Candidate #2**
Not selected
score=11.27

With the advent of new technologies for sneakers such as Vac Tech, Hyperfuse and Flyknit, the mid 90s and early 2000s methods of production and designing are becoming obsolete in this sneaker world. Nike Running is the future for Nike, generating billions of dollars per year, and we see Nike also not afraid to experiment with technology s...

**Sample 29**
**Candidate #19**
Not selected
score=11.04

Dividing Fractions Using Models Worksheet. This worksheet has six division with fractions issues to be solved — three must be solved with fashions and three with algorithms — options are on the second page. Answer key divide the unit fractions by whole numbers using th e fashions given. Use these resources to help reinforce the following...

**Sample 30**
**Candidate #12**
Not selected
score=10.94

Can you please give us a little short bio? (education, professional experiences, select publications, academic specialty, awards won) Public school teacher for 5 years BA art (UC Irvine) PhD. (UCLA) educational psychology Professor of Child Development, (25 years) CSUS Senior Research Scientist (Oregon Research Institute with Institute of...

**Sample 31**
**Candidate #1**
Not selected
score=10.23

Wedding & Party Venues - Sort By: Edgartown : (508) 627-9510 A 19th century gothic revival home transformed into the island's premier eco-boutique hotel. Guests either stay in the 17-room Hob Knob hotel or in the privacy of their own Hob Knob House. Guests can expect individualized Hob Knob hospitality and modern luxury amenities in a rel...

**Sample 32**
**Candidate #3**
Not selected
score=9.77

starring John Travolta and Sam Jackson The first thing to understand about Basic –the basic thing, let's say– is that although the commercials make it look like a war movie, it is not, for which we can all be grateful. No, Basic is a plot-twisty whodunnit. If The Usual Suspects died, and its body turned to cheese, and then that cheese-b...

## QuRating

**Sample 1**
**Candidate #26**
**Selected**
score=11.87

Unveiling the Power: Key Provisions of the Civil Rights Act of 1864 What were the Civil Rights Act of 1864's key provisions? The Civil Rights Act of 1864 was a pivotal moment in American history, establishing crucial legal protections for African Americans in the face of rampant discrimination. Editor Note: The Civil Rights Act of 1864 la...

**Sample 2**
**Candidate #14**
**Selected**
score=10.85

Is your major sustainable enough? Whether you're pursuing a sustainability degree and want to further your knowledge, or are interested in supplementing your major in another area with sustainability education, plenty of independent learning resources are available. A wide range of credit and noncredit courses—including university- and or...

**Sample 3**
**Candidate #22**
**Selected**
score=10.82

How To Choose Decodable Readers for First Grade To decode or not to decode: really, there is no question. To help rising first graders become successful and enthusiastic readers this summer, decodable readers are essential reading resources. Although "decodable text" might sound like yet another form of educational lingo, parents and educ...

**Sample 4**
**Candidate #31**
**Selected**
score=10.42

One of the challenges of working with ancient DNA samples is that damage accumulates over time, breaking the double helix structure into ever-smaller fragments. In the samples we worked with, these fragments were scattered and mixed with contaminants, making genome reconstruction a major technical challenge. But a shocking paper published...

**Sample 5**
**Candidate #23**
**Selected**
score=10.27

The St. James kindergarteners have been working up to Project Week over the past month. We started slowly by taking walks in our neighborhood while Ms. Meghan and I noted what caught the children's interest. It became apparent that the class was very interested in the L trains that they saw on our walks. It started with a simple question,...

**Sample 6**
**Candidate #12**
**Selected**
score=10.05

Can you please give us a little short bio? (education, professional experiences, select publications, academic specialty, awards won) Public school teacher for 5 years BA art (UC Irvine) PhD. (UCLA) educational psychology Professor of Child Development, (25 years) CSUS Senior Research Scientist (Oregon Research Institute with Institute of...

**Sample 7**
**Candidate #29**
**Selected**
score=9.76

Earthquakes are the result of sudden movement along faults within the Earth. The movement releases stored-up 'elastic strain' energy in the form of seismic waves, which propagate through the Earth and cause the ground surface to shake. Such movement on the faults is generally a response to long-term deformation and the buildup of stress....

**Sample 8**
**Candidate #20**
**Selected**
score=9.62

Conduct Disorder (CD) is a complex and serious behavioural and emotional disorder that can occur in children and adolescents. It's characterised by a repetitive and persistent pattern of behaviour where the basic rights of others or major age-appropriate societal norms or rules are violated. Here's an outline of Conduct Disorder in line w...

**Sample 9**
**Candidate #25**
**Selected**
score=8.96

KS2 Maths is an important core subject in the National Curriculum and this area of the website covers all the major aspects of the curriculum including numbers, calculations, problems and measures. Each subject area is designed to help children develop their knowledge, whether they are learning in a classroom or home schooling environment...

**Sample 10**
**Candidate #30**
**Selected**
score=8.95

Over 1.8 million professionals use CFI to learn accounting, financial analysis, modeling and more. Start with a free account to explore 20+ always-free courses and hundreds of finance templates and cheat sheets. What is the Central Limit Theorem (CLT)? The Central Limit Theorem (CLT) is a statistical concept that states that the sample me...

**Sample 11**
**Candidate #15**
**Selected**
score=8.39

Origami is an art form that combines precision, creativity, and patience. While basic origami is obtainable to every one, mastering complex origami designs can be quite a rewarding and impressive achievement. In this article, we'll show you with the procedure for creating intricate origami while highlighting essential techniques for achie...

**Sample 12**
**Candidate #28**
**Selected**
score=8.17

You really have to be alert when studying science. Galaxies were created after matter. The stars in those galaxies were supposed to move slowly because there was more mass in the center of the galaxy. However, after dark matter was added, the stars appeared to move faster; however, this is not the case in our galaxy, suggesting that there...

**Sample 13**
**Candidate #24**
**Selected**
score=8.12

Next we will talk about solar radiation, that is, the forms of solar radiation that we receive on earth. Solar radiation is generated by a series of nuclear fusion reactions that occur in the Sun and, as a consequence, emit electromagnetic radiation that reaches the earth. This radiation received by the earth's surface is measured in W /...

**Sample 14**
**Candidate #16**
**Selected**
score=8.04

What is rotavirus and why does my baby need to be immunised? Rotavirus is a very infectious virus that causes the majority of serious cases of gastroenteritis in babies. It causes diarrhoea, vomiting and abdominal pain, usually lasting around a week. Most children will be infected by rotavirus once by the age of five. Gastroenteritis (cau...

**Sample 15**
**Candidate #21**
**Selected**
score=8.02

Nestled in the leafy suburbs of western Berlin, the Wannsee Conference House stands as a poignant reminder of a dark chapter in human history. The Wannsee Conference: A Pivotal Moment The Wannsee Conference, held on January 20, 1942, marked a pivotal moment in the implementation of Nazi Germany's genocidal plans. Organized by SS-Obergrupp...

**Sample 16**
**Candidate #11**
**Selected**
score=7.76

In decades past, classroom design was often an afterthought and followed a standardised layout. Plain boxed shaped classrooms, with identical chairs and tables throughout were commonplace in many schools. Read the latest issue of School News HERE Recently, though, there has been a shift away from this one-size-fits all approach to classro...

**Sample 17**
**Candidate #18**
Not selected
score=7.47

This article originally appeared in the December 2015 issue of Resource Recycling. Subscribe today for access to all print content. Since the 1990s, curbside and drop-off recycling has grown substantially – nearly 90 percent of households now have access, according to recent surveys from Moore Recycling Associates, the American Forest and...

**Sample 18**
**Candidate #8**
Not selected
score=7.46

The Unsung Heroes of Your HVAC System: Understanding the Importance of Filters When it comes to your HVAC (Heating, Ventilation, and Air Conditioning) system, you might be quick to think about the thermostat, air ducts, or even the unit itself. However, there's an unsung hero in your HVAC system that plays a pivotal role in maintaining in...

**Sample 19**
**Candidate #10**
Not selected
score=7.21

Deforestation isn't just happening in well-known global hotspots like Indonesia and Brazil's rainforest. A new analysis says forests are also shrinking on state and private land in Oregon, where an estimated 522,000 acres of forest cover have disappeared since 2000. That's an area six times larger than the city of Portland, equal to more...

**Sample 20**
**Candidate #27**
Not selected
score=7.05

24/7 writing help on your phone Save to my list Remove from my list In the tumultuous 19th century, both Italy and Germany found themselves fragmented into numerous separate ruling states. The impetus for change came in the form of rising nationalism and liberalism, paving the way for the unification of these disparate entities. However,...

**Sample 21**
**Candidate #19**
Not selected
score=5.30

Dividing Fractions Using Models Worksheet. This worksheet has six division with fractions issues to be solved — three must be solved with fashions and three with algorithms — options are on the second page. Answer key divide the unit fractions by whole numbers using th e fashions given. Use these resources to help reinforce the following...

**Sample 22**
**Candidate #17**
Not selected
score=5.29

Political Parties and Elections Political parties are an established part of modern mass democracy, and the conduct of elections in India is largely dependent on the behaviour of political parties. Although many candidates for Indian elections are independent, the winning candidates for Lok Sabha and Vidhan Sabha elections usually stand a...

**Sample 23**
**Candidate #9**
Not selected
score=4.21

"Last night three cargoes of Bohea Tea were emptied into the sea. This is the most magnificent movement of all. There is a dignity, a majesty, a sublimity, in this last effort of the Patriots that I greatly admire." - John Adams, diary entry, December 17, 1773 - John Adams, diary entry, December 17, 1773 A Novel Idea Is something so new a...

**Sample 24**
**Candidate #0**
Not selected
score=3.94

As it turns out, the exercises synonymous with strong, attractive abs may not be the best way to train your core—and may be doing damage to your back. Read more If you are worried about the excess holiday pounds many of us are still carrying around. There are a few easy, natural things you can do to shed them, and none of them requires an...

**Sample 25**
**Candidate #6**
Not selected
score=2.82

Skaters need to check their skate helmets every so often and ask yourself, "Is it time to replace this helmet?" Well, that depends. Did you crash in it? For starters, most people are aware that you must replace a helmet after any crash where your head hit. The foam part of a helmet is made for one-time use, and after crushing once it is n...

**Sample 26**
**Candidate #13**
Not selected
score=2.34

In Heart of Darkness it is the white invaders for instance, who are, almost without exception, embodiments of blindness, selfishness, and cruelty; and even in the cognitive domain, where such positive phrases as "to enlighten," for instance, are conventionally opposed to negative ones such as "to be in the dark," the traditional expectati...

**Sample 27**
**Candidate #7**
Not selected
score=1.46

Elizabeth Hurley played as Dalila Release: Dec 8, 1996 Mara and her husband Manoa are both upstanding and religious Israelites living under the harsh and unjust rule of the Philistines. Much to their regret, they have not been able to have children. One day, a mysterious stranger appears to Mara and promises her that she will bear a son w...

**Sample 28**
**Candidate #2**
Not selected
score=0.321

With the advent of new technologies for sneakers such as Vac Tech, Hyperfuse and Flyknit, the mid 90s and early 2000s methods of production and designing are becoming obsolete in this sneaker world. Nike Running is the future for Nike, generating billions of dollars per year, and we see Nike also not afraid to experiment with technology s...

**Sample 29**
**Candidate #1**
Not selected
score=-0.429

Wedding & Party Venues - Sort By: Edgartown : (508) 627-9510 A 19th century gothic revival home transformed into the island's premier eco-boutique hotel. Guests either stay in the 17-room Hob Knob hotel or in the privacy of their own Hob Knob House. Guests can expect individualized Hob Knob hospitality and modern luxury amenities in a rel...

**Sample 30**
**Candidate #3**
Not selected
score=-2.09

starring John Travolta and Sam Jackson The first thing to understand about Basic –the basic thing, let's say– is that although the commercials make it look like a war movie, it is not, for which we can all be grateful. No, Basic is a plot-twisty whodunnit. If The Usual Suspects died, and its body turned to cheese, and then that cheese-b...

**Sample 31**
**Candidate #4**
Not selected
score=-2.84

5 Types of Women's Underwear That Men Love Underwear can say a lot about a woman. It's something that men are obsessed with, to the point that, a mere glimpse of a thong waistband causes us to go into shock. On the surface we find them sexy, revealing. We're able to see who a woman actually is—or maybe some guys are just plain horny. Howe...

**Sample 32**
**Candidate #5**
Not selected
score=-4.08

Well this is the big one. So big apparently, that I had to take it there and raise the number from 10 to 15. There's just that many fails in the world of female rap. Some slight missteps, some EPIC. Nevertheless, they are all worth mentioning. You can probably think of a bunch more, but this is what I have gathered picking up from my prev...

## FineWeb-Edu

**Sample 1**
**Candidate #28**
**Selected**
score=4.62

You really have to be alert when studying science. Galaxies were created after matter. The stars in those galaxies were supposed to move slowly because there was more mass in the center of the galaxy. However, after dark matter was added, the stars appeared to move faster; however, this is not the case in our galaxy, suggesting that there...

**Sample 2**
**Candidate #25**
**Selected**
score=4.61

KS2 Maths is an important core subject in the National Curriculum and this area of the website covers all the major aspects of the curriculum including numbers, calculations, problems and measures. Each subject area is designed to help children develop their knowledge, whether they are learning in a classroom or home schooling environment...

**Sample 3**
**Candidate #29**
**Selected**
score=4.61

Earthquakes are the result of sudden movement along faults within the Earth. The movement releases stored-up 'elastic strain' energy in the form of seismic waves, which propagate through the Earth and cause the ground surface to shake. Such movement on the faults is generally a response to long-term deformation and the buildup of stress....

**Sample 4**
**Candidate #31**
**Selected**
score=4.57

One of the challenges of working with ancient DNA samples is that damage accumulates over time, breaking the double helix structure into ever-smaller fragments. In the samples we worked with, these fragments were scattered and mixed with contaminants, making genome reconstruction a major technical challenge. But a shocking paper published...

**Sample 5**
**Candidate #24**
**Selected**
score=4.54

Next we will talk about solar radiation, that is, the forms of solar radiation that we receive on earth. Solar radiation is generated by a series of nuclear fusion reactions that occur in the Sun and, as a consequence, emit electromagnetic radiation that reaches the earth. This radiation received by the earth's surface is measured in W /...

**Sample 6**
**Candidate #26**
**Selected**
score=4.53

Unveiling the Power: Key Provisions of the Civil Rights Act of 1864 What were the Civil Rights Act of 1864's key provisions? The Civil Rights Act of 1864 was a pivotal moment in American history, establishing crucial legal protections for African Americans in the face of rampant discrimination. Editor Note: The Civil Rights Act of 1864 la...

**Sample 7**
**Candidate #27**
**Selected**
score=4.53

24/7 writing help on your phone Save to my list Remove from my list In the tumultuous 19th century, both Italy and Germany found themselves fragmented into numerous separate ruling states. The impetus for change came in the form of rising nationalism and liberalism, paving the way for the unification of these disparate entities. However,...

**Sample 8**
**Candidate #30**
**Selected**
score=4.50

Over 1.8 million professionals use CFI to learn accounting, financial analysis, modeling and more. Start with a free account to explore 20+ always-free courses and hundreds of finance templates and cheat sheets. What is the Central Limit Theorem (CLT)? The Central Limit Theorem (CLT) is a statistical concept that states that the sample me...

**Sample 9**
**Candidate #20**
**Selected**
score=4.18

Conduct Disorder (CD) is a complex and serious behavioural and emotional disorder that can occur in children and adolescents. It's characterised by a repetitive and persistent pattern of behaviour where the basic rights of others or major age-appropriate societal norms or rules are violated. Here's an outline of Conduct Disorder in line w...

**Sample 10**
**Candidate #19**
**Selected**
score=4.08

Dividing Fractions Using Models Worksheet. This worksheet has six division with fractions issues to be solved — three must be solved with fashions and three with algorithms — options are on the second page. Answer key divide the unit fractions by whole numbers using th e fashions given. Use these resources to help reinforce the following...

**Sample 11**
**Candidate #21**
**Selected**
score=3.96

Nestled in the leafy suburbs of western Berlin, the Wannsee Conference House stands as a poignant reminder of a dark chapter in human history. The Wannsee Conference: A Pivotal Moment The Wannsee Conference, held on January 20, 1942, marked a pivotal moment in the implementation of Nazi Germany's genocidal plans. Organized by SS-Obergrupp...

**Sample 12**
**Candidate #22**
**Selected**
score=3.92

How To Choose Decodable Readers for First Grade To decode or not to decode: really, there is no question. To help rising first graders become successful and enthusiastic readers this summer, decodable readers are essential reading resources. Although "decodable text" might sound like yet another form of educational lingo, parents and educ...

**Sample 13**
**Candidate #17**
**Selected**
score=3.85

Political Parties and Elections Political parties are an established part of modern mass democracy, and the conduct of elections in India is largely dependent on the behaviour of political parties. Although many candidates for Indian elections are independent, the winning candidates for Lok Sabha and Vidhan Sabha elections usually stand a...

**Sample 14**
**Candidate #23**
**Selected**
score=3.73

The St. James kindergarteners have been working up to Project Week over the past month. We started slowly by taking walks in our neighborhood while Ms. Meghan and I noted what caught the children's interest. It became apparent that the class was very interested in the L trains that they saw on our walks. It started with a simple question,...

**Sample 15**
**Candidate #16**
**Selected**
score=3.63

What is rotavirus and why does my baby need to be immunised? Rotavirus is a very infectious virus that causes the majority of serious cases of gastroenteritis in babies. It causes diarrhoea, vomiting and abdominal pain, usually lasting around a week. Most children will be infected by rotavirus once by the age of five. Gastroenteritis (cau...

**Sample 16**
**Candidate #18**
**Selected**
score=3.56

This article originally appeared in the December 2015 issue of Resource Recycling. Subscribe today for access to all print content. Since the 1990s, curbside and drop-off recycling has grown substantially – nearly 90 percent of households now have access, according to recent surveys from Moore Recycling Associates, the American Forest and...

**Sample 17**
**Candidate #10**
Not selected
score=3.30

Deforestation isn't just happening in well-known global hotspots like Indonesia and Brazil's rainforest. A new analysis says forests are also shrinking on state and private land in Oregon, where an estimated 522,000 acres of forest cover have disappeared since 2000. That's an area six times larger than the city of Portland, equal to more...

**Sample 18**
**Candidate #9**
Not selected
score=3.30

"Last night three cargoes of Bohea Tea were emptied into the sea. This is the most magnificent movement of all. There is a dignity, a majesty, a sublimity, in this last effort of the Patriots that I greatly admire." - John Adams, diary entry, December 17, 1773 - John Adams, diary entry, December 17, 1773 A Novel Idea Is something so new a...

**Sample 19**
**Candidate #11**
Not selected
score=2.95

In decades past, classroom design was often an afterthought and followed a standardised layout. Plain boxed shaped classrooms, with identical chairs and tables throughout were commonplace in many schools. Read the latest issue of School News HERE Recently, though, there has been a shift away from this one-size-fits all approach to classro...

**Sample 20**
**Candidate #15**
Not selected
score=2.93

Origami is an art form that combines precision, creativity, and patience. While basic origami is obtainable to every one, mastering complex origami designs can be quite a rewarding and impressive achievement. In this article, we'll show you with the procedure for creating intricate origami while highlighting essential techniques for achie...

**Sample 21**
**Candidate #13**
Not selected
score=2.86

In Heart of Darkness it is the white invaders for instance, who are, almost without exception, embodiments of blindness, selfishness, and cruelty; and even in the cognitive domain, where such positive phrases as "to enlighten," for instance, are conventionally opposed to negative ones such as "to be in the dark," the traditional expectati...

**Sample 22**
**Candidate #8**
Not selected
score=2.83

The Unsung Heroes of Your HVAC System: Understanding the Importance of Filters When it comes to your HVAC (Heating, Ventilation, and Air Conditioning) system, you might be quick to think about the thermostat, air ducts, or even the unit itself. However, there's an unsung hero in your HVAC system that plays a pivotal role in maintaining in...

**Sample 23**
**Candidate #12**
Not selected
score=2.72

Can you please give us a little short bio? (education, professional experiences, select publications, academic specialty, awards won) Public school teacher for 5 years BA art (UC Irvine) PhD. (UCLA) educational psychology Professor of Child Development, (25 years) CSUS Senior Research Scientist (Oregon Research Institute with Institute of...

**Sample 24**
**Candidate #14**
Not selected
score=2.68

Is your major sustainable enough? Whether you're pursuing a sustainability degree and want to further your knowledge, or are interested in supplementing your major in another area with sustainability education, plenty of independent learning resources are available. A wide range of credit and noncredit courses—including university- and or...

**Sample 25**
**Candidate #6**
Not selected
score=1.77

Skaters need to check their skate helmets every so often and ask yourself, "Is it time to replace this helmet?" Well, that depends. Did you crash in it? For starters, most people are aware that you must replace a helmet after any crash where your head hit. The foam part of a helmet is made for one-time use, and after crushing once it is n...

**Sample 26**
**Candidate #0**
Not selected
score=1.76

As it turns out, the exercises synonymous with strong, attractive abs may not be the best way to train your core—and may be doing damage to your back. Read more If you are worried about the excess holiday pounds many of us are still carrying around. There are a few easy, natural things you can do to shed them, and none of them requires an...

**Sample 27**
**Candidate #7**
Not selected
score=1.39

Elizabeth Hurley played as Dalila Release: Dec 8, 1996 Mara and her husband Manoa are both upstanding and religious Israelites living under the harsh and unjust rule of the Philistines. Much to their regret, they have not been able to have children. One day, a mysterious stranger appears to Mara and promises her that she will bear a son w...

**Sample 28**
**Candidate #5**
Not selected
score=0.957

Well this is the big one. So big apparently, that I had to take it there and raise the number from 10 to 15. There's just that many fails in the world of female rap. Some slight missteps, some EPIC. Nevertheless, they are all worth mentioning. You can probably think of a bunch more, but this is what I have gathered picking up from my prev...

**Sample 29**
**Candidate #3**
Not selected
score=0.919

starring John Travolta and Sam Jackson The first thing to understand about Basic –the basic thing, let's say– is that although the commercials make it look like a war movie, it is not, for which we can all be grateful. No, Basic is a plot-twisty whodunnit. If The Usual Suspects died, and its body turned to cheese, and then that cheese-b...

**Sample 30**
**Candidate #2**
Not selected
score=0.880

With the advent of new technologies for sneakers such as Vac Tech, Hyperfuse and Flyknit, the mid 90s and early 2000s methods of production and designing are becoming obsolete in this sneaker world. Nike Running is the future for Nike, generating billions of dollars per year, and we see Nike also not afraid to experiment with technology s...

**Sample 31**
**Candidate #1**
Not selected
score=0.798

Wedding & Party Venues - Sort By: Edgartown : (508) 627-9510 A 19th century gothic revival home transformed into the island's premier eco-boutique hotel. Guests either stay in the 17-room Hob Knob hotel or in the privacy of their own Hob Knob House. Guests can expect individualized Hob Knob hospitality and modern luxury amenities in a rel...

**Sample 32**
**Candidate #4**
Not selected
score=0.163

5 Types of Women's Underwear That Men Love Underwear can say a lot about a woman. It's something that men are obsessed with, to the point that, a mere glimpse of a thong waistband causes us to go into shock. On the surface we find them sexy, revealing. We're able to see who a woman actually is—or maybe some guys are just plain horny. Howe...

## Ultra-FineWeb

**Sample 1**
**Candidate #26** **Selected** `score=1.000`

Unveiling the Power: Key Provisions of the Civil Rights Act of 1864 What were the Civil Rights Act of 1864's key provisions? The Civil Rights Act of 1864 was a pivotal moment in American history, establishing crucial legal protections for African Americans in the face of rampant discrimination. Editor Note: The Civil Rights Act of 1864 la...

**Sample 2**
**Candidate #29** **Selected** `score=0.999`

Earthquakes are the result of sudden movement along faults within the Earth. The movement releases stored-up 'elastic strain' energy in the form of seismic waves, which propagate through the Earth and cause the ground surface to shake. Such movement on the faults is generally a response to long-term deformation and the buildup of stress....

**Sample 3**
**Candidate #20** **Selected** `score=0.998`

Conduct Disorder (CD) is a complex and serious behavioural and emotional disorder that can occur in children and adolescents. It's characterised by a repetitive and persistent pattern of behaviour where the basic rights of others or major age-appropriate societal norms or rules are violated. Here's an outline of Conduct Disorder in line w...

**Sample 4**
**Candidate #22** **Selected** `score=0.997`

How To Choose Decodable Readers for First Grade To decode or not to decode: really, there is no question. To help rising first graders become successful and enthusiastic readers this summer, decodable readers are essential reading resources. Although "decodable text" might sound like yet another form of educational lingo, parents and educ...

**Sample 5**
**Candidate #31** **Selected** `score=0.994`

One of the challenges of working with ancient DNA samples is that damage accumulates over time, breaking the double helix structure into ever-smaller fragments. In the samples we worked with, these fragments were scattered and mixed with contaminants, making genome reconstruction a major technical challenge. But a shocking paper published...

**Sample 6**
**Candidate #19** **Selected** `score=0.987`

Dividing Fractions Using Models Worksheet. This worksheet has six division with fractions issues to be solved — three must be solved with fashions and three with algorithms — options are on the second page. Answer key divide the unit fractions by whole numbers using th e fashions given. Use these resources to help reinforce the following...

**Sample 7**
**Candidate #30** **Selected** `score=0.978`

Over 1.8 million professionals use CFI to learn accounting, financial analysis, modeling and more. Start with a free account to explore 20+ always-free courses and hundreds of finance templates and cheat sheets. What is the Central Limit Theorem (CLT)? The Central Limit Theorem (CLT) is a statistical concept that states that the sample me...

**Sample 8**
**Candidate #24** **Selected** `score=0.971`

Next we will talk about solar radiation, that is, the forms of solar radiation that we receive on earth. Solar radiation is generated by a series of nuclear fusion reactions that occur in the Sun and, as a consequence, emit electromagnetic radiation that reaches the earth. This radiation received by the earth's surface is measured in W /...

**Sample 9**
**Candidate #28** **Selected** `score=0.964`

You really have to be alert when studying science. Galaxies were created after matter. The stars in those galaxies were supposed to move slowly because there was more mass in the center of the galaxy. However, after dark matter was added, the stars appeared to move faster; however, this is not the case in our galaxy, suggesting that there...

**Sample 10**
**Candidate #25** **Selected** `score=0.958`

KS2 Maths is an important core subject in the National Curriculum and this area of the website covers all the major aspects of the curriculum including numbers, calculations, problems and measures. Each subject area is designed to help children develop their knowledge, whether they are learning in a classroom or home schooling environment...

**Sample 11**
**Candidate #8** **Selected** `score=0.955`

The Unsung Heroes of Your HVAC System: Understanding the Importance of Filters When it comes to your HVAC (Heating, Ventilation, and Air Conditioning) system, you might be quick to think about the thermostat, air ducts, or even the unit itself. However, there's an unsung hero in your HVAC system that plays a pivotal role in maintaining in...

**Sample 12**
**Candidate #27** **Selected** `score=0.928`

24/7 writing help on your phone Save to my list Remove from my list In the tumultuous 19th century, both Italy and Germany found themselves fragmented into numerous separate ruling states. The impetus for change came in the form of rising nationalism and liberalism, paving the way for the unification of these disparate entities. However,...

**Sample 13**
**Candidate #15** **Selected** `score=0.928`

Origami is an art form that combines precision, creativity, and patience. While basic origami is obtainable to every one, mastering complex origami designs can be quite a rewarding and impressive achievement. In this article, we'll show you with the procedure for creating intricate origami while highlighting essential techniques for achie...

**Sample 14**
**Candidate #23** **Selected** `score=0.927`

The St. James kindergarteners have been working up to Project Week over the past month. We started slowly by taking walks in our neighborhood while Ms. Meghan and I noted what caught the children's interest. It became apparent that the class was very interested in the L trains that they saw on our walks. It started with a simple question,...

**Sample 15**
**Candidate #21** **Selected** `score=0.745`

Nestled in the leafy suburbs of western Berlin, the Wannsee Conference House stands as a poignant reminder of a dark chapter in human history. The Wannsee Conference: A Pivotal Moment The Wannsee Conference, held on January 20, 1942, marked a pivotal moment in the implementation of Nazi Germany's genocidal plans. Organized by SS-Obergrupp...

**Sample 16**
**Candidate #12** **Selected** `score=0.718`

Can you please give us a little short bio? (education, professional experiences, select publications, academic specialty, awards won) Public school teacher for 5 years BA art (UC Irvine) PhD. (UCLA) educational psychology Professor of Child Development, (25 years) CSUS Senior Research Scientist (Oregon Research Institute with Institute of...

**Sample 17**
**Candidate #14**
Not selected
score=0.695

Is your major sustainable enough? Whether you're pursuing a sustainability degree and want to further your knowledge, or are interested in supplementing your major in another area with sustainability education, plenty of independent learning resources are available. A wide range of credit and noncredit courses—including university- and or...

**Sample 18**
**Candidate #18**
Not selected
score=0.648

This article originally appeared in the December 2015 issue of Resource Recycling. Subscribe today for access to all print content. Since the 1990s, curbside and drop-off recycling has grown substantially – nearly 90 percent of households now have access, according to recent surveys from Moore Recycling Associates, the American Forest and...

**Sample 19**
**Candidate #11**
Not selected
score=0.547

In decades past, classroom design was often an afterthought and followed a standardised layout. Plain boxed shaped classrooms, with identical chairs and tables throughout were commonplace in many schools. Read the latest issue of School News HERE Recently, though, there has been a shift away from this one-size-fits all approach to classro...

**Sample 20**
**Candidate #13**
Not selected
score=0.532

In Heart of Darkness it is the white invaders for instance, who are, almost without exception, embodiments of blindness, selfishness, and cruelty; and even in the cognitive domain, where such positive phrases as "to enlighten," for instance, are conventionally opposed to negative ones such as "to be in the dark," the traditional expectati...

**Sample 21**
**Candidate #10**
Not selected
score=0.477

Deforestation isn't just happening in well-known global hotspots like Indonesia and Brazil's rainforest. A new analysis says forests are also shrinking on state and private land in Oregon, where an estimated 522,000 acres of forest cover have disappeared since 2000. That's an area six times larger than the city of Portland, equal to more...

**Sample 22**
**Candidate #17**
Not selected
score=0.470

Political Parties and Elections Political parties are an established part of modern mass democracy, and the conduct of elections in India is largely dependent on the behaviour of political parties. Although many candidates for Indian elections are independent, the winning candidates for Lok Sabha and Vidhan Sabha elections usually stand a...

**Sample 23**
**Candidate #9**
Not selected
score=0.224

"Last night three cargoes of Bohea Tea were emptied into the sea. This is the most magnificent movement of all. There is a dignity, a majesty, a sublimity, in this last effort of the Patriots that I greatly admire." - John Adams, diary entry, December 17, 1773 - John Adams, diary entry, December 17, 1773 A Novel Idea Is something so new a...

**Sample 24**
**Candidate #16**
Not selected
score=0.211

What is rotavirus and why does my baby need to be immunised? Rotavirus is a very infectious virus that causes the majority of serious cases of gastroenteritis in babies. It causes diarrhoea, vomiting and abdominal pain, usually lasting around a week. Most children will be infected by rotavirus once by the age of five. Gastroenteritis (cau...

**Sample 25**
**Candidate #7**
Not selected
score=0.095

Elizabeth Hurley played as Dalila Release: Dec 8, 1996 Mara and her husband Manoa are both upstanding and religious Israelites living under the harsh and unjust rule of the Philistines. Much to their regret, they have not been able to have children. One day, a mysterious stranger appears to Mara and promises her that she will bear a son w...

**Sample 26**
**Candidate #3**
Not selected
score=0.069

starring John Travolta and Sam Jackson The first thing to understand about Basic –the basic thing, let's say– is that although the commercials make it look like a war movie, it is not, for which we can all be grateful. No, Basic is a plot-twisty whodunnit. If The Usual Suspects died, and its body turned to cheese, and then that cheese-b...

**Sample 27**
**Candidate #2**
Not selected
score=0.058

With the advent of new technologies for sneakers such as Vac Tech, Hyperfuse and Flyknit, the mid 90s and early 2000s methods of production and designing are becoming obsolete in this sneaker world. Nike Running is the future for Nike, generating billions of dollars per year, and we see Nike also not afraid to experiment with technology s...

**Sample 28**
**Candidate #4**
Not selected
score=0.024

5 Types of Women's Underwear That Men Love Underwear can say a lot about a woman. It's something that men are obsessed with, to the point that, a mere glimpse of a thong waistband causes us to go into shock. On the surface we find them sexy, revealing. We're able to see who a woman actually is—or maybe some guys are just plain horny. Howe...

**Sample 29**
**Candidate #5**
Not selected
score=0.019

Well this is the big one. So big apparently, that I had to take it there and raise the number from 10 to 15. There's just that many fails in the world of female rap. Some slight missteps, some EPIC. Nevertheless, they are all worth mentioning. You can probably think of a bunch more, but this is what I have gathered picking up from my prev...

**Sample 30**
**Candidate #0**
Not selected
score=0.018

As it turns out, the exercises synonymous with strong, attractive abs may not be the best way to train your core—and may be doing damage to your back. Read more If you are worried about the excess holiday pounds many of us are still carrying around, There are a few easy, natural things you can do to shed them, and none of them requires an...

**Sample 31**
**Candidate #6**
Not selected
score=0.016

Skaters need to check their skate helmets every so often and ask yourself, "Is it time to replace this helmet?" Well, that depends. Did you crash in it? For starters, most people are aware that you must replace a helmet after any crash where your head hit. The foam part of a helmet is made for one-time use, and after crushing once it is n...

**Sample 32**
**Candidate #1**
Not selected
score=0.000579

Wedding & Party Venues - Sort By: Edgartown : (508) 627-9510 A 19th century gothic revival home transformed into the island's premier eco-boutique hotel. Guests either stay in the 17-room Hob Knob hotel or in the privacy of their own Hob Knob House. Guests can expect individualized Hob Knob hospitality and modern luxury amenities in a rel...

## DCLM-FastText

| Sample 1 | Selected |
|---|---|
| **Candidate #28** | score=0.902 |

You really have to be alert when studying science. Galaxies were created after matter. The stars in those galaxies were supposed to move slowly because there was more mass in the center of the galaxy. However, after dark matter was added, the stars appeared to move faster; however, this is not the case in our galaxy, suggesting that there...

| Sample 2 | Selected |
|---|---|
| **Candidate #29** | score=0.761 |

Earthquakes are the result of sudden movement along faults within the Earth. The movement releases stored-up 'elastic strain' energy in the form of seismic waves, which propagate through the Earth and cause the ground surface to shake. Such movement on the faults is generally a response to long-term deformation and the buildup of stress....

| Sample 3 | Selected |
|---|---|
| **Candidate #15** | score=0.632 |

Origami is an art form that combines precision, creativity, and patience. While basic origami is obtainable to every one, mastering complex origami designs can be quite a rewarding and impressive achievement. In this article, we'll show you with the procedure for creating intricate origami while highlighting essential techniques for achie...

| Sample 4 | Selected |
|---|---|
| **Candidate #26** | score=0.612 |

Unveiling the Power: Key Provisions of the Civil Rights Act of 1864 What were the Civil Rights Act of 1864's key provisions? The Civil Rights Act of 1864 was a pivotal moment in American history, establishing crucial legal protections for African Americans in the face of rampant discrimination. Editor Note: The Civil Rights Act of 1864 la...

| Sample 5 | Selected |
|---|---|
| **Candidate #31** | score=0.564 |

One of the challenges of working with ancient DNA samples is that damage accumulates over time, breaking the double helix structure into ever-smaller fragments. In the samples we worked with, these fragments were scattered and mixed with contaminants, making genome reconstruction a major technical challenge. But a shocking paper published...

| Sample 6 | Selected |
|---|---|
| **Candidate #27** | score=0.483 |

24/7 writing help on your phone Save to my list Remove from my list In the tumultuous 19th century, both Italy and Germany found themselves fragmented into numerous separate ruling states. The impetus for change came in the form of rising nationalism and liberalism, paving the way for the unification of these disparate entities. However,...

| Sample 7 | Selected |
|---|---|
| **Candidate #3** | score=0.367 |

starring John Travolta and Sam Jackson The first thing to understand about Basic –the basic thing, let's say– is that although the commercials make it look like a war movie, it is not, for which we can all be grateful. No, Basic is a plot-twisty whodunnit. If The Usual Suspects died, and its body turned to cheese, and then that cheese-b...

| Sample 8 | Selected |
|---|---|
| **Candidate #30** | score=0.294 |

Over 1.8 million professionals use CFI to learn accounting, financial analysis, modeling and more. Start with a free account to explore 20+ always-free courses and hundreds of finance templates and cheat sheets. What is the Central Limit Theorem (CLT)? The Central Limit Theorem (CLT) is a statistical concept that states that the sample me...

| Sample 9 | Selected |
|---|---|
| **Candidate #20** | score=0.242 |

Conduct Disorder (CD) is a complex and serious behavioural and emotional disorder that can occur in children and adolescents. It's characterised by a repetitive and persistent pattern of behaviour where the basic rights of others or major age-appropriate societal norms or rules are violated. Here's an outline of Conduct Disorder in line w...

| Sample 10 | Selected |
|---|---|
| **Candidate #16** | score=0.168 |

What is rotavirus and why does my baby need to be immunised? Rotavirus is a very infectious virus that causes the majority of serious cases of gastroenteritis in babies. It causes diarrhoea, vomiting and abdominal pain, usually lasting around a week. Most children will be infected by rotavirus once by the age of five. Gastroenteritis (cau...

| Sample 11 | Selected |
|---|---|
| **Candidate #21** | score=0.126 |

Nestled in the leafy suburbs of western Berlin, the Wannsee Conference House stands as a poignant reminder of a dark chapter in human history. The Wannsee Conference: A Pivotal Moment The Wannsee Conference, held on January 20, 1942, marked a pivotal moment in the implementation of Nazi Germany's genocidal plans. Organized by SS-Obergrupp...

| Sample 12 | Selected |
|---|---|
| **Candidate #24** | score=0.108 |

Next we will talk about solar radiation, that is, the forms of solar radiation that we receive on earth. Solar radiation is generated by a series of nuclear fusion reactions that occur in the Sun and, as a consequence, emit electromagnetic radiation that reaches the earth. This radiation received by the earth's surface is measured in W /...

| Sample 13 | Selected |
|---|---|
| **Candidate #8** | score=0.107 |

The Unsung Heroes of Your HVAC System: Understanding the Importance of Filters When it comes to your HVAC (Heating, Ventilation, and Air Conditioning) system, you might be quick to think about the thermostat, air ducts, or even the unit itself. However, there's an unsung hero in your HVAC system that plays a pivotal role in maintaining in...

| Sample 14 | Selected |
|---|---|
| **Candidate #17** | score=0.080 |

Political Parties and Elections Political parties are an established part of modern mass democracy, and the conduct of elections in India is largely dependent on the behaviour of political parties. Although many candidates for Indian elections are independent, the winning candidates for Lok Sabha and Vidhan Sabha elections usually stand a...

| Sample 15 | Selected |
|---|---|
| **Candidate #12** | score=0.067 |

Can you please give us a little short bio? (education, professional experiences, select publications, academic specialty, awards won) Public school teacher for 5 years BA art (UC Irvine) PhD. (UCLA) educational psychology Professor of Child Development, (25 years) CSUS Senior Research Scientist (Oregon Research Institute with Institute of...

| Sample 16 | Selected |
|---|---|
| **Candidate #7** | score=0.041 |

Elizabeth Hurley played as Dalila Release: Dec 8, 1996 Mara and her husband Manoa are both upstanding and religious Israelites living under the harsh and unjust rule of the Philistines. Much to their regret, they have not been able to have children. One day, a mysterious stranger appears to Mara and promises her that she will bear a son w...

**Sample 17**
**Candidate #9**
Not selected
score=0.030

"Last night three cargoes of Bohea Tea were emptied into the sea. This is the most magnificent movement of all. There is a dignity, a majesty, a sublimity, in this last effort of the Patriots that I greatly admire." - John Adams, diary entry, December 17, 1773 - John Adams, diary entry, December 17, 1773 A Novel Idea Is something so new a. . .

**Sample 18**
**Candidate #5**
Not selected
score=0.027

Well this is the big one. So big apparently, that I had to take it there and raise the number from 10 to 15. There's just that many fails in the world of female rap. Some slight missteps, some EPIC. Nevertheless, they are all worth mentioning. You can probably think of a bunch more, but this is what I have gathered picking up from my prev. . .

**Sample 19**
**Candidate #10**
Not selected
score=0.026

Deforestation isn't just happening in well-known global hotspots like Indonesia and Brazil's rainforest. A new analysis says forests are also shrinking on state and private land in Oregon, where an estimated 522,000 acres of forest cover have disappeared since 2000. That's an area six times larger than the city of Portland, equal to more. . .

**Sample 20**
**Candidate #4**
Not selected
score=0.024

5 Types of Women's Underwear That Men Love Underwear can say a lot about a woman. It's something that men are obsessed with, to the point that, a mere glimpse of a thong waistband causes us to go into shock. On the surface we find them sexy, revealing. We're able to see who a woman actually is—or maybe some guys are just plain horny. Howe. . .

**Sample 21**
**Candidate #6**
Not selected
score=0.019

Skaters need to check their skate helmets every so often and ask yourself, "Is it time to replace this helmet?" Well, that depends. Did you crash in it? For starters, most people are aware that you must replace a helmet after any crash where your head hit. The foam part of a helmet is made for one-time use, and after crushing once it is n. . .

**Sample 22**
**Candidate #23**
Not selected
score=0.012

The St. James kindergarteners have been working up to Project Week over the past month. We started slowly by taking walks in our neighborhood while Ms. Meghan and I noted what caught the children's interest. It became apparent that the class was very interested in the L trains that they saw on our walks. It started with a simple question,. . .

**Sample 23**
**Candidate #19**
Not selected
score=0.012

Dividing Fractions Using Models Worksheet. This worksheet has six division with fractions issues to be solved — three must be solved with fashions and three with algorithms — options are on the second page. Answer key divide the unit fractions by whole numbers using th e fashions given. Use these resources to help reinforce the following. . .

**Sample 24**
**Candidate #13**
Not selected
score=0.00832

In Heart of Darkness it is the white invaders for instance, who are, almost without exception, embodiments of blindness, selfishness, and cruelty; and even in the cognitive domain, where such positive phrases as "to enlighten," for instance, are conventionally opposed to negative ones such as "to be in the dark," the traditional expectati. . .

**Sample 25**
**Candidate #11**
Not selected
score=0.00455

In decades past, classroom design was often an afterthought and followed a standardised layout. Plain boxed shaped classrooms, with identical chairs and tables throughout were commonplace in many schools. Read the latest issue of School News HERE Recently, though, there has been a shift away from this one-size-fits all approach to classro. . .

**Sample 26**
**Candidate #22**
Not selected
score=0.00335

How To Choose Decodable Readers for First Grade To decode or not to decode: really, there is no question. To help rising first graders become successful and enthusiastic readers this summer, decodable readers are essential reading resources. Although "decodable text" might sound like yet another form of educational lingo, parents and educ. . .

**Sample 27**
**Candidate #2**
Not selected
score=0.00324

With the advent of new technologies for sneakers such as Vac Tech, Hyperfuse and Flyknit, the mid 90s and early 2000s methods of production and designing are becoming obsolete in this sneaker world. Nike Running is the future for Nike, generating billions of dollars per year, and we see Nike also not afraid to experiment with technology s. . .

**Sample 28**
**Candidate #18**
Not selected
score=0.00124

This article originally appeared in the December 2015 issue of Resource Recycling. Subscribe today for access to all print content. Since the 1990s, curbside and drop-off recycling has grown substantially – nearly 90 percent of households now have access, according to recent surveys from Moore Recycling Associates, the American Forest and. . .

**Sample 29**
**Candidate #0**
Not selected
score=0.00109

As it turns out, the exercises synonymous with strong, attractive abs may not be the best way to train your core—and may be doing damage to your back. Read more If you are worried about the excess holiday pounds many of us are still carrying around. There are a few easy, natural things you can do to shed them, and none of them requires an. . .

**Sample 30**
**Candidate #14**
Not selected
score=0.000783

Is your major sustainable enough? Whether you're pursuing a sustainability degree and want to further your knowledge, or are interested in supplementing your major in another area with sustainability education, plenty of independent learning resources are available. A wide range of credit and noncredit courses—including university- and or. . .

**Sample 31**
**Candidate #1**
Not selected
score=0.000569

Wedding & Party Venues - Sort By: Edgartown : (508) 627-9510 A 19th century gothic revival home transformed into the island's premier eco-boutique hotel. Guests either stay in the 17-room Hob Knob hotel or in the privacy of their own Hob Knob House. Guests can expect individualized Hob Knob hospitality and modern luxury amenities in a rel. . .

**Sample 32**
**Candidate #25**
Not selected
score=0.00016

KS2 Maths is an important core subject in the National Curriculum and this area of the website covers all the major aspects of the curriculum including numbers, calculations, problems and measures. Each subject area is designed to help children develop their knowledge, whether they are learning in a classroom or home schooling environment. . .

## DSIR

| | |
|---|---|
| **Sample 1** **Selected**
**Candidate #9** `score=11.70` | **Sample 2** **Selected**
**Candidate #26** `score=8.06` |

**Sample 1**  **Candidate #9**  Selected  score=11.70

"Last night three cargoes of Bohea Tea were emptied into the sea. This is the most magnificent movement of all. There is a dignity, a majesty, a sublimity, in this last effort of the Patriots that I greatly admire." - John Adams, diary entry, December 17, 1773 - John Adams, diary entry, December 17, 1773 A Novel Idea Is something so new a...

**Sample 2**  **Candidate #26**  Selected  score=8.06

Unveiling the Power: Key Provisions of the Civil Rights Act of 1864 What were the Civil Rights Act of 1864's key provisions? The Civil Rights Act of 1864 was a pivotal moment in American history, establishing crucial legal protections for African Americans in the face of rampant discrimination. Editor Note: The Civil Rights Act of 1864 la...

**Sample 3**  **Candidate #5**  Selected  score=4.71

Well this is the big one. So big apparently, that I had to take it there and raise the number from 10 to 15. There's just that many fails in the world of female rap. Some slight missteps, some EPIC. Nevertheless, they are all worth mentioning. You can probably think of a bunch more, but this is what I have gathered picking up from my prev...

**Sample 4**  **Candidate #4**  Selected  score=4.62

5 Types of Women's Underwear That Men Love Underwear can say a lot about a woman. It's something that men are obsessed with, to the point that, a mere glimpse of a thong waistband causes us to go into shock. On the surface we find them sexy, revealing. We're able to see who a woman actually is—or maybe some guys are just plain horny. Howe...

**Sample 5**  **Candidate #28**  Selected  score=3.89

You really have to be alert when studying science. Galaxies were created after matter. The stars in those galaxies were supposed to move slowly because there was more mass in the center of the galaxy. However, after dark matter was added, the stars appeared to move faster; however, this is not the case in our galaxy, suggesting that there...

**Sample 6**  **Candidate #0**  Selected  score=3.42

As it turns out, the exercises synonymous with strong, attractive abs may not be the best way to train your core—and may be doing damage to your back. Read more If you are worried about the excess holiday pounds many of us are still carrying around. There are a few easy, natural things you can do to shed them, and none of them requires an...

**Sample 7**  **Candidate #3**  Selected  score=3.39

starring John Travolta and Sam Jackson The first thing to understand about Basic –the basic thing, let's say– is that although the commercials make it look like a war movie, it is not, for which we can all be grateful. No, Basic is a plot-twisty whodunnit. If The Usual Suspects died, and its body turned to cheese, and then that cheese-b...

**Sample 8**  **Candidate #12**  Selected  score=3.30

Can you please give us a little short bio? (education, professional experiences, select publications, academic specialty, awards won) Public school teacher for 5 years BA art (UC Irvine) PhD. (UCLA) educational psychology Professor of Child Development, (25 years) CSUS Senior Research Scientist (Oregon Research Institute with Institute of...

**Sample 9**  **Candidate #2**  Selected  score=2.96

With the advent of new technologies for sneakers such as Vac Tech, Hyperfuse and Flyknit, the mid 90s and early 2000s methods of production and designing are becoming obsolete in this sneaker world. Nike Running is the future for Nike, generating billions of dollars per year, and we see Nike also not afraid to experiment with technology s...

**Sample 10**  **Candidate #18**  Selected  score=2.51

This article originally appeared in the December 2015 issue of Resource Recycling. Subscribe today for access to all print content. Since the 1990s, curbside and drop-off recycling has grown substantially – nearly 90 percent of households now have access, according to recent surveys from Moore Recycling Associates, the American Forest and...

**Sample 11**  **Candidate #15**  Selected  score=2.07

Origami is an art form that combines precision, creativity, and patience. While basic origami is obtainable to every one, mastering complex origami designs can be quite a rewarding and impressive achievement. In this article, we'll show you with the procedure for creating intricate origami while highlighting essential techniques for achie...

**Sample 12**  **Candidate #11**  Selected  score=2.06

In decades past, classroom design was often an afterthought and followed a standardised layout. Plain boxed shaped classrooms, with identical chairs and tables throughout were commonplace in many schools. Read the latest issue of School News HERE Recently, though, there has been a shift away from this one-size-fits all approach to classro...

**Sample 13**  **Candidate #31**  Selected  score=1.04

One of the challenges of working with ancient DNA samples is that damage accumulates over time, breaking the double helix structure into ever-smaller fragments. In the samples we worked with, these fragments were scattered and mixed with contaminants, making genome reconstruction a major technical challenge. But a shocking paper published...

**Sample 14**  **Candidate #24**  Selected  score=0.518

Next we will talk about solar radiation, that is, the forms of solar radiation that we receive on earth. Solar radiation is generated by a series of nuclear fusion reactions that occur in the Sun and, as a consequence, emit electromagnetic radiation that reaches the earth. This radiation received by the earth's surface is measured in W /...

**Sample 15**  **Candidate #22**  Selected  score=0.487

How To Choose Decodable Readers for First Grade To decode or not to decode: really, there is no question. To help rising first graders become successful and enthusiastic readers this summer, decodable readers are essential reading resources. Although "decodable text" might sound like yet another form of educational lingo, parents and educ...

**Sample 16**  **Candidate #10**  Selected  score=0.378

Deforestation isn't just happening in well-known global hotspots like Indonesia and Brazil's rainforest. A new analysis says forests are also shrinking on state and private land in Oregon, where an estimated 522,000 acres of forest cover have disappeared since 2000. That's an area six times larger than the city of Portland, equal to more...

**Sample 17**
**Candidate #30**
Not selected
`score=0.272`

Over 1.8 million professionals use CFI to learn accounting, financial analysis, modeling and more. Start with a free account to explore 20+ always-free courses and hundreds of finance templates and cheat sheets. What is the Central Limit Theorem (CLT)? The Central Limit Theorem (CLT) is a statistical concept that states that the sample me...

**Sample 18**
**Candidate #29**
Not selected
`score=-0.299`

Earthquakes are the result of sudden movement along faults within the Earth. The movement releases stored-up 'elastic strain' energy in the form of seismic waves, which propagate through the Earth and cause the ground surface to shake. Such movement on the faults is generally a response to long-term deformation and the buildup of stress....

**Sample 19**
**Candidate #6**
Not selected
`score=-0.307`

Skaters need to check their skate helmets every so often and ask yourself, "Is it time to replace this helmet?" Well, that depends. Did you crash in it? For starters, most people are aware that you must replace a helmet after any crash where your head hit. The foam part of a helmet is made for one-time use, and after crushing once it is n...

**Sample 20**
**Candidate #23**
Not selected
`score=-0.429`

The St. James kindergarteners have been working up to Project Week over the past month. We started slowly by taking walks in our neighborhood while Ms. Meghan and I noted what caught the children's interest. It became apparent that the class was very interested in the L trains that they saw on our walks. It started with a simple question,...

**Sample 21**
**Candidate #8**
Not selected
`score=-0.675`

The Unsung Heroes of Your HVAC System: Understanding the Importance of Filters When it comes to your HVAC (Heating, Ventilation, and Air Conditioning) system, you might be quick to think about the thermostat, air ducts, or even the unit itself. However, there's an unsung hero in your HVAC system that plays a pivotal role in maintaining in...

**Sample 22**
**Candidate #14**
Not selected
`score=-0.743`

Is your major sustainable enough? Whether you're pursuing a sustainability degree and want to further your knowledge, or are interested in supplementing your major in another area with sustainability education, plenty of independent learning resources are available. A wide range of credit and noncredit courses—including university- and or...

**Sample 23**
**Candidate #20**
Not selected
`score=-0.804`

Conduct Disorder (CD) is a complex and serious behavioural and emotional disorder that can occur in children and adolescents. It's characterised by a repetitive and persistent pattern of behaviour where the basic rights of others or major age-appropriate societal norms or rules are violated. Here's an outline of Conduct Disorder in line w...

**Sample 24**
**Candidate #1**
Not selected
`score=-0.923`

Wedding & Party Venues - Sort By: Edgartown : (508) 627-9510 A 19th century gothic revival home transformed into the island's premier eco-boutique hotel. Guests either stay in the 17-room Hob Knob hotel or in the privacy of their own Hob Knob House. Guests can expect individualized Hob Knob hospitality and modern luxury amenities in a rel...

**Sample 25**
**Candidate #7**
Not selected
`score=-0.971`

Elizabeth Hurley played as Dalila Release: Dec 8, 1996 Mara and her husband Manoa are both upstanding and religious Israelites living under the harsh and unjust rule of the Philistines. Much to their regret, they have not been able to have children. One day, a mysterious stranger appears to Mara and promises her that she will bear a son w...

**Sample 26**
**Candidate #13**
Not selected
`score=-1.18`

In Heart of Darkness it is the white invaders for instance, who are, almost without exception, embodiments of blindness, selfishness, and cruelty; and even in the cognitive domain, where such positive phrases as "to enlighten," for instance, are conventionally opposed to negative ones such as "to be in the dark," the traditional expectati...

**Sample 27**
**Candidate #25**
Not selected
`score=-1.26`

KS2 Maths is an important core subject in the National Curriculum and this area of the website covers all the major aspects of the curriculum including numbers, calculations, problems and measures. Each subject area is designed to help children develop their knowledge, whether they are learning in a classroom or home schooling environment...

**Sample 28**
**Candidate #21**
Not selected
`score=-1.61`

Nestled in the leafy suburbs of western Berlin, the Wannsee Conference House stands as a poignant reminder of a dark chapter in human history. The Wannsee Conference: A Pivotal Moment The Wannsee Conference, held on January 20, 1942, marked a pivotal moment in the implementation of Nazi Germany's genocidal plans. Organized by SS-Obergrupp...

**Sample 29**
**Candidate #16**
Not selected
`score=-1.98`

What is rotavirus and why does my baby need to be immunised? Rotavirus is a very infectious virus that causes the majority of serious cases of gastroenteritis in babies. It causes diarrhoea, vomiting and abdominal pain, usually lasting around a week. Most children will be infected by rotavirus once by the age of five. Gastroenteritis (cau...

**Sample 30**
**Candidate #17**
Not selected
`score=-2.02`

Political Parties and Elections Political parties are an established part of modern mass democracy, and the conduct of elections in India is largely dependent on the behaviour of political parties. Although many candidates for Indian elections are independent, the winning candidates for Lok Sabha and Vidhan Sabha elections usually stand a...

**Sample 31**
**Candidate #27**
Not selected
`score=-4.50`

24/7 writing help on your phone Save to my list Remove from my list In the tumultuous 19th century, both Italy and Germany found themselves fragmented into numerous separate ruling states. The impetus for change came in the form of rising nationalism and liberalism, paving the way for the unification of these disparate entities. However,...

**Sample 32**
**Candidate #19**
Not selected
`score=-8.76`

Dividing Fractions Using Models Worksheet. This worksheet has six division with fractions issues to be solved — three must be solved with fashions and three with algorithms — options are on the second page. Answer key divide the unit fractions by whole numbers using th e fashions given. Use these resources to help reinforce the following...

