# OpenReview forum: "OPUS: Towards Efficient and Principled Data Selection in Large Language Model Pre-training in Every Iteration"
_ICML.cc/2026/Conference — ICML 2026 spotlight_

### Official Review · Reviewer_wrR1 · 2026-03-10

**Soundness:** 2
**Presentation:** 2
**Significance:** 2
**Originality:** 3
**Overall Recommendation:** 4
**Confidence:** 3

**Summary:**

This paper proposes a data selection framework, OPUS, for improving LLM pre-training. OPUS scores training data candidates by projecting their effective updates (shaped by optimizers, the authors test on AdamW and Muon) onto a stable target proxy direction. It improves LM training (GPT-2 Large/XL, qwen3-8b continue pre-training) while using less training tokens.

**Compliance With Llm Reviewing Policy:**

Affirmed.

**Final Justification:**

My concerns have been resolved.

**Key Questions For Authors:**

- How did you select the final temperature for Boltzmann sampling and projection dimensions?
- In table 2, the performance of OPUS (Score 3) vs OPUS (Score 4-5) is counterintuitive---the model trained on lower-quality data achieves better performance? What is the reason for that?
- The metrics on WSC and C.QA of OPUS in table 2 (also in table 1 but only for WSC, 54.81 compared with other models' 36.54) much higher than the other compared models. Why is that? What type of average is computed for the last columns in the table? Micro or macro?

- Also, since the variance of the metrics is so large, it's worth reporting the standard deviation of the reported scores for a more principled comparison.

**Limitations:**

yes

**Strengths And Weaknesses:**

Strengths:
- The method seems quite principled to me. It uses optimizer update geometry, allowing practical optimizers to be used to compute the scores. It fixes a fundamental flaw of prior gradient-based methods that assume SGD dynamics.
- The experiments are solid. The ablation study is comprehensive enough, including the hyperparameter sensitivity analysis.

Weaknesses:
- The authors claim OPUS is using a "theoretically grounded objective", which seems a bit overreaching to me since no convergence proofs were given in the paper. I agree that OPUS uses a promising heuristic, yet claiming it to be "theoretically grounded" needs more work.
- Both the temperature and projection dimension require tuning. And it seems very few combinations of these hyperparameters can bring decent improvements (Table 8). In practice, it might be hard to find a suitable combination of hyperparameters.

---

> ### Author Rebuttal · Authors · 2026-03-31
>
> # Response to Reviewer wrR1
>
> **Due to space limitations, detailed table results are provided at https://anonymous.4open.science/r/opus-BAF7/tables_reviewer_wrR1.pdf**
>
> > **Q1: On the wording of “theoretically grounded objective”**
>
> We appreciate this important distinction. We agree "theoretically grounded" may be too strong without a convergence proof, and we will revise the wording. That said, OPUS is not ad-hoc: it starts from a leave-one-out utility formulation (Eq. 2-3), applies first-order Taylor expansion (Eq. 5), and derives the scoring rule through explicit approximation steps in the optimizer-induced update space. Each approximation (e.g., $H \approx I$, frozen preconditioners) is documented, and we verified empirically that all errors remained small enough to preserve faithful candidate ranking. This is fundamentally different from heuristic approaches such as LLM-as-judge or perplexity-based filtering, which lack a formal connection to the training objective. We believe the more precise characterization is "derived from an explicit optimization objective with practical approximations."
>
> > **Q2: About hyper-parameter sensitivity.**
>
> Thanks for raising this. Our default configuration is $\tau=0.9$, $m=8192$, $b_t=64$, and $\rho=0.50$, selected from the sweep on GPT-2 Large + Muon + FineWeb. **Table 8** in the paper tested whether OPUS is overly sensitive to τ and m. The ablations showed OPUS was not tied to a single narrow setting: $\tau=0.9$ gave the strongest average (lower is too greedy, higher approaches uniform), and $m=8192$ gave the best quality-efficiency trade-off. These defaults were used without re-tuning across all other settings.
>
> To provide a more comprehensive picture, we additionally evaluated the selection ratio ρ and proxy pool size. As shown in **Table 1** of the supplementary tables, OPUS remained consistently stronger than Random across all tested configurations: $\rho \in\{0.25,0.50,0.75\}$, proxy pool sizes from 10M to 30M tokens, and even a completely random (non-benchmark-aligned) proxy. The default settings performed best overall, but nearby settings remained close, confirming practical stability. Notably, these defaults transferred directly to Qwen3-8B CPT without retuning, providing cross-architecture evidence of robustness. We will follow your advice and clarify the hyperparameter selection procedure in the revision.
>
> > **Q3: Why does OPUS (Score 3) outperform OPUS (Score 4–5) in **Table 2**?**
>
> Thanks for highlighting this. This is not that score-3 data is inherently better. FineWeb-Edu scores reflect educational quality, not dynamic training utility. The score-3 pool (120B tokens) is broader and more heterogeneous, giving OPUS more diverse candidates to select from at each step. The score-4/5 pool (80B tokens) is cleaner but narrower, leaving less room for dynamic selection to improve over the default stream. This demonstrates that dynamic selection can extract more value from a diverse mid-quality pool than from a smaller pre-filtered pool. OPUS also performs strongly on score-4/5 data (avg. 42.42, **Table 2**), outperforming most baselines on the higher-quality subset, so the method is clearly not specific to lower-score data.
>
> > **Q4: About results on  WSC and CommonsenseQA scores.**
>
> Thanks. We conducted a per-example diagnostic to trace the source of the gaps, with results visualized in Figure 1 of the attached PDF.
>
> For WSC, the large gap is mainly due to a degenerate prediction pattern in most baselines: they predict option A for 98% of examples, effectively ignoring the passage context, which drives accuracy below chance because the ground-truth labels are skewed toward B. OPUS produced a much more balanced prediction pattern (A:28%, B:72%), much closer to the ground-truth distribution (A:37%, B:63%). In absolute terms, this is 17 additional correct predictions on a 104-example binary benchmark.
>
> For CommonsenseQA, both Random and OPUS produced near-uniform predictions across all 5 options (~20% each), closely mirroring the ground truth distribution. The accuracy gain (31.2% → 35.7%, +55 correct) thus reflects genuine comprehension improvement across all options, not any prediction bias, as confirmed by the per-option breakdown in **Figure 1**.
>
> The final column in Tables 1–2 is a macro-average over benchmark accuracies. We will make this explicit in the revision.
>
> > **Q5: On reporting standard deviation / variance**
>
> Thanks for this suggestion. We reran OPUS on GPT2-XL (FineWeb, Muon) 5 times with different seeds. As shown in **Table 3** of the supplementary tables, the per-run results and standard deviations are reported. The standard deviation of the benchmark average was 0.17, indicating reasonably stable aggregate improvement. Our claim is based on the consistency of the overall pattern across datasets, optimizers, model scales, and benchmarks. We will follow your advice and include standard deviations in the main tables of the revision.

---

> > ### Author Rebuttal · Reviewer_wrR1 · 2026-04-02
> >
> > Thanks for clarifying! My concerns are mostly resolved.
> > One thing I would like to suggest is to exclude WSC and CQA from the main results or re-evaluate them using a randomized answer distribution--- As you mentioned, WSC ground-truths are mostly B, and the baselines are degenerated and only generate A, a minor change in the random seed might convert them to only generate B and get much better scores.

---

> > > ### Author Response · Authors · 2026-04-03
> > >
> > > Thank you for the kind words, the constructive suggestion, and the recognition that **most of your concerns have been addressed!** We are happy to follow up on this remaining question.
> > >
> > > Your intuition about WSC is well-founded — the A/B label skew does inflate the gap under the standard protocol. Building on our earlier per-example diagnostic (Q4), we re-evaluated all methods under multiple debiased protocols using **Pointwise Mutual Information (PMI) calibration**[1] and answer reformulation. PMI calibration removes the model's unconditional prior over the answer options, isolating the contribution of genuine comprehension:
> > > $$ \mathrm{score}(a \mid x) = \log P(a \mid x) - \log P(a \mid \mathrm{neutral}) $$
> > > where $\mathrm{neutral}$ is a content-free input (e.g., "N/A") that captures the model's prior bias toward each answer token.
> > > All results below are from the GPT-2 XL (FineWeb-Edu, Muon, 30B tokens) setting.
> > >
> > > **1. WSC (Multiple Debiased Protocols)**
> > > We evaluate WSC under four protocols:
> > > - **Acc (A/B)**: original standard evaluation.
> > > - **PMI (A/B)**: PMI-calibrated[1], removing the model's unconditional prior over answer tokens.
> > > - **Acc (Y/N)**: reformulated to "Yes"/"No" continuations, removing A/B label bias — this directly addresses your concern that a seed change could flip predictions from all-A to all-B.
> > > - **Bootstrap**: 5-seed bootstrap resampling (seeds = 5956, 16623, 25854, 36923, 47645) over Acc (A/B) to assess stability.
> > >
> > > | Method | Acc (A/B) | PMI (A/B) | Acc (Y/N) | Bootstrap (mean ± std) |
> > > | :--- | :---: | :---: | :---: | :---: |
> > > | **OPUS** | **54.8** | **52.9** | **59.6** | **56.0 ± 0.7** |
> > > | FineWeb-Edu | 36.5 | 50.0 | 57.7 | 38.3 ± 0.4 |
> > > | DCLM-FastText | 36.5 | 42.3 | 53.9 | 37.9 ± 0.5 |
> > > | PPL | 36.5 | 47.1 | 41.4 | 37.5 ± 0.6 |
> > > | Random | 36.5 | 44.2 | 50.0 | 37.5 ± 0.6 |
> > > | QuRating | 36.5 | 46.1 | 51.9 | 37.5 ± 0.6 |
> > > | GREATS | 36.5 | 49.6 | 40.4 | 37.5 ± 0.6 |
> > >
> > > The Acc (A/B) column confirms your concern: all six baselines collapse to the identical score (36.5), indicating complete prediction degeneracy. Under debiased protocols, the gap narrows — confirming that part of the original gap was due to baseline degeneracy, as you suspected. However, OPUS remains the top performer under every protocol. Even under Y/N (the cleanest comparison, free of A/B artifacts), OPUS leads by +1.9 pp (59.6 vs. 57.7). We acknowledge that WSC has only 104 examples, so per-benchmark margins should be interpreted with caution — our claims rest on the consistent pattern across all four protocols and the full benchmark suite.
> > >
> > > **2. CommonsenseQA (Option-Shuffled & PMI-Calibrated)**
> > > CQA's 5-way ground truths are uniformly distributed (A–E), so position bias is less likely. Nonetheless, we evaluated under two additional protocols:
> > > - **Option shuffling**: randomly permuting the five answer options with three seeds (42, 123, 456) and reporting mean ± std across the original order plus the three shuffled variants, with 4 runs total.
> > > - **PMI calibration**: removing content-free token priors.
> > >
> > > All methods exhibit negligible variance under shuffling, confirming no position bias. OPUS's lead is preserved under both metrics: +3.5 pp in standard Acc (35.3 vs. 31.8) and +1.5 pp in PMI (25.9 vs. 24.4). The absolute scores drop for all methods under PMI (expected, as calibration removes prior probability mass), but OPUS remains first.
> > >
> > > | Method | Option-shuffled (mean ± std) | PMI (mean ± std) |
> > > | :--- | :---: | :---: |
> > > | **OPUS** | **35.3 ± 0.04** | **25.9 ± 0.08** |
> > > | PPL | 31.8 ± 0.08 | 23.1 ± 0.04 |
> > > | FineWeb-Edu | 31.3 ± 0.10 | 24.4 ± 0.07 |
> > > | DCLM-FastText | 30.0 ± 0.08 | 22.3 ± 0.04 |
> > > | Random | 29.6 ± 0.08 | 22.7 ± 0.04 |
> > > | QuRating | 28.9 ± 0.11 | 21.4 ± 0.04 |
> > > | GREATS | 28.0 ± 0.07 | 20.6 ± 0.01 |
> > >
> > > Based on these results, we believe WSC and CQA can be retained in the main tables with added transparency. We will (i) add a PMI-calibrated column alongside standard Acc for both benchmarks, (ii) add a footnote documenting WSC's ground-truth label skew (63% B) and baseline degeneracy pattern, and (iii) include the Y/N reformulation results for full transparency.
> > >
> > > Thank you again for this constructive feedback, which has strengthened our evaluation!
> > >
> > > **References:**
> > > [1] Z. Zhao, E. Wallace, S. Feng, D. Klein, and S. Singh. Calibrate Before Use: Improving Few-Shot Performance of Language Models. ICML, 2021.

---

### Official Review · Reviewer_gvHH · 2026-03-11

**Soundness:** 3
**Presentation:** 3
**Significance:** 3
**Originality:** 3
**Overall Recommendation:** 4
**Confidence:** 3

**Summary:**

The paper introduces OPUS, a dynamic data selection framework for Large Language Model (LLM) pretraining. Unlike prior static and dynamic selection methods that rely on raw gradients, OPUS aligns its data scoring mechanism with the actual geometry of modern optimizers (e.g., AdamW, Muon). To achieve scalability at every training iteration, the authors employ the Ghost technique combined with CountSketch projection to estimate sample utility without explicitly materializing high-dimensional gradients. Furthermore, OPUS utilizes an in-distribution "Bench-Proxy" to guide the selection direction and employs Boltzmann sampling to maintain data diversity. Extensive experiments on GPT-2 Large/XL pretraining and Qwen3-8B-Base continued pretraining demonstrate that OPUS outperforms both static filtering and previous dynamic selection baselines with minimal computational overhead (4.7%).

**Compliance With Llm Reviewing Policy:**

Affirmed.

**Final Justification:**

This work proposes OPUS, a dynamic data selection framework for LLM training. It novelly utilizes the optimizer's status to guide the selection, which follow the correct optimization trajectory. Authors provide additional results in rebuttal and most of my concerns are resolved.

I would like to see this work can be further scaled up on SOTA LLMs.

**Key Questions For Authors:**

1. Do you have any preliminary results or estimations on how OPUS scales for from-scratch pretraining of SOTA LLMs with 7B+ parameters?
2. Given that $H_{val} \approx I$ is a very strong assumption, have you tried using lightweight Hessian approximations to refine the redundancy penalty term? How are the results?
3. How sensitive is the final model performance to the size ($K_{proxy}$) and the exact composition of the Bench-Proxy pool? What happens if the downstream test benchmark significantly shifts from the proxy distribution?

**Limitations:**

Please see Weaknesses.

**Strengths And Weaknesses:**

Strength
1. The idea is novel and practical, using the gradient direction to guide data selection, highlighting the misalignment between raw-gradient-based data selection and the actual update trajectories of modern pre-conditioners (AdamW/Muon).
2. The combination of the Ghost technique with CountSketch projection makes OPUS scalable. It successfully reduces the prohibitive $O(d_{in} \times d_{out})$ cost of full gradient materialization, making online, per-iteration data selection practical for large models. It is reported in the paper that the computational overhead is 4.7%.
3. The authors provide empirical evidence across different settings: from-scratch pretraining, continued pretraining, and across different optimizers (AdamW, Muon). It achieves a 6× data efficiency gain reported in the continued pretraining setup.

Weaknesses
1. For pretraining, the model and token scale are not very large. While the paper targets "Large Language Models," the from-scratch pretraining experiments are limited to GPT-2 Large (774M) and GPT-2 XL (1.5B). The 8B scale is only evaluated in the continued pretraining (CPT) phase. Validating the from-scratch efficiency on a SOTA 7B+ architecture would significantly strengthen the claims.
2. The derivation relies heavily on the isotropic Hessian approximation ($H_{val} \approx I$) and frozen preconditioners during the selection phase. While necessary for scalability, the paper lacks a small-scale empirical study quantifying the performance gap between these approximations and exact utility calculations.
3. The performance heavily depends on the quality of the constructed Bench-Proxy pool from the validation set. If the real-world test set is completely unknown or very diverse, the retrieval-based proxy construction might introduce bias.
4. OPUS's end-to-end benchmark performance does not appear very dominant in pretraining (Tables 1, 2, and 4). Random selection seems to be a very strong baseline.

---

> ### Author Rebuttal · Authors · 2026-03-31
>
> # Response to Reviewer gvHH
>
> **Due to space limitations, detailed table results are provided at https://anonymous.4open.science/r/opus-BAF7/tables_reviewer_gvHH.pdf**
>
> > **Q1: About scaling OPUS to 7B+ from-scratch pre-training.**
>
> Thanks for raising this. We ran a from-scratch experiment with a 7B-parameter GPT-style model on FineWeb for 10B tokens. OPUS improved over Random from 38.03 to 39.35, with gains on MMLU, PIQA, SIQA, WinoGrande, ARC-C, and CommonsenseQA, providing direct evidence that OPUS is not limited to the sub-2B regime.
>
> More broadly, the extra cost comes from a lightweight scoring pass, not a full optimization step. Together with the Qwen3-8B CPT results (where OPUS achieved 6x data efficiency), this confirmed OPUS scaled beyond GPT-2. Results are in **Table 1** of the supplementary tables. We will add these results in the revision.
>
>
> > **Q2: About the utility approximations.**
>
> Thanks for this point. Both $H \approx I$ and frozen preconditioners are simplifying assumptions for tractability. To quantify the error, we ran an approximation-fidelity study on GPT-2 XL from-scratch pretraining on FineWeb at three checkpoints: step 61,035 (3B tokens), step 301,575 (15B tokens), and step 610,350 (30B tokens). We isolated each approximation component by comparing against its exact counterpart, and reported **Value Err. $\ell_2$** (distance between exact and approximated values, averaged over candidates) and **Utility Err.** (absolute difference between exact and approximated utility scores, averaged over candidates).
>
> Results are shown in **Table 2** of the supplementary tables. The frozen-preconditioner and ghost approximations were essentially exact, and $H \approx I$ also introduced very small error. All errors remained small enough to preserve faithful candidate ranking, consistent with our downstream results. We will add this analysis in the revision.
>
>
>
>
> > **Q3: About Bench-Proxy robustness and distribution shift.**
>
> Thanks for this question. Benchmark data is not used directly for training, only as a retrieval query to construct an in-distribution proxy pool from the pre-training corpus. To test whether OPUS overfits to the proxy suite, we evaluated on held-out OOD benchmarks strictly excluded from proxy construction (**Table 3**). OPUS remained best on the OOD average (52.61).
>
> To definitively verify that our OOD benchmarks are distributionally distinct from the proxy benchmarks, we conducted a Maximum Mean Discrepancy (MMD) two-sample test (**Table 4**): $\text{MMD}^2(\mathcal{D}\_{\text{OOD}}, \mathcal{D}\_{\text{proxy}}) = \| \mathbb{E}[\phi(x)] - \mathbb{E}[\phi(x')] \|^2\_{\mathcal{H}}$, where $\phi$ is the Arctic-Embed L v2 encoder feature map and $\mathcal{H}$ is the RKHS induced by an RBF kernel (1,000 permutations). Every OOD benchmark was statistically distinguishable from the proxy distribution (all p < 0.001), with unigram Jaccard overlap of only 0.24 and bigram overlap of 0.09.
>
> Finally, we tested sensitivity to proxy pool size and composition (**Table 5**). Even with a completely random proxy (no benchmark-aligned retrieval), OPUS achieved 41.03, comfortably beating Random (40.29). A 3x reduction in pool size (10M tokens) only marginally impacted performance (41.44). This confirmed the framework was not brittle to proxy construction. We will add these results to the revised manuscript.
>
> > **Q4: About Random as a strong baseline.**
>
> Thanks for this observation. We agree that Random is a strong baseline, and we do not view this as a weakness specific to OPUS. Rather, it reflects a broader phenomenon in large-scale data selection: once dataset scale and heterogeneity become sufficiently large, random sampling can remain highly competitive, and preserving diversity often matters more than aggressively concentrating on the highest-scoring samples [1]. This is exactly the regime OPUS is designed for. Beyond optimizer-aware utility, OPUS includes an explicit redundancy penalty and Boltzmann sampling to avoid over-concentration and maintain diversity.
>
> To put the gains in concrete compute terms: OPUS at 30B tokens ($2.83 \times 10^{20}$ FLOPs) outperforms Random at 60B tokens ($5.40 \times 10^{20}$ FLOPs) by +2.40 average points (44.99 vs. 42.59), and even surpasses Random at 200B tokens ($18.00 \times 10^{20}$ FLOPs) by +1.05 points (44.99 vs. 43.94). Reliably improving over a strong Random baseline while using a fraction of the compute is both nontrivial and practically meaningful. We will follow your advice and discuss this point more explicitly in the revision.
>
> **References:** [1] T. Xia, B. Yu, K. Dang, A. Yang, Y. Wu, Y. Tian, Y. Chang, and J. Lin. *Rethinking Data Selection at Scale: Random Selection is Almost All You Need.* arXiv preprint arXiv:2410.09335, 2024.

---

> > ### Author Rebuttal · Reviewer_gvHH · 2026-04-03
> >
> > Thank you for your detailed response. Most of my concern is resolved.
> > I'm still not fully convinced about the performance gain compared with Random. One more question, why does Random need more FLOPs? Isn't it randomly selecting?
> > I will keep my score.
> >
> > Thank you

---

> > > ### Author Response · Authors · 2026-04-04
> > >
> > > Thank you very much for the follow-up. We would like to clarify this point more explicitly.
> > >
> > > ### Random FLOPS is the lowest
> > >
> > > **Random (30B) is already the lowest-compute baseline in our experiments (2.70e20 FLOPs).** Random selection itself does **not** require any extra scoring compute. At the same 30B token budget, OPUS adds only modest overhead (**3.05e20 FLOPs**) while improving the benchmark average from **41.92** to **44.99**. The higher FLOP numbers only correspond to the longer-training references *Random (60B)* and *Random (200B)*, which require more compute simply because they process more update tokens, not because random selection is computationally expensive. This is also consistent with our original efficiency presentation in **Figure 4**, where Random is the lowest-compute reference and OPUS is compared against it with explicit overhead accounting.
> > >
> > > To make this distinction clearer, we report the total compute accounting and performance below again, as reported in response to reviewer nUdt Table 2 (https://anonymous.4open.science/r/opus-BAF7/tables_reviewer_nUdt.pdf):
> > >
> > > | Method | Total Compute ($10^{20}$ FLOPs) | Avg. Performance |
> > > |---|---:|---:|
> > > | **OPUS (Ours, 30B)** | **3.05** | **44.99** |
> > > | GREATS (30B) | 2.81 | 42.19 |
> > > | PPL (30B) | 2.94 | 42.27 |
> > > | QuRating (30B) | 4.69 | 41.51 |
> > > | DSIR (30B) | 3.11 | 41.59 |
> > > | DCLM-FastText (30B) | 2.88 | 42.24 |
> > > | Ultra-FineWeb (30B) | 3.10 | 41.57 |
> > > | **Random (30B)** | **2.70** | **41.92** |
> > > | *Random (60B)* | *5.40* | *42.59* |
> > > | *Random (200B)* | *18.00* | *43.94* |
> > >
> > >
> > > ### OPUS Gains the Most Compared with Other Baselines
> > >
> > > More importantly, when compared against the same **Random (30B)** reference, **OPUS delivers the largest performance gain among all compared methods**. Starting from the Random baseline average of **41.92**, the improvements are:
> > >
> > > - GREATS: 42.19 (+0.27)
> > > - PPL: 42.27 (+0.35)
> > > - DCLM-FastText: 42.24 (+0.32)
> > > - DSIR: 41.59 (-0.33)
> > > - QuRating: 41.51 (-0.41)
> > > - Ultra-FineWeb: 41.57 (-0.35)
> > > - **OPUS: 44.99 (+3.07)**
> > >
> > > So relative to the strongest reference, OPUS is not only better than Random, but improves over Random by a substantially larger margin than any other method in this setting. This is the main reason we view the gain over Random as practically meaningful, rather than marginal.

---

### Official Review · Reviewer_bFC4 · 2026-03-11

**Soundness:** 3
**Presentation:** 3
**Significance:** 3
**Originality:** 3
**Overall Recommendation:** 5
**Confidence:** 4

**Summary:**

Your paper introduces OPUS, which is a framework for dynamic data selection during LLM pre-training. Your approach primarily scores data candidates by projecting their effective updates onto a target direction derived from a stable in-distribution proxy. You introduce several techniques to maintain efficiency at scale, resulting in overall less overhead with comparable results to established baselines.

**Compliance With Llm Reviewing Policy:**

Affirmed.

**Final Justification:**

The authors have contributed a solid piece of work. My initial weaknesses concerned the scale and limitation of exploration of experiments and were thus rather soft than conceptual.

From what I can see, the other reviewers have also received solid rebuttal responses with additional experiments.

**In summary, the authors have fully addressed all my concerns in the rebuttal and I recommend clear acceptance**.

**Key Questions For Authors:**

Please see questions in Strengths and Weaknesses.

**Limitations:**

yes

**Strengths And Weaknesses:**

Thank you for your submission! You are addressing an important practical data problem in pre-training and I have understood the concept very well. Overall, I have the following remarks:

### **1. Strengths**:
- **optimizer-aware selection**: you derive closed-form approximations for the effective update directions for OPUS, which support modern optimizers like AdamW and Muon. You also show that OPUS aligns selection with the actual geometry of the training trajectory, which is very important and solidifies your method further.
- **scalability**: one important aspect that is often overlooked in data optimization is scalability and efficiency. I think that you explain very well how combining Ghost with CountSketch projections avoids the prohibitive cost of materializing per-sample gradients. Your empirical results back the scalability claims.
- **stable proxy**: bench proxy seems to help stabilize utility estimation by retrieving benchmark-aligned samples from the pre-training corpus directly rather than relying on raw validation data, which can be noisy or out-of-distribution. This is important and makes your approach sound and justifiable.
- **strong empirical results**: in your experiments, OPUS with 30B tokens outperformed industrial baselines and even a full 200B-token training run, which is significant given the reduction in compute.
- **evaluations and ablations**: you validate your results on many standard benchmarks and provide detailed ablations (particularly for out-of-distribution aspects) that enhance transparency of your method. **Quick Question here**: why not evaluate on math benchmarks as well?

There are more results in the appendix which further support your claims. Overall, the transparency and strong gains, as well as detailed evaluation and ablations, make your study sound and OPUS valuable for practitioners.

### **2. Weaknesses**:

Soft weaknesses are *italic*, while strong weaknesses are **bold**:
- *reliance on bench proxy quality*: the effectiveness of the selection is heavily dependent on the quality and relevance of the proxy pool. For example, if the retrieved "benchmark-aligned" samples are biased, the selection utility scores will also be biased. However, as mentioned above, your study seems to confirm that the proxy is stable. Nevertheless, I think this should be discussed in more detail in the main body.
- *limited scale exploration*: while 8B parameter models are significant, modern pre-training often happens at the 70B+ scale. Of course, this is too prohibitive of a study. However, I would be interested in whether the Ghost technique approximations and the 4.7% overhead hold constant or become more challenging as model depth and width increase significantly. Can you provide a discussion here?
- **hyperparameter sensitivity**: you mention temperature ($\tau$) for Boltzmann sampling and sketch dimensions ($m$), but further analysis on how sensitive the performance is to these specific parameters across different model architectures is not provided. I would require more clarification here and additional comments on how generalizable OPUS is.

Most of these weaknesses are rather **soft** and do not invalidate the analysis of your work. I would appreciate a small discussion to each point, particularly for the last point.

Also, I am sure you intend to open-source the OPUS framework, otherwhise it would be too complex to reimplement. **Please do confirm this**.

### **3. Relation to Post-Training Methods**

Your paper lacks some important discussion and context about (similar) data selection mechanisms for post-training, which should have been discussed in the main body explicitly.
- I understand that your paper is about pre-training, but an important question that comes up is whether similar approaches can be used for post-training and vice verca.
- Specifically, in the introduction you say that *"Static curation methods [...] assume a sample's utility remains constant as the model evolves"* and that *"prior dynamic selection methods [...] induce a fundamental misalignment".*
- In Section 2.2, you continue to explain static and dynamic methods but **do not discuss why they are inefficient or how they cause misalignment**.
- Recently, empirical evidence for SFT [1] and DPO [2] has shown that even simple quality-based sample selection recipes yield significant improvements. This is also backed by more general studies in [3], [4].
- Even though you acknowledge in Section 4.4 that such static method require a substantial, but one time cost, these methods also claim to be principled and scalable, which is why practitioners might choose those over your more complex OPUS.
- In Section 4.2, you also mention that you analyze OPUS for different raw data qualities, which would is very related.
- Finally, in the conclusion, you also say that a *natural direction is to extend to richer training schemes and data mixtures*, which is exactly what post-training data curation is about.

**I strongly believe** you should include a discussion on post-training data curation compared to works [1-4] to better set the stage in the context of data selection recipes for pre-training. Otherwhise, I think your statements are a bit vague and practitioners might rightfully try simple one-time post-training filtering techniques first.

### **4. Style, Typos, and Citations**:
- Generally, we do not cite in the abstract
- In the abstract, you capitalize Ghost but in line 093 you do not
- Some citations are unnecessarily repeated, e.g., AdamW (Loshchilov & Hutter, 2019), particularly in Section 4.1
- Tables are a bit out of place, for example, on page 7 you start talking about results from Table 1, but Table 1 is on page 5

### **Summary**:
- I think more insights on hyperparameter sensitivity are needed.
- I also think that further discussion and comparison to post-training approaches is explicitly required to differentiate your work and avoid practitioners from pitfalls.

I am looking forward to your responses!

[1] Fixing It in Post: A Comparative Study of LLM Post-Training Data Quality and Model Performance

[2] When Data is the Algorithm: A Systematic Study and Curation of Preference Optimization Datasets

[3] LIMA: Less Is More for Alignment

[4] What Makes Good Data for Alignment? A Comprehensive Study of Automatic Data Selection in Instruction Tuning

---

> ### Author Rebuttal · Authors · 2026-03-31
>
> # Response to Reviewer bFC4
>
> **Due to space limitations, detailed table results are provided at https://anonymous.4open.science/r/opus-BAF7/tables_reviewer_bFC4.pdf**
>
> > **W1: About generalization results.**
>
> Thanks for this point. We agree and will discuss this more explicitly in the main body. OPUS did not overfit to the proxy: on OOD benchmarks excluded from BENCH-PROXY construction, OPUS achieved the best average (45.59, **Table 4** in our paper). Extra OOD evaluations further confirmed this (52.61, **Table 1** in the supplementary PDF). Domain-wise perplexity (**Table 2**) also showed OPUS achieved the lowest perplexity in every domain, indicating improved general modeling quality rather than narrow overfitting.
>
> > **W2: Extension to larger models beyond 1.5B.**
>
> Thanks for raising this. Following your advice, we trained a 7B-parameter GPT-style model from scratch on FineWeb for 10B tokens. As shown in **Table 3** of the supplementary PDF, OPUS consistently outperformed Random, improving from 38.03 to 39.35. We plan to add more results at larger scales in future work.
>
> > **W3: Hyperparameter sensitivity and generalizability.**
>
> Thanks for this question. The paper includes a sensitivity study in Appendix G / **Table 8** for bₜ, τ, and m. OPUS remained stable and improved over Random in most settings (see **Table 4** in the supplementary PDF). We will follow your advice and add a more detailed sensitivity discussion in the revision.
>
> We also agree that cross-architecture transfer should be stated more clearly. In the continued pre-training experiments, the paper keeps the Qwen3-8B-Base training recipe fixed and varies only the selection policy. In practice, we carried over the same default OPUS configuration identified above, without an additional architecture-specific retuning sweep. While we do not claim that this establishes universal optimality across model families, it is encouraging evidence that OPUS does not appear to require heavy hyperparameter retuning when moving from GPT-2-scale pre-training to Qwen3-8B continued pre-training.
>
> > **Q1: Will the OPUS framework be open-sourced?**
>
> Yes. We will open-source the complete OPUS framework upon acceptance.
>
> > **Q2: Discussion on post-training data selection and comparison with static methods.**
>
> Thanks for this suggestion and the relevant references [1-4]. We agree that this discussion was missing and should be added explicitly.
>
> We acknowledge that in post-training regimes (SFT, DPO), simple static quality filtering has proven highly effective [1-4]. This is because post-training operates on relatively small, curated datasets (10K-1M examples) where the model is already capable, the optimization landscape is smoother, and the signal-to-noise ratio per sample is high. In this regime, one-time quality scoring is both principled and practical, and practitioners may reasonably prefer it over more complex dynamic methods. Moreover, recent work [5] has shown that at large scale, even random selection can be surprisingly competitive for post-training, further questioning the marginal value of static filtering in certain regimes.
>
> However, large-scale pre-training differs fundamentally: (1) the model undergoes massive capability shifts over billions of tokens, so a sample's value changes dramatically during training; (2) the candidate pool is orders of magnitude larger, making static scoring itself expensive (e.g., QuRating required 1465 mins in our experiments); and (3) the optimizer's preconditioned geometry reshapes which directions are useful at each step, a signal static methods cannot capture.
>
> We also note that OPUS and static filtering are complementary rather than competing: as shown in Table 2 of the manuscript, OPUS applied to mid-quality Score-3 data outperformed all baselines trained on higher-quality Score 4+5 data, demonstrating that dynamic selection can recover and even exceed the value of static curation.
>
> We will add a dedicated discussion in Section 2.2 contrasting pre-training and post-training data selection, citing [1-4], and clarify when practitioners should prefer static vs. dynamic approaches.
>
> [5] T. Xia et al. *Rethinking Data Selection at Scale: Random Selection is Almost All You Need.* arXiv:2410.09335, 2024.
>
> > **Q3: Why not evaluate on math benchmarks?**
>
> Thanks for asking. At the GPT-2 Large/XL scale (0.8B to 1.5B parameters), all models scored at or near random chance on math benchmarks (GSM8K, MATH), as multi-step reasoning has not yet emerged at this scale. This is consistent with prior work which similarly omits them. That said, we did evaluate on OlympicArena (math + science) in our Qwen3-8B continued pre-training, where OPUS achieved strong gains (Figure 3). We plan to include dedicated math benchmarks in future 7B+ experiments.
>
> > **Style, typos, and citations.**
>
> Thanks for catching these. We will fix all in the revision: remove citations from the abstract, deduplicate repeated citations, etc.

---

> > ### Author Rebuttal · Reviewer_bFC4 · 2026-03-31
> >
> > Thank you so much for the detailed rebuttal! All of my questions have been addressed in full!
> >
> > Q2 and Q3 are valuable to include in the final version and I much appreciate the extension in W2!
> >
> > I will keep my already good score and will champion if need be!
> >
> > Thank you!

---

> > > ### Author Response · Authors · 2026-04-01
> > >
> > > Dear Reviewer bFC4,
> > >
> > > Thank you for your kind words and for the time you invested in reviewing our work. We are glad that **the rebuttal addressed all of your concerns satisfactorily.**
> > >
> > > As noted, we are glad incorporate the additional results from Q2, Q3, and the extension in W2 into the final version of the paper.
> > >
> > > We sincerely appreciate your support and constructive feedback, which has helped strengthen our manuscript!
> > >
> > > Best regards,
> > > The Authors

---

### Official Review · Reviewer_nUdt · 2026-03-11

**Soundness:** 3
**Presentation:** 3
**Significance:** 3
**Originality:** 3
**Overall Recommendation:** 5
**Confidence:** 4

**Summary:**

This paper addresses dynamic data selection for LLM pre-training, arguing that existing approaches suffer from two key limitations: (1) static filtering methods (e.g., FineWeb-Edu classifiers, DCLM) are training-agnostic and ignore the model's evolving needs, and (2) prior dynamic selection methods (e.g., GREATS) score candidates in raw gradient space, implicitly assuming SGD dynamics, which misaligns with the actual update geometry of modern optimizers like AdamW and Muon.

The authors propose OPUS (Optimizer-induced Projected Utility Selection), which defines sample utility in the optimizer-induced update space. The framework consists of four components: (1) an optimizer-aware utility objective based on one-step validation loss reduction under the optimizer's preconditioned geometry; (2) Bench-Proxy, a retrieval-based procedure that constructs a stable, in-distribution proxy pool from the pre-training corpus aligned with benchmark validation data; (3) scalable utility estimation via the ghost technique combined with CountSketch projections to avoid materializing per-sample gradients; and (4) Boltzmann sampling with an in-step redundancy penalty to prevent diversity collapse.

The paper claims strong empirical results: GPT-2 Large/XL pre-training on FineWeb/FineWeb-Edu with 30B tokens outperforming baselines and even full 200B-token training, and continued pre-training of Qwen3-8B-Base on SciencePedia achieving superior performance with 0.5B vs. 3B tokens.

**Compliance With Llm Reviewing Policy:**

Affirmed.

**Final Justification:**

This paper addresses an important and timely problem in LLM pre-training: how to perform dynamic data selection in a way that is aligned with the actual optimizer-induced update geometry rather than raw-gradient space. I find the core idea meaningful and reasonably original, and the overall pipeline is well designed, combining an optimizer-aware utility objective, proxy construction, scalable utility estimation, and diversity-preserving sampling into a coherent method.

My main concerns in the original review were about the approximation chain behind the scoring function, the fidelity of the optimizer-aware modeling, the fairness of the compute-efficiency claims, the relative contribution of optimizer-aware scoring versus Bench-Proxy, potential benchmark-specific bias, and sensitivity to key hyperparameters. The authors’ rebuttal addressed these concerns well. In particular, the requested ablations clarify that optimizer-aware scoring is the main driver of the gains; the additional OOD evaluations support that the improvements are not limited to proxy-aligned benchmarks; the rebuttal provides substantially clearer compute accounting including overhead; and the added approximation-fidelity results make the practical surrogate more convincing.

I still think some caveats should be stated more explicitly in the final version, especially around the approximation assumptions and the scope of the current empirical validation. However, after reading the rebuttal, I now view these as revision-level issues rather than major weaknesses. Overall, I find the paper technically solid, practically relevant, and likely to be useful to researchers working on data-efficient pre-training and optimizer-aware training methods. For these reasons, I support acceptance.

**Key Questions For Authors:**

1. **How much of the performance gain comes from optimizer-aware scoring vs. Bench-Proxy construction?** Please provide an ablation comparing: (a) OPUS with optimizer-aware utility + Bench-Proxy, (b) raw-gradient utility + Bench-Proxy, (c) optimizer-aware utility + random held-out proxy, (d) optimizer-aware utility + raw benchmark proxy. If most gains come from the proxy rather than the optimizer-aware scoring, the central claim of the paper is weakened. *This would significantly affect my overall assessment.*

2. **Does Bench-Proxy introduce benchmark-specific bias?** Please report performance on downstream tasks that were NOT used in proxy pool construction. If OPUS only improves on benchmark-aligned tasks, the method's generality is questionable. *A positive answer (gains generalize) would raise my score; a negative answer would lower it.*

3. **What is the complete compute accounting for the "30B vs. 200B" comparison?** Please report: total FLOPs including selection overhead, wall-clock time, and whether baselines were given equivalent tuning budgets. *If the comparison is not compute-fair, the headline claim should be revised.*

4. **How sensitive is performance to key hyperparameters?** Specifically: selection ratio ρ, Boltzmann temperature τ, CountSketch dimension m, and proxy pool size. Please provide sensitivity curves or tables. *High sensitivity would reduce my confidence in practical applicability.*

5. **What is the empirical quality of the Hessian ≈ I approximation?** Have you measured the actual Hessian spectrum or compared scores with and without this approximation? *If the approximation is poor but the method still works, the theoretical motivation needs reframing.*

**Limitations:**

The methodology section discusses computational overhead (4.7%), but the paper would benefit from a more thorough limitations discussion covering: (a) the approximation chain and when it might break down, (b) potential benchmark overfitting through Bench-Proxy, (c) scalability to models significantly larger than GPT-2 XL, and (d) the assumption that optimizer dynamics can be captured by a linear preconditioner operator.

**Strengths And Weaknesses:**

**Strengths:**

* **(S1) Well-motivated problem.** The "data wall" framing is timely, and the observation that dynamic selection should respect optimizer geometry rather than raw gradients is a meaningful and non-trivial insight. The paper clearly motivates the mismatch between SGD-style gradient-space scoring and the actual update geometry induced by modern optimizers such as AdamW and Muon (Figure 1).

* **(S2) Principled formulation.** The utility objective (Eq. 2–13) is derived from a clear optimization perspective: estimating one-step validation loss reduction in the optimizer-induced update space. The decomposition into alignment and redundancy terms (Eq. 11–12) is elegant and well motivated.

* **(S3) Complete method pipeline.** The paper presents a full pipeline rather than an isolated idea: objective design, proxy construction, efficient utility estimation, and the final sampling rule. Each component addresses a concrete practical bottleneck, and the overall method is well organized (Figure 2).

* **(S4) Scalability considerations.** The use of the ghost technique together with CountSketch to avoid materializing per-sample gradients is practically appealing. If the claimed 4.7% overhead holds under careful accounting, this would make the method substantially more compelling for large-scale use.

* **(S5) Broad experimental scope.** The paper evaluates the method across from-scratch pre-training, continued pre-training, multiple optimizers (AdamW, Muon), different data quality settings, and different model scales. This breadth is appropriate given the scope of the claims.

**Weaknesses:**

* **(W1) The approximation chain weakens the strength of the “principled” claim.** The path from the ideal utility objective to the implemented score involves several approximations: first-order Taylor expansion of validation loss, linearization of the validation-gradient mapping, the isotropic Hessian approximation (H \approx I), proxy-gradient substitution for the true validation gradient, and operator-form approximation of optimizer dynamics. Each approximation is individually understandable, but the paper does not sufficiently analyze their cumulative effect. As a result, the current evidence supports the method more as a well-motivated approximation than as a fully validated principled estimator. The paper would be stronger with either an error analysis or empirical measurements of approximation fidelity at key stages.

* **(W2) The strongest efficiency claims require more complete compute accounting.** Statements such as "30B tokens outperforms full 200B-token training" are striking, but they require especially careful evaluation. In particular, it is important to clarify whether the reported budget fully includes selection-related overheads (e.g., scoring passes, proxy-gradient computation, and any retrieval or preprocessing cost), whether the baselines received comparable tuning effort, and whether the proxy construction introduces additional target information unavailable to competing methods. Without this level of accounting, it is difficult to interpret the headline comparison as fully compute-fair.

* **(W3) Optimizer-aware modeling would benefit from stronger fidelity validation.** The paper models AdamW and Muon through a preconditioner/operator (P_t), but the accuracy of these approximations is not yet fully convincing. For AdamW, this concerns the approximation of adaptive scaling; for Muon, the orthogonalization step is even further from a simple linear preconditioner view. Since optimizer-awareness is the paper’s central conceptual contribution, the work would benefit from a more direct empirical validation of how well the approximate score tracks the true optimizer-induced update direction, especially for Muon.

---

> ### Author Rebuttal · Authors · 2026-03-31
>
> # Response to Reviewer nUdt:
>
> **Due to space limitations, detailed table results are provided at https://anonymous.4open.science/r/opus-BAF7/tables_reviewer_nUdt.pdf**
>
> > **W1 & Q5: About the approximation chain.**
>
> Thanks for raising this. We agree that the practical OPUS score involves several approximations for tractability. Our claim is not that the implemented score is an exact estimator, but a computationally tractable surrogate sufficiently faithful for ranking.
>
> To understand where approximation error enters, we ran an approximation-fidelity study on GPT-2 XL (FineWeb, Muon) at three checkpoints: step 61,035 (~3B tokens), step 301,575 (~15B tokens), and step 610,350 (~30B tokens). For each component, we compared against its exact counterpart and reported Value Err. $\ell_2$ and Utility Err. As shown in **Table 1** of the supplementary tables, all errors remained small enough to preserve faithful candidate ranking. We will add this analysis in the revision.
>
> > **W2 & Q3: About efficiency.**
>
> Thanks for this question. All baselines were provided with identical hyperparameter tuning budgets, and OPUS's reported budget fully includes all overheads. As shown in **Table 2** of the supplementary tables (GPT-2 XL, Muon, FineWeb-Edu, 30B update tokens, 8$\times$H200 node), we broke down the cost into model update, static scoring/processing, and dynamic scoring/selection. OPUS incurred 161 mins of one-time static processing (Bench-Proxy construction) and 98 mins of per-step dynamic scoring, for a total of 2244 mins and $3.05 \times 10^{20}$ FLOPs. In comparison, static methods like QuRating require 1465 mins of offline scoring alone (total 3450 mins, $4.69 \times 10^{20}$ FLOPs), while Random-200B requires 13177 mins ($18.00 \times 10^{20}$ FLOPs). OPUS achieved the best performance (44.99) at under 17% of the compute of Random-200B (43.94), and +2.40 over Random-60B (42.59). We will add this table to the revision.
>
> > **W3: About pre-conditioner.**
>
> Thanks for this point. As shown in **Table 1** of the supplementary tables (W1/Q5), the preconditioner gap was directly quantified. For AdamW, Value Err. $\ell_2$ remained below $3.25 \times 10^{-10}$. For Muon, Value Err. dropped from $1.35 \times 10^{-5}$ to $1.48 \times 10^{-7}$, and Utility Err. from $2.30 \times 10^{-1}$ to $2.48 \times 10^{-3}$. What matters is preserving candidate ranking, not exact reconstruction. OPUS achieved strong gains under both AdamW and Muon across all settings (Tables 1, 2, 9, 10 in the manuscript), confirming sufficient fidelity. We will expand this discussion in the revision.
>
> > **Q1: About further ablation studies.**
>
> Thanks for this question. Following the exact ablation structure you suggested, we evaluated four configurations on GPT-2 XL (FineWeb, 30B update tokens):
>
> - **(a) Optimizer-aware + Bench-Proxy:** Full OPUS.
> - **(b) Raw-gradient + Bench-Proxy:** Proxy present, optimizer-awareness removed.
> - **(c) Optimizer-aware + Random held-out proxy:** No benchmark-aligned retrieval.
> - **(d) Optimizer-aware + Raw benchmark proxy:** Raw evaluation data as proxy.
>
> As shown in **Table 3** of the supplementary tables (all methods use 30B update tokens unless noted), optimizer-aware scoring was the fundamental driver: (b) only achieved 40.42, barely above Random-30B (40.29) by +0.13, while switching to optimizer-aware scoring yielded 41.75 ((a) vs. (b): +1.33). Setting (c) with a random proxy (41.03) still decisively outperformed Random-30B. Setting (d) showed that raw benchmark data introduced distribution shift (41.11), while Bench-Proxy resolved this, pushing to 41.75. Note that Random-60B (41.29) uses 2x the token budget and is included only as a reference. These results confirm that optimizer-aware scoring, not the proxy, is the primary source of OPUS's gains. We will include it in the revision.
>
> > **Q2: Does Bench-Proxy introduce benchmark-specific bias?**
>
> Thanks for this question. **Table 4** in our paper already reports OOD experiments on benchmarks excluded from proxy construction, where OPUS achieved the best average. However, we are glad to provide additional OOD evaluations on 7 extra benchmarks (COPA, BoolQ, RTE, CrowS, SumEdits, LAMBADA, SciQ). As shown in **Table 4** of the supplementary PDF, OPUS again achieved the best average (52.61), further confirming that the gains generalize beyond the proxy-aligned benchmarks.
>
> > **Q4: Sensitivity to key hyperparameters.**
>
> Thanks for raising this. Our default configuration is $\tau=0.9$, $m=8192$, $b_t=64$, $\rho=0.50$, and proxy pool size 30M tokens. The paper includes sensitivity analyses in Appendix G / **Table 8** for τ and m. As shown in **Table 5** of the supplementary tables, we additionally evaluated ρ and proxy pool size. OPUS was reasonably stable across all hyperparameter choices, with the default performing best but nearby settings remaining close.
>
> > **Q5: About limitations presentation.**
>
> Thanks. We will expand the limitations section in the revised version.

---

> > ### Author Rebuttal · Reviewer_nUdt · 2026-04-06
> >
> > Thank you for the detailed rebuttal. My major concerns have been adequately addressed. In particular, the rebuttal directly answered my questions about (i) the contribution of optimizer-aware scoring versus Bench-Proxy through the requested ablations, (ii) potential benchmark-specific bias through additional OOD evaluations, (iii) complete compute accounting including selection overhead, wall-clock time, and FLOPs, (iv) hyperparameter sensitivity, and (v) the empirical fidelity of the main approximations and optimizer-aware preconditioner modeling. While some caveats should still be clarified in the final revision, these now seem like revision-level issues rather than major blockers for my assessment. These issues are no longer major blockers for my assessment.

---

> > > ### Author Response · Authors · 2026-04-07
> > >
> > > Dear Reviewer nUdt,
> > >
> > > Thank you very much for your thoughtful follow-up. We are glad to hear that **our rebuttal has adequately addressed your major concerns.**
> > >
> > > We also appreciate your note on the remaining caveats, and we will make sure to clarify these points in the final revision.
> > >
> > > Thank you again for your time, constructive feedback, and support.
> > >
> > > Best regards,
> > >
> > > The Authors

---

### Decision · Program_Chairs · 2026-04-30

**Decision:**

Accept (spotlight)

**Comment:**

This paper aims for dynamic data selection for LLM pretraining and proposes OPUS (Optimizer-induced Projected Utility Selection), which defines sample utility in the optimizer-induced update space.

The paper claims strong empirical results: GPT-2 Large/XL pre-training on FineWeb/FineWeb-Edu with 30B tokens outperforming baselines and even full 200B-token training, and continued pre-training of Qwen3-8B-Base on SciencePedia achieving superior performance with 0.5B vs. 3B tokens.

The common strengths from the reviewers are

- timely, well-motivated, important, and practical problem (nUdt, gvHH)
- principled formulation: closed-form approximations for the effective update directions for OPUS, which support modern optimizers like AdamW and Muon (nUdt, wrR1)
- strong empirical results (nUdt, bFC4, gvHH, wrR1)
- scalability to large models (nUdt, bFC4)

The common weaknesses are

- hyperparameter sensitivity (nUdt, bFC4, wrR1)
- further experiments and ablations (nUdt, bFC4)
- discussion on post-training data selection (bFC4)
- performance gain compared with Random selection (gvHH)

In summary, two reviewers (nUdt, bFC4) score Accept, and their concerns are fully resolved. The other reviewers (gvHH, wrR1) score Weak Accept, though most of their concerns are resolved, leaving follow-up questions: the performance gain comparison to Random (gvHH), re-evaluation of all methods under multiple debiased protocols (wrR1). The authors sufficiently responded to the reviewers' further questions.

As all reviewers lean toward acceptance, and the weaknesses and remaining questions have been sufficiently addressed, I believe this paper would contribute to efficient data selection.